

# Control design, implementation and evaluation for an in-field 500 kW wind turbine with a fixed-displacement hydraulic drivetrain

Sebastiaan Paul Mulders[1], Niels Frederik Boudewijn Diepeveen[2], and Jan-Willem van Wingerden[1]

[1]Delft Center for Systems and Control, Faculty of Mechanical Engineering, Delft University of Technology, Mekelweg 2, 2628 CD Delft, The Netherlands
[2]DOT B.V., Raam 180, 2611 WP Delft, The Netherlands

**Correspondence:** Sebastiaan Paul Mulders (s.p.mulders@tudelft.nl)

**Abstract.** The business case for compact hydraulic wind turbine drivetrains is becoming ever stronger, as offshore wind turbines are getting larger in terms of size and power output. Hydraulic transmissions are generally employed in high load systems, and form an opportunity for application in multi-megawatt turbines. The Delft Offshore Turbine (DOT) is a hydraulic wind turbine concept replacing conventional drivetrain components with a single seawater pump. Pressurized seawater is directed

5    to a combined Pelton-generator combination on a central multi-megawatt electricity generation platform. This paper presents the control design, implementation and evaluation for an intermediate version of the ideal DOT concept: an in-field 500 kW hydraulic wind turbine. It is shown that the overall drivetrain efficiency and controllability is increased by operating the rotor at maximum rotor torque in the below-rated region using a passive torque control strategy. An active valve control scheme is employed and evaluated in near-rated conditions.

## 1  Introduction

The drivetrain of horizontal-axis wind turbines (HAWTs) generally consists of a rotor-gearbox-generator configuration in the nacelle, which enables each wind turbine to produce and deliver electrical energy independent of other wind turbines. While the HAWT is a proven concept, the turbine rotation speed decreases asymptotically and torque increases exponentially with increasing blade length and power ratings (Burton et al., 2011). As offshore wind turbines are getting ever larger, this results

15    in a lower rotation speed and higher torque at the rotor axis. This inevitably leads to design challenges when scaling up conventional turbine drivetrains for the high-load subsystems (Kotzalas and Doll, 2010). The increased loads primarily affect high-weight components in the turbine such as the generator, bearings and gearbox, and makes repair and replacement a costly and challenging task (Spinato et al., 2009; Ragheb and Ragheb, 2010). Furthermore, due to the contribution of all nacelle components to the total nacelle mass, the complete wind turbine support structure and foundation is designed to carry this

20    weight for the entire expected lifetime, which in turn leads to extra material, weight and thus total cost of the wind turbine (EWEA, 2009; Fingersh et al., 2006).

In an effort to reduce turbine weight, maintenance requirements, complexity, and thus the Levelized Cost Of Energy (LCOE) for offshore wind, hydraulic drivetrain concepts have been considered in the past (Piña Rodriguez, 2012). Integration of hydraulic transmission systems in offshore wind turbines seems to be an interesting opportunity, as they are generally employed




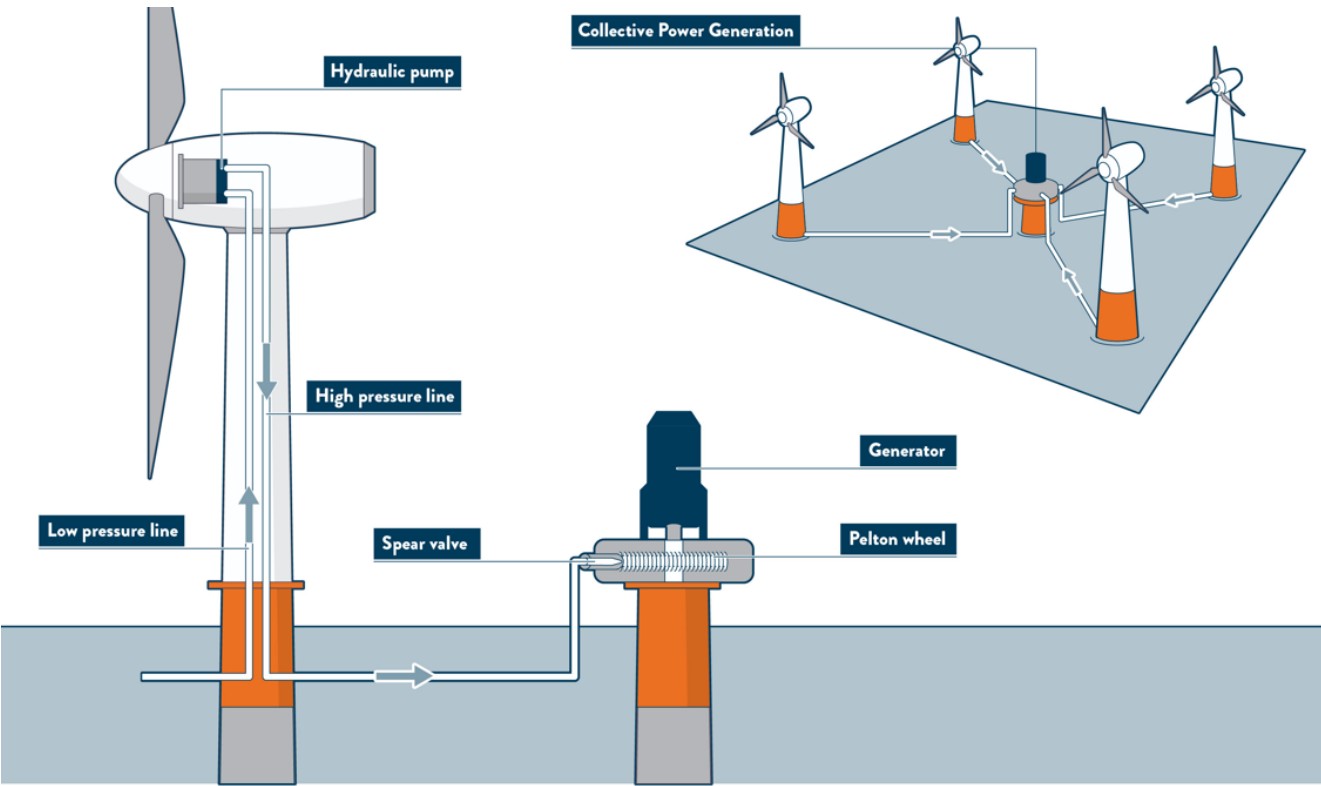

**Figure 1.** Schematic overview of an ideal DOT hydraulic wind turbine configuration. A radial piston seawater pump is coupled to the rotor in the nacelle. The flow is converted to a high-velocity water jet by a spear valve, and a Pelton turbine-generator configuration harvests the hydraulic into electric energy. Multiple turbines can be connected to the central power generation platform.

in high-load applications and have the advantage of a high power-to-weight ratio (Merritt, 1967). It is concluded in (Silva et al., 2014) that hydrostatic transmissions could lower drivetrain costs, improve system reliability and reduce the nacelle mass.

Various hydraulic turbine concepts have been considered in the past. The first 3 MW wind turbine with a hydrostatic power transmission was the SWT3, developed and build from 1976 to 1980 by Bendix (Rybak, 1981). After deployment of the turbine with fourteen fixed-displacement oil pumps in the nacelle and eighteen variable-displacement motors at the tower base, it proved to be overly complex, inefficient and unreliable. With the aim to eliminate the power electronics entirely by application of a synchronous generator, Voith introduced the WinDrive in 2003 and provides a hydraulic gearbox with variable transmission ratio (Muller et al., 2006). In 2004, the ChapDrive hydraulic drivetrain solution was developed with the aim to drive a synchronous generator by a fixed-displacement oil pump and variable displacement oil motor (Chapple et al., 2011; Thomsen et al., 2012). Although the company acquired funding from Statoil for a 5 MW concept, the company ceased operations. In cooperation with Hägglunds, Statoil modelled a drivetrain with hydrostatic transmission, consisting of a single oil pump connected to the rotor and six motors at ground level (Skaare et al., 2011, 2013). Each motor can be enabled or disabled to obtain a discrete variable transmission ratio to drive a synchronous generator. In 2005, Artemis Ltd. developed a




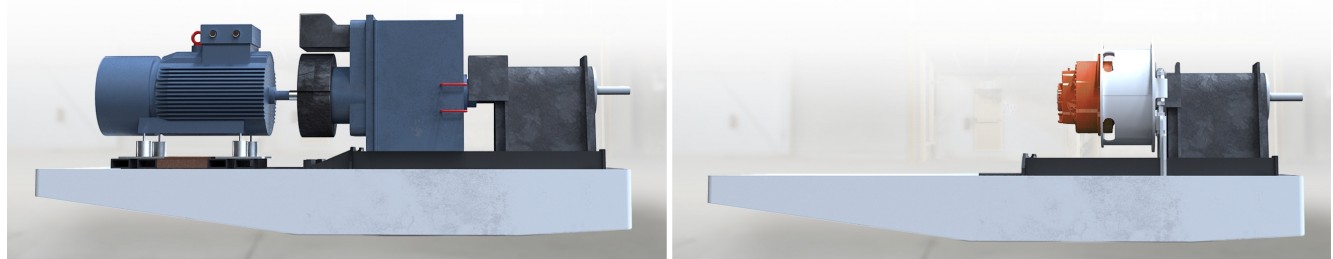

**Figure 2.** The high power-to-weight ratio of hydraulic components and the possibility to abandon power electronics from the nacelle, make the advantages of mass and space reductions in the nacelle self-evident.

digital displacement pump, meaning that it can continuously adjust its volume displacement in a *digital* way by enabling and disabling individual cylinders (Rampen et al., 2006; Artemis Intelligent Power, 2018). In 2010, Mitsubishi acquired Artemis Intelligent Power and started testing of its SeaAngel 7 MW turbine with hydraulic power drive technology in 2014 (Sasaki et al., 2014).

To date, none of the above described full hydraulic concepts made its way to a commercial product. All concepts use oil as the hydraulic medium because of the favorable fluid properties and wide component availability, but therefore also need to operate in closed-loop. The concepts aim to eliminate power electronics from the turbine for use of a synchronous generator, and therefore need a mechanism to vary the hydraulic gear ratio. A novel and patented hydraulic concept with an open-loop drivetrain using seawater as the hydraulic medium is the Delft Offshore Turbine (DOT) (van der Tempel, 2009). This

concept shown in Figure 1, only requires a single seawater pump directly connected to the turbine rotor. The pump replaces components with high-maintenance requirements in the nacelle, which reduces the weight, support structure requirements and turbine maintenance frequency. This effect is clearly visualized in Figure 2. All maintenance critical components are located at sea level, and the centralized generator is coupled to a Pelton turbine. Turbines collectively drive the Pelton turbine to harvest the hydraulic into electrical energy. A feasibility study and modeling of a hydraulic wind turbine based on the DOT concept is

performed in (Diepeveen, 2013). Hydraulic wind turbine networks employing variable displacement components are modeled and simulated in (Jarquin Laguna, 2017). Besides, using the DOT concept, a wind turbine drivetrain to generate electricity and simultaneously extract thermal energy has been proposed in (Buhagiar and Sant, 2014).

     This paper presents the first steps in realizing the integrated hydraulic wind turbine concept, by full-scale prototype tests with a retrofitted Vestas V44 600 kW wind turbine, of which the conventional drivetrain is replaced by a 500 kW hydraulic

configuration. A spear valve is used to control the nozzle outlet area, which in effect influences the fluid pressure in the hydraulic discharge line of the water pump, and forms an alternative way of controlling the reaction torque to the rotor. The main contribution of this paper is to elaborate on modeling and controller design for a hydraulic drivetrain with fixed-displacement components, subject to efficiency and controllability maximization of the system. The control design is based on steady-state and dynamic turbine models, and is subsequently evaluated on the actual in-field retrofitted 500 kW wind turbine.



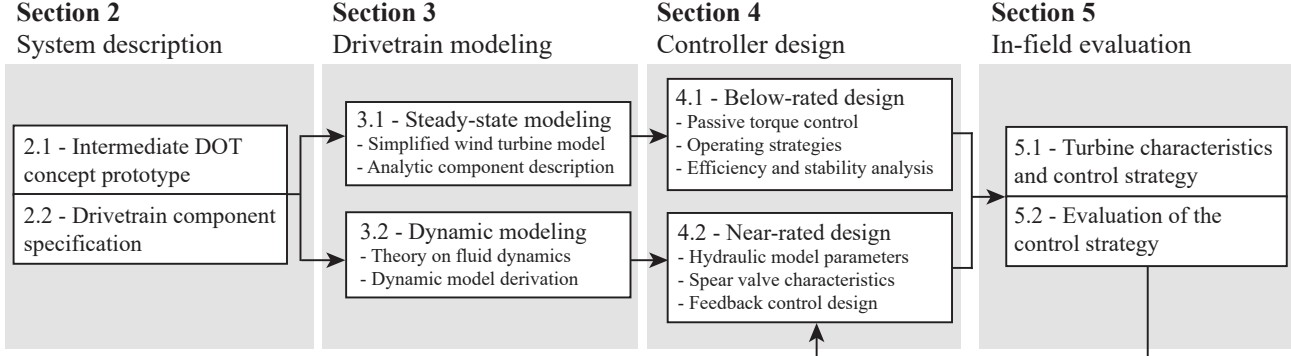

**Figure 3.** Paper organization flow diagram.

The organization of this paper is visualized in Figure 3. In Sect. 2, the DOT configuration used during the in-field tests is explained, and drivetrain components are specified. Section 3, which involves drivetrain modeling, is divided into two parts: a steady-state drivetrain model is derived in Sect. 3.1, and a drivetrain model including fluid dynamics is presented in Sect. 3.2. Controller design is performed in Sect. 4, where the steady-state model is used in Sect. 4.1 for the design of a passive control

5  strategy for below-rated operation, whereas in Sect. 4.2 the dynamic model is used to derive an active control strategy for the region between below- and above-rated operation (near-rated region). Because the in-field tests are performed prior to theoretical model derivation and control design, preliminary conclusions from these tests are incorporated. In Sect. 5, in-field test results are presented for evaluation of the overall control design. Finally, in Sect. 6, conclusions are drawn and an outlook of the DOT concept is given.

## 10  2  The DOT500 - prototype turbine with off-the-shelf components

In this section, the intermediate prototype DOT turbine on which in-field tests are performed is described in Sect. 2.1. Subsequently, the drivetrain components used for the intermediate concept are discussed in Sect. 2.2.

### 2.1  The intermediate DOT500 prototype

At the time of writing, a low-speed high-torque seawater pump required for the ideal DOT concept is not commercially avail-
15  able. This pump is being developed by DOT, enabling the ideal concept in later stages of the project (Nijssen et al., 2018). An intermediate set-up using off-the-shelf components is proposed to speed up development and test the practical feasibility. A visualization of the DOT500 set-up is given in Figure 4.

A Vestas V44 600 kW turbine is used and its drivetrain is retrofitted into a 500 kW hydraulic configuration. The original Vestas V44 turbine is equipped with a conventional drivetrain consisting of the main bearing, a gearbox and a 600 kW three-phase
20  asynchronous generator. The blades are pitched collectively by means of a hydraulic cylinder, driven by a HPU (Hydraulic Power Unit) with a safety pressure accumulator. The drivetrain of the Vestas turbine is replaced by a hydraulic drivetrain. This





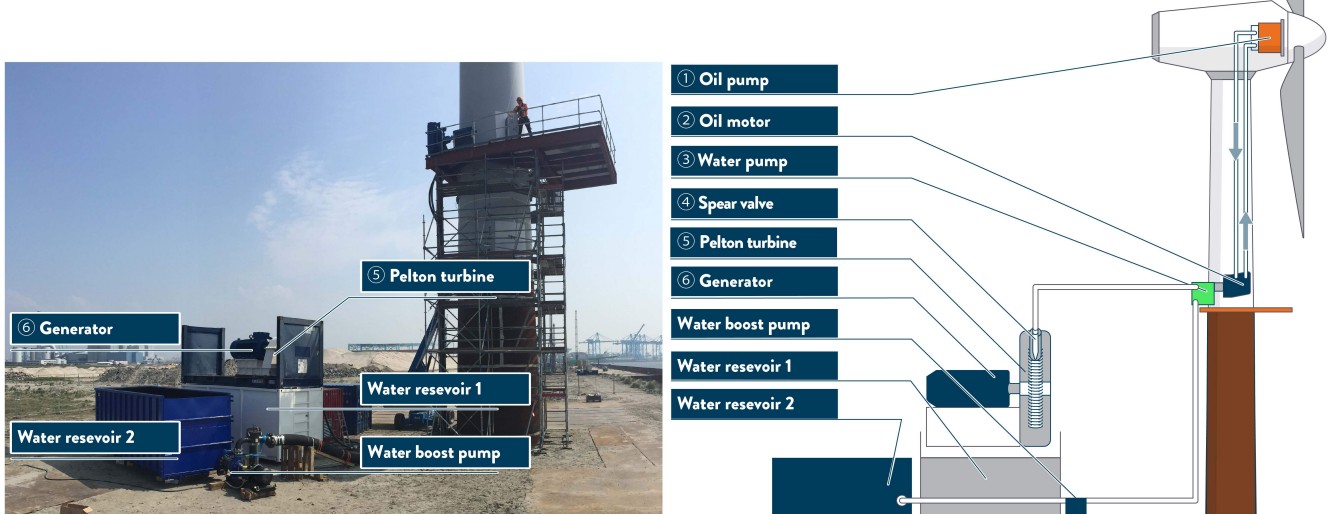

**Figure 4.** Overview of the intermediate DOT500 hydraulic wind turbine configuration. An oil pump is coupled to the rotor low-speed shaft in the nacelle and hydraulically drives an oil motor in the bottom of the tower. This closed oil circuit serves as a hydraulic gearbox between the rotor and water pump. The motor is mechanically coupled to a water pump which produces a pressurized water flow. The flow is converted to high-velocity water jets by spear valves and a Pelton turbine-generator configuration harvests the hydraulic into electric energy.

means that the gearbox and generator are removed from the nacelle, and replaced by a single oil pump coupled to the rotor via the main-bearing.

In the retrofitted hydraulic configuration, a low-speed oil pump is coupled to the rotor and its flow hydraulically drives a high-speed oil motor-water pump combination at the bottom of the turbine tower. The oil circuit acts as a hydraulic gearbox between 5 the rotor and the water pump. The water circuit including Pelton-generator combination is as in the ideal set-up depicted in Figure 1. It is known and taken into account that the additional components and energy conversions result in a reduced overall efficiency. However, this allows for prototyping, and provides a proof of concept for faster development towards the ideal DOT concept. From this point onwards, all discussions and calculations will refer to the intermediate DOT500 set-up, including the described oil circuit (Diepeveen et al., 2018). A prototype was erected in June 2016 at Rotterdam Maasvlakte II, 10 the Netherlands.

## 2.2 Drivetrain component specification

A schematic overview and photograph of the DOT500 set-up is presented in Figure 4. The components are described according to the enumerated labels in the figure, and the drivetrain components specifications are summarized in Table 1

1. **Oil pump:** The rotor drives a Hägglunds CB840, which is a fixed-displacement radial piston motor. The motor is used 15 here as pump and will be referred to as the oil pump in the remainder of this paper. The pump is supplied with a constant



**Table 1.** Hydraulic oil pump, oil motor and water pump specifications (Hägglunds, 2015; Bosch-Rexroth, 2012; KAMAT, 2017)

| Description | Oil pump | Oil motor | Water pump |
|---|---|---|---|
| Brand and type | Hägglunds CB840 | Bosch-Rexroth A6VLM | Kamat K80120G-5M |
| Volumetric displacement | 52.8 l rev$^{-1}$ | 1.0 l rev$^{-1}$ | 2.3 l rev$^{-1}$ |
| Nominal speed | 32 rev min$^{-1}$ | 1600 rev min$^{-1}$ | 1500 rev min$^{-1}$ |
| Torque range available | 0 - 280 kNm | 0 - 5571 Nm | 0 - 5093 Nm |
| Pressure range available | 0 - 350 bar | 0 - 350 bar | 0 - 125 bar |
| Torque range applied | 0 - 210 kNm | 0 - 3000 Nm | 0 - 3000 Nm |
| Pressure range applied | 0 - 230 bar | 0 - 230 bar | 0 - 70 bar |

feed pressure and flow to keep the pistons in continuous contact with the cam ring (Hägglunds, 2015). Load-pins are integrated in the suspension of the pump in the nacelle, which enables measurement of the pump torque.

2. **Oil motor:** The high-pressure hydraulic discharge line of the oil pump drives a Bosch-Rexroth A6VLM variable-displacement axial piston oil motor. The volumetric displacement can be adjusted by a built-in servo (Bosch-Rexroth, 2012). However, in the considered set-up the device is configured to act as a fixed-displacement oil motor, as the DOT concept does not consider variable-displacement components.

3. **Water pump:** The oil motor is mechanically coupled to a Kamat K80120-5G fixed-displacement water plunger pump (KAMAT, 2017). An external centrifugal pump supplies the water pump with sufficient flow and a constant feed pressure.

4. **Spear valve:** The pressure in the water pump discharge line is controlled by variable-area orifices in the form of two spear valves. The valves are used to control the system torque and rotor speed in below- and near-rated operating conditions, and form high-speed water jets towards the Pelton turbine. A schematic visualization of this system is shown in Figure 5. The spear positions are individually actuated by DC-motors, and are measured by resistive potentiometers. The spear valves are actuated in such a way, that only full on-off spear position actuation is possible. This means that either the spear moves forwards or backwards at full speed, or the spear valve position remains at its current position. A deadband logic-controller is implemented to enable position control of the valve within a predefined band around the set point.

5. **Pelton turbine:** When the water flow exits the spear valve, the hydrostatic energy in the high-pressure line is converted to a hydrodynamic high-velocity water jet. The momentum of the jet exerts a force on the Pelton turbine buckets (Zhang, 2007).

6. **Generator:** The mechanical rotational energy of the Pelton turbine is harvested by a mechanically coupled generator. As a grid connection was unavailable at the test location, the electrical energy excess is dissipated by a brake resistor.





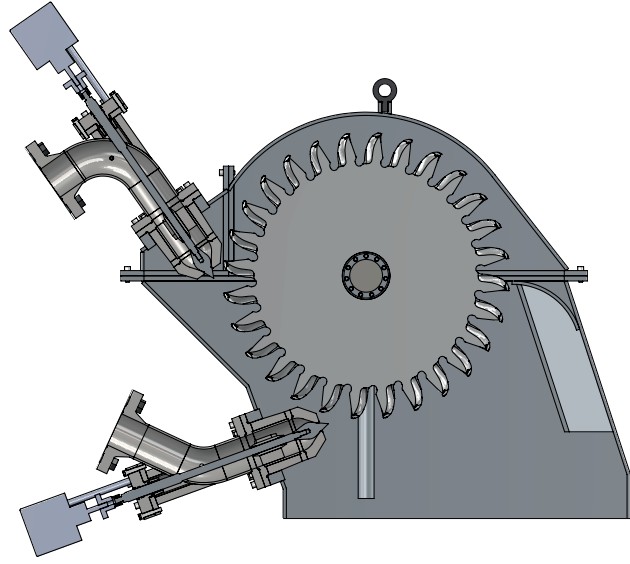

**Figure 5.** The pressurized hydrostatic water flow is converted to a hydrodynamic water jet using spear valves. The high-speed jets exert a force on the buckets of the Pelton turbine.

## 3  Theory and model derivation of the hydraulic drivetrain

The theory for model derivation of the DOT500 hydraulic drivetrain is presented in this section. As control actions influence the turbine operating behavior to the point where the hydrodynamic water jet exits the spear valve, modeling of the turbine drivetrain will be performed up to that point. After the water flow exits the spear valve, the aim to operate the Pelton turbine-
generator combination at maximum efficiency is a decoupled control objective from the rest of the drivetrain, and is outside the scope of this paper.

A simplified hydraulic diagram of the set-up is given in Figure 6. The symbols in this figure are specified throughout the different parts of this section. In this section, pressures are generally given as a pressure difference $\Delta p$ over a component, but when the pressure with respect to the atmospheric pressure $p_0$ is intended, the $\Delta$-indication is omitted.

The organization is divided in two parts. First, a steady-state drivetrain model is derived in Sect. 3.1. This model is used later for derivation of a passive torque control strategy. Secondly, in Sect. 3.2, a drivetrain model including oil line dynamics is derived, and is used for design of an active spear valve control strategy.

### 3.1  Steady-state drivetrain modeling

A steady-state model of the drivetrain is derived for hydraulic torque control design in below-rated operating conditions.
Mathematical models of hydraulic wind turbines have been established, but mostly incorporate a drivetrain with variable displacement components (Buhagiar et al., 2016; Jarquin Laguna, 2017). In (Skaare et al., 2011, 2013) a more simple, robust and efficient drivetrain with discrete hydraulic gear ratio is proposed, by enabling and disabling hydraulic motors. Recently, a



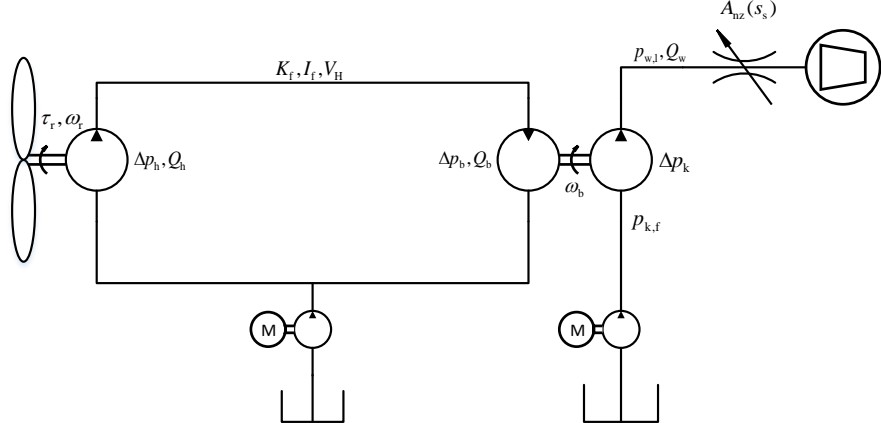

**Figure 6.** Schematic overview of the DOT500 hydraulic wind turbine drivetrain. All pumps and motors have a fixed volume displacement.

feedback control strategy for wind turbines with digital fluid power transmission is described in (Pedersen et al., 2017). However, only fixed-displacement components are considered and modeling of such a DOT drivetrain is described in (Diepeveen, 2013). The model derivation in this section incorporates the components employed in the actual DOT500 set-up. A simplified wind turbine model is introduced in Sect. 3.1.1. This model is complemented in Sect. 3.1.2 by analytic models of drivetrain components.

### 3.1.1 Simplified wind turbine model

The Newton law for rotational motion is employed as a basis for modeling the wind turbine rotor speed dynamics

$$J_r \dot{\omega}_r = \tau_r - \tau_{sys}, \tag{1}$$

where $J_r$ is the rotor inertia, $\omega_r$ the rotor rotational speed, $\tau_r$ the mechanical torque supplied by the rotor to the low-speed shaft, and $\tau_{sys}$ the system torque supplied by the hydraulic oil pump to the shaft. The rotor inertia $J_r$ of the rotor is not publicly available. However, an estimation of the rotor inertia is obtained using an empiric relation on blade length given in (Rodriguez et al., 2007), resulting in a value of $6.6 \cdot 10^5 \, \mathrm{kg \, m^2}$. Moreover, experiments were performed on the actual turbine and confirm this theoretical result (Jager, 2017). The torque supplied by the rotor (Bianchi et al., 2006) is given by

$$\tau_r = \frac{1}{2} \rho_{air} \pi R^3 U^2 C_p(\lambda, \beta) / \lambda, \tag{2}$$

where the density of air $\rho_{air}$ is taken as a constant value of $1.225 \, \mathrm{kg \, m^{-3}}$, $U$ is the velocity of the upstream wind, and $R$ is the blade length of $22 \, \mathrm{m}$. The power coefficient $C_p$ represents the fraction between the captured rotor power $P_r$ and the available wind power $P_{wind}$, and is a function of the blade pitch angle $\beta$ and the dimensionless tip-speed ratio $\lambda$ given by

$$\lambda = \omega_r R / U. \tag{3}$$





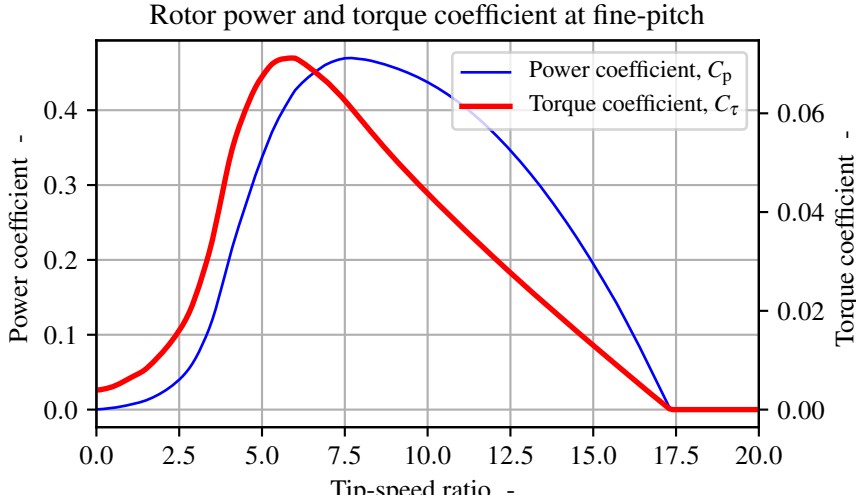

**Figure 7.** Rotor power and torque coefficient curve of the rotor, obtained from a BEM analysis performed on measured blade-geometry data. The maximum power coefficient $C_{p,max}$ of 0.48 is attained at a tip-speed ratio of 7.8. The maximum torque coefficient of $C_{\tau,max}$ is given by $7.2 \cdot 10^{-2}$ at a lower tip-speed ratio of 5.9.

The power coefficient $C_p$ is related to the torque coefficient

$$C_\tau(\lambda,\beta) = C_p(\lambda,\beta)/\lambda, \tag{4}$$

such that Eq. (2) can be rewritten as

$$\tau_r = \frac{1}{2}\rho_{air}\pi R^3 U^2 C_\tau(\lambda,\beta). \tag{5}$$

The rotor power and torque extraction capabilities from the wind are characterized in respective power and torque coefficient curves. These curves of the actual DOT500 rotor are generated by mapping the actual blade airfoils, and applying Blade Element Momentum (BEM) theory (Burton et al., 2011), and are given in Figure 7 at the blade fine-pitch angle. The fine-pitch angle $\beta_0$ indicates the blade angle resulting in maximum rotor power extraction in the below-rated operating region (Bossanyi, 2000). The theoretical maximum rotor power and torque coefficients equal $C_{p,max} = 0.48$ and $C_{\tau,max} = 7.2 \cdot 10^{-2}$, at tip-speed

ratios of 7.8 and 5.9, respectively.

    The system torque $\tau_{sys}$ is supplied by the hydraulic drivetrain to the rotor low-speed shaft. This torque is influenced by the components in the drivetrain, which all have their own energy conversion characteristics expressed in efficiency curves. All components are off-the-shelf and their combined efficiency characteristics influence the operating behavior of the turbine. As hydraulic components are known for their high torque-to-inertia ratio, this results in high acceleration capabilities (Merritt,

1967). Therefore, it is assumed in this paper that the pumps and motor included in the drivetrain have negligible dynamics, and the power conversion (flow-speed, torque-pressure) is given by static relations. This assumption is supported by an analysis





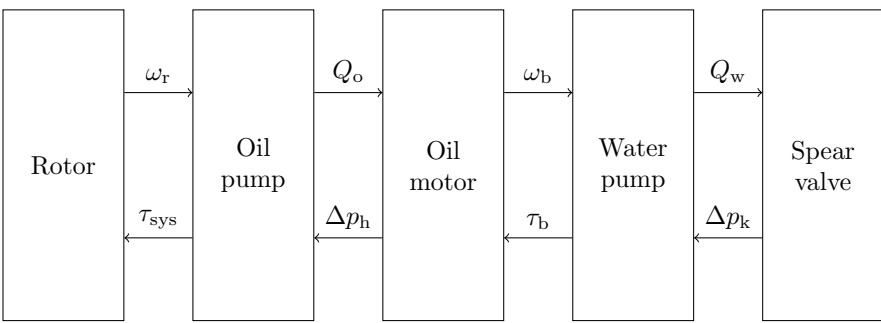

**Figure 8.** Flow diagram of the DOT500 hydraulic drivetrain. For steady-state modeling purposes, first the flow path is calculated up to the spear valve. The effective nozzle area and the water flow through the spear valve determine the hydraulic feed line pressure, which influences the system torque $\tau_\mathrm{sys}$ to the rotor.

carried out in (Kempenaar, 2012; Diepeveen, 2013), where it is concluded that inclusion of component dynamics does not result in significantly improved model accuracy.

### 3.1.2 Analytic drivetrain components description

A flow diagram of the modeling strategy is presented in Figure 8. To obtain an expression for the system torque $\tau_\mathrm{sys}$, the
complete hydraulic flow path with its volumetric losses is modeled first. When the flow path reaches the spear valve at the water discharge to the Pelton turbine, the simulation path is reversed to calculate the effect of all component characteristics to the line pressures. The spear valve allows for control of the water discharge pressure, of which the effect propagates back to the system torque $\tau_\mathrm{sys}$. The high-pressure oil flow by the oil pump is proportional to the rotor speed

$$Q_\mathrm{o} = V_\mathrm{p,h}\omega_\mathrm{r}\eta_\mathrm{v,h}, \tag{6}$$

where the $V_\mathrm{p,h}$ is the pump volumetric displacement, and $\eta_\mathrm{v,h}$ the volumetric efficiency. The volumetric efficiency indicates the volume loss as a fraction of the total displaced flow, and decreases as the pressure increases. However, the volumetric efficiency of a pump or motor is generally high and fairly constant over the entire operating range (Diepeveen, 2013; Trostmann, 1995). Therefore, volumetric losses will be taken constant in this paper. The resulting oil flow $Q_\mathrm{o}$ drives the oil motor, which result in a rotational speed of the motor shaft subject to volumetric losses

$$\omega_\mathrm{b} = \frac{Q_\mathrm{o}}{V_\mathrm{p,b}}\eta_\mathrm{v,b}, \tag{7}$$

where $V_\mathrm{p,b}$ is the oil motor volumetric displacement, and $\eta_\mathrm{v,b}$ the volumetric efficiency of the oil motor. As the water pump is mechanically coupled to the oil motor axis, its rotational speed is equal to $\omega_\mathrm{b}$. The expression relating the rotational speed to the water pump discharge water flow $Q_\mathrm{w}$ is given by

$$Q_\mathrm{w} = V_\mathrm{p,k}\omega_\mathrm{b}\eta_\mathrm{v,k}, \tag{8}$$





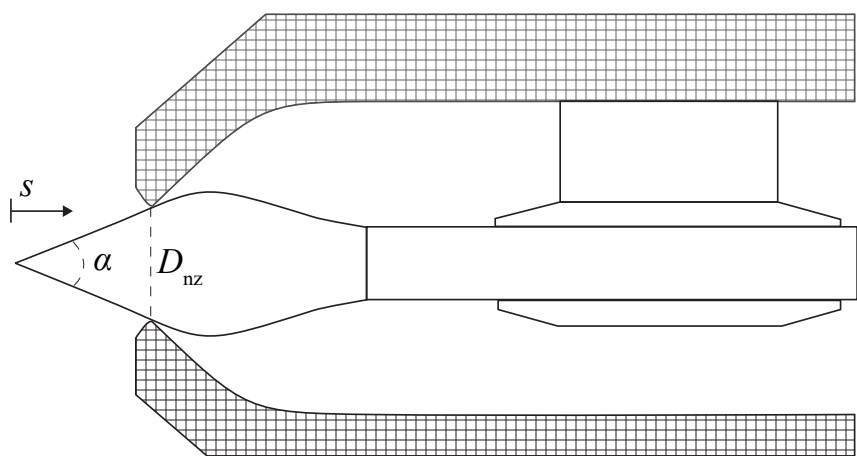

**Figure 9.** Cross-section of the spear valve used. The coning angle of the spear is given by $\alpha$, the nozzle diameter by $D_{\mathrm{nz}}$, and the position of the spear tip in the nozzle by $s$. The nozzle heads are adjustable for adjustment of the outlet area.

where $V_{\mathrm{p,k}}$ is the volumetric displacement of the water pump, and $\eta_{\mathrm{v,k}}$ the volumetric efficiency of the water pump. The pressure in the water discharge line is controlled by a spear valve of which a visualization is given in Figure 9, with the spear position coordinate system. The effective nozzle area is variable according to the relation

$$A_{\mathrm{nz}}(s) = N_{\mathrm{s}}\pi \left( D_{\mathrm{nz}}^2/4 - (s_{\max} - s)^2 \tan^2(\alpha/2) \right), \tag{9}$$

where $\{s \subset \mathbb{R} \,|\, 0 \leq s \leq s_{\max}\}$ represents the position of the spear in the circular nozzle cross-section, $D_{\mathrm{nz}}$ is the nominal nozzle diameter, $\alpha$ the spear coning angle, and $N_{\mathrm{s}}$ indicates the amount of spear valves on the same line. Modeling multiple spear valves by $N_{\mathrm{s}}$ assumes equal effective nozzle areas for all valves. The maximum spear position (fully open) is given by

$$s_{\max} = \frac{D_{\mathrm{nz}}}{2\tan(\alpha/2)}. \tag{10}$$

and a mapping for spear position to effective nozzle area for different nozzle diameters $D_{\mathrm{nz}}$ is given in Figure 10. The spear
valve is closed for all cases at position $s = 0\,\mathrm{mm}$. The spear valve converts the hydrostatic water flow into a high-speed hydrodynamic water jet, that exerts a thrust force on the buckets of the Pelton turbine (Zhang, 2007). To describe this energy conversion, the Bernoulli equation for incompressible flows (White, 2011) is used

$$\frac{1}{2}\rho_{\mathrm{w}}v_{\mathrm{w,l}}^2 + \rho_{\mathrm{w}}gz_{\mathrm{w,l}} + p_{\mathrm{w,l}} = \frac{1}{2}\rho_{\mathrm{w}}v_{\mathrm{w,nz}}^2 + \rho_{\mathrm{w}}gz_{\mathrm{w,nz}} + p_{\mathrm{w,nz}}, \tag{11}$$

where $v_{\mathrm{w}}$ represents the water velocity, $p_{\mathrm{w}}$ the water pressure, and the subscripts $(\cdot)_{\mathrm{w,l}}$ and $(\cdot)_{\mathrm{w,nz}}$ indicate the locations before
and after the nozzle, respectively. As in the given control volume the height difference $\Delta z$ equals zero, and the water density $\rho_{\mathrm{w}}$ and gravitational constant $g$ are constant, the potential term is disregarded. Using Eq. (11), an expression for the discharge



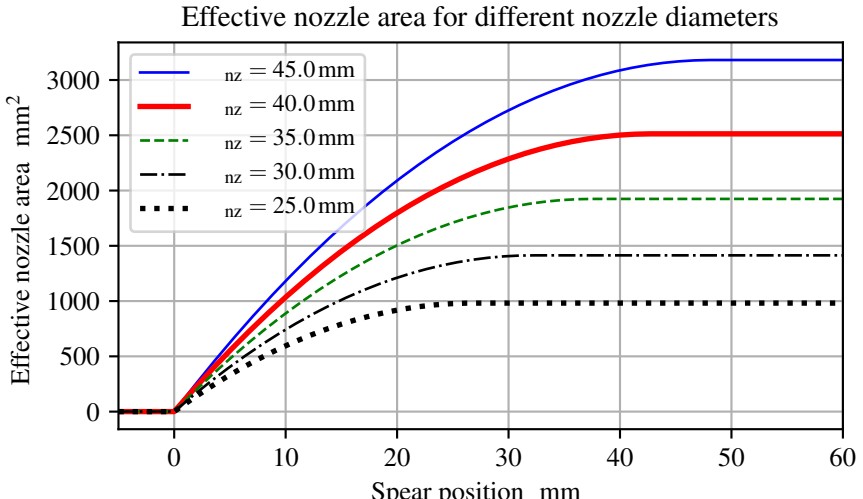

**Figure 10.** Effective nozzle area as function of the spear position in the circular nozzle cross-section. The spear valve is fully closed at $s = 0\,\mathrm{mm}$ and fully opened at $s = s_{\mathrm{max}}$, of which the latter is variable according to the nozzle head diameter.

water pressure $p_{\mathrm{w,l}}$ is obtained

$$p_{\mathrm{w,l}}(s) = \frac{\rho_{\mathrm{w}}}{2} \left( \frac{Q_{\mathrm{w}}}{C_{\mathrm{d}} A_{\mathrm{nz}}(s)} \right)^2, \tag{12}$$

where the flow and effective nozzle area $\{Q_{\mathrm{w}}, A_{\mathrm{nz}}\} \subset \mathbb{R}^+$. As observed in the above given relation, the pressure can be controlled by varying the feed flow and spear position, as the latter influences the effective nozzle area $A_{\mathrm{nz}}$. The discharge

5  coefficient $C_{\mathrm{d}}$ is introduced to account for pressure losses due to the geometry and flow regime at the nozzle exit (Al'tshul' and Margolin, 1968). The discharge coefficient of an orifice is defined as the ratio between the vena contracta area and the orifice area (Bragg, 1960). The vena contracta is the point where the streamlines become parallel, which usually occurs downstream of the orifice where the streamlines are still converging.

The pressure in the water discharge line propagates back into the system, and can be used as a substitute to conventional

10  wind turbine torque control. A relation for the mechanical torque at the axis between the water pump and oil motor is given by

$$\tau_{\mathrm{b}} = \frac{V_{\mathrm{p,k}} \Delta p_{\mathrm{k}}}{\eta_{\mathrm{m,k}}(\omega_{\mathrm{b}}, \Delta p_{\mathrm{k}})}, \tag{13}$$

where $\eta_{\mathrm{m,k}}$ is the mechanical efficiency of the water pump as function of the rotational speed and pressure difference over the pump

$$\Delta p_{\mathrm{k}} = p_{\mathrm{w,l}} - p_{\mathrm{k,f}}, \tag{14}$$





where $p_{k,f}$ is a known and constant feed pressure. The torque $\tau_b$ is used to calculate the pressure difference over the oil motor and pump by

$$\Delta p_b = \Delta p_h = \frac{\tau_b}{V_{p,b}\eta_{m,b}(\omega_b, \tau_b)}, \tag{15}$$

where $\eta_{m,b}$ is the mechanical efficiency of the oil motor. It is assumed that the pressure at the discharge outlet of the oil motor

$\Delta p_b$ is constant as the feed pressure to the oil pump is regulated. Finally, the system torque supplied to the rotor low-speed shaft is given by

$$\tau_{sys} = \frac{V_{p,h}\Delta p_h}{\eta_{m,h}(\omega_r, \Delta p_h)}. \tag{16}$$

Using the relations derived, a passive strategy for below-rated turbine control is presented in Sect. 4.1.

## 3.2 Dynamic drivetrain modeling

In contrast to the steady-state model presented previously, this section elaborates on the derivation of a drivetrain model including fluid dynamics for validation of the control design in the near-rated region. First, preliminary knowledge on fluid dynamics is given in Sect. 3.2.1, whereafter a dynamic DOT500 drivetrain model is presented in Sect. 3.2.2.

### 3.2.1 Analysis of fluid dynamics in a hydraulic line

The dynamics of a volume in a hydraulic line are modeled in this section. For this, analogies between mechanical and hydraulic

systems are employed for modeling convenience. The system considered, representing the high-pressure discharge oil line, is a cylindrical control volume $V_H = AL_l = \pi r_l^2 L_l$, with radius $r_l$ and length $L_l$, excited by a pressure $\Delta p = p_{in} - p_{out}$. The net flow into the control volume is defined as $Q = Q_{in} - Q_{out}$. First, the differential equation for a standard mass-damper-spring system driven by an external force $F$ is given by

$$F = m\ddot{x} + c\dot{x} + kx. \tag{17}$$

For conversion to a hydraulic equivalent expression, the driving mechanical force is substituted by $F = \Delta pA$, the control volume mass is taken as $m = \rho V_H = \rho AL_l$, and the fluid inflow velocity defined as $\dot{x} = Q/A$. By rearranging terms, one obtains

$$\Delta p = \frac{\rho L_l}{A}\dot{Q} + \frac{c}{A^2}Q + \frac{k}{A^2}\int Qdt, \tag{18}$$

which is further simplified into

25 $$\Delta p = L_H\dot{Q} + R_H Q + \frac{1}{C_H}\int Qdt, \tag{19}$$

where $L_H$, $R_H$ and $C_H$ are the hydraulic induction, resistance and capacitance (Esposito, 1969), respectively, and are defined in Appendix A. The former two of these three quantities depend on the flow Reynolds number, which shows whether the inertial



or viscosity terms are dominant in the Navier-Stokes equations (Merritt, 1967). The Reynolds number is defined as

$$Re = \frac{D_l v \rho}{\mu}, \tag{20}$$

where $D_l = 2r_l$ is the line diameter, and $\mu$ the fluid dynamic viscosity. For Reynolds numbers larger than $4000$ the flow is considered as turbulent and the inertial terms are dominant, whereas for values smaller than $2300$ the viscosity terms are

deemed dominant.

For evaluation of the natural frequency $\omega_n$ and damping ratio $\zeta$ for the considered system, the characteristic equation by neglecting the external excitation force ($\Delta p = 0$) is defined as

$$0 = \dot{Q} + \frac{R_H}{L_H} Q + \frac{1}{C_H L_H} \int Q dt \tag{21}$$

$$= \dot{Q} + 2\zeta\omega_n Q + \omega_n^2 \int Q dt. \tag{22}$$

Evaluating the quantities $\omega_n$ and $\zeta$, results in

$$\omega_n = \sqrt{\frac{1}{C_H L_H}}, \tag{23}$$

$$\zeta_p = \frac{R_H}{2} \sqrt{\frac{C_H}{L_H}}. \tag{24}$$

The inverse result of Eq. (19) is obtained (Murrenhoff, 2012), with flow $Q$ as the external excitation and $\Delta p$ as output

$$Q = C_H \Delta\dot{p} + \frac{1}{R_H} \Delta p + \frac{1}{L_H} \int \Delta p dt. \tag{25}$$

Now by evaluating the characteristic equation

$$0 = C_H \Delta\dot{p} + \frac{1}{R_H} \Delta p + \frac{1}{L_H} \int \Delta p dt \tag{26}$$

$$= \Delta\dot{p} + \frac{1}{R_H C_H} \Delta p + \frac{1}{L_H C_H} \int \Delta p dt, \tag{27}$$

and using Eq. (22), it is seen that the natural frequency remains unchanged with the result obtained in Eq. (23), but the definition of the damping ratio changes

$$\zeta_Q = \frac{1}{2R_H} \sqrt{\frac{L_H}{C_H}}. \tag{28}$$

Finally, the differential equation defined by Eq. (19) is expressed as a transfer function in

$$G_{Q/\Delta_P}(s) = \frac{1/L_H}{s + (R_H/L_H) + 1/(C_H L_H s)} \tag{29}$$

$$= \frac{s/L_H}{s^2 + (R_H/L_H)s + 1/(C_H L_H)}, \tag{30}$$

and the same is done for Eq. (25)

$$G_{\Delta_P/Q}(s) = \frac{1/C_H}{s + 1/(R_H C_H) + 1/(C_H L_H s)} \tag{31}$$

$$= \frac{s/C_H}{s^2 + 1/(R_H C_H)s + 1/(C_H L_H)}. \tag{32}$$



The transfer functions defined in Eq.s (30) and (32) show the characteristics of an inverted notch with $+1$ and $-1$ slopes on the left and right side of the natural frequency, respectively. This physically means that exciting the system pressure results in a volume velocity change predominantly at the system natural frequency for the former mentioned case, and vice versa for the latter mentioned.

### 3.2.2 Drivetrain model derivation

A dynamic model of the DOT500 drivetrain is derived by application of the theory presented in the previous section. The drivetrain is defined from the rotor up to the spear valve, and the following assumptions are made:

- Because of the high torque to inertia ratio of hydraulic components (Merritt, 1967), the dynamics of oil pumps and motors are disregarded and taken as analytic relations;
- Because of the longer line length and higher compressibility of oil compared to the shorter water column, the high-pressure oil line is more critical for control design, and a dynamic model will be implemented for this column only;
- Hydraulic lines are modeled as being rigid;
- The fluids have a constant temperature.

The dynamic system is governed by the following differential equations

$$\mathcal{V} = \mathcal{V}_{\text{in}} - \mathcal{V}_{\text{out}}, \qquad \dot{\mathcal{V}} = Q = Q_{\text{in}} - Q_{\text{out}}, \tag{33}$$

$$\Delta p_{\text{h}} = \underbrace{\left(\frac{J_{\text{r}}\eta_{\text{m,h}}}{V_{\text{p,h}}^2\eta_{\text{v,h}}} + L_{\text{H}}\right)}_{L_{\text{R}}^*}\dot{Q}_{\text{in}} + R_{\text{H}}(Q_{\text{in}} - Q_{\text{out}}) + \frac{K_{\text{f}}}{V_{\text{H}}}(\mathcal{V}_{\text{in}} - \mathcal{V}_{\text{out}}) = L_{\text{R}}^*\dot{Q}_{\text{in}} + R_{\text{H}}(Q_{\text{in}} - Q_{\text{out}}) + \frac{1}{C_{\text{H}}}\mathcal{V}, \tag{34}$$

$$\Delta p_{\text{b}} = L_{\text{H}}\dot{Q}_{\text{out}} + R_{\text{H}}(Q_{\text{out}} - Q_{\text{in}}) + \frac{K_{\text{f}}}{V_{\text{H}}}(\mathcal{V}_{\text{out}} - \mathcal{V}_{\text{in}}) = L_{\text{H}}\dot{Q}_{\text{out}} + R_{\text{H}}(Q_{\text{out}} - Q_{\text{in}}) - \frac{1}{C_{\text{H}}}\mathcal{V}, \tag{35}$$

where $\mathcal{V}$ is the net volume inflow to the considered oil line, between the oil pump discharge and oil motor feed port. For convenience, mechanical model quantities are expressed hydraulically in terms of fluid flows and pressure differences over the components. Therefore, the rotor inertia $J_{\text{r}}$ is expressed in terms of fluid induction, and is combined with the hydraulic induction term into $L_{\text{R}}^*$.

Both the spear position and pitch angle are modeled by a first-order actuator model

$$\dot{s} = \frac{1}{t_{\text{s}}}(s_{\text{ref}} - s), \tag{36}$$

$$\dot{\beta} = \frac{1}{t_{\beta}}(\beta_{\text{ref}} - \beta), \tag{37}$$

where $t_{\text{s}}$ and $t_{\beta}$ are the time constant for the spear valve and pitch actuators, respectively, and the phase loss at the actuator bandwidth is assumed to account for actuation delay effects.





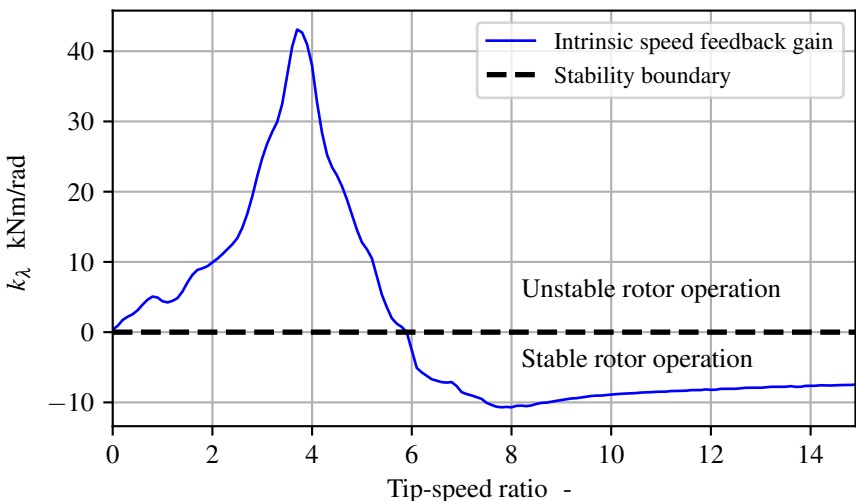

**Figure 11.** The intrinsic speed feedback gain $k_\lambda(\lambda, \bar\beta, \bar U)$ as function of tip-speed ratio $\lambda$, at a fixed pitch angle and wind speed of $-2\,\mathrm{deg}$ and $8\,\mathrm{m\,s^{-1}}$. Stable turbine operation is attained for non-positive values of $k_\lambda$.

The above given dynamic equations are written in a state-space representation as

$$
\begin{bmatrix}
\dot{\mathcal{V}} \\
\dot{Q}_{\mathrm{in}} \\
\dot{Q}_{\mathrm{out}} \\
\dot{s} \\
\dot{\beta}
\end{bmatrix}
=
\begin{bmatrix}
0 & 1 & -1 & 0 & 0 \\
-\frac{1}{C_{\mathrm{H}} L_{\mathrm{R}}^*} & -\frac{R_{\mathrm{H}}}{L_{\mathrm{R}}^*} & \frac{R_{\mathrm{H}}}{L_{\mathrm{R}}^*} & 0 & 0 \\
\frac{1}{C_{\mathrm{H}} L_{\mathrm{H}}} & \frac{R_{\mathrm{H}}}{L_{\mathrm{H}}} & -\frac{R_{\mathrm{H}}}{L_{\mathrm{H}}} & 0 & 0 \\
0 & 0 & 0 & -\frac{1}{t_{\mathrm{s}}} & 0 \\
0 & 0 & 0 & 0 & -\frac{1}{t_\beta}
\end{bmatrix}
\begin{bmatrix}
\mathcal{V} \\
Q_{\mathrm{in}} \\
Q_{\mathrm{out}} \\
s \\
\beta
\end{bmatrix}
+
\begin{bmatrix}
0 \\
\frac{1}{L_{\mathrm{R}}^*} \Delta p_{\mathrm{h}} \\
-\frac{1}{L_{\mathrm{H}}} \Delta p_{\mathrm{b}} \\
\frac{1}{t_{\mathrm{s}}} s_{\mathrm{ref}} \\
\frac{1}{t_\beta} \beta_{\mathrm{ref}}
\end{bmatrix}.
\tag{38}
$$

It is seen that the pressure difference over the oil pump and motor appear as inputs, but these quantities cannot be controlled directly. For this reason, linear expressions of the rotor torque and spear valves are defined next. The rotor torque is linearized

5   with respect to the tip-speed ratio, pitch angle and wind speed

$$
\hat\tau_{\mathrm{r}}(\bar\omega_{\mathrm{r}}, \bar\beta, \bar U) = k_{\omega_{\mathrm{r}}}(\bar\omega_{\mathrm{r}}, \bar\beta, \bar U)\hat\omega_{\mathrm{r}} + k_\beta(\bar\omega_{\mathrm{r}}, \bar\beta, \bar U)\hat\beta + k_{\mathrm{U}}(\bar\omega_{\mathrm{r}}, \bar\beta, \bar U)\hat U,
\tag{39}
$$

where $\hat{(\cdot)}$ indicates a value deviation from the operating point, and $\bar{(\cdot)}$ is the value at the operating point (Bianchi et al., 2006). Furthermore,

$$
k_{\omega_{\mathrm{r}}}(\omega_{\mathrm{r}}, \beta, U) = \frac{\partial \tau_{\mathrm{r}}}{\partial \omega_{\mathrm{r}}} = c_{\mathrm{r}} R U \frac{\partial C_\tau(\omega_{\mathrm{r}} R/U, \beta)}{\partial \lambda},
\tag{40}
$$

$$
k_\beta(\omega_{\mathrm{r}}, \beta, U) = \frac{\partial \tau_{\mathrm{r}}}{\partial \beta} = c_{\mathrm{r}} U^2 \frac{\partial C_\tau(\omega_{\mathrm{r}} R/U, \beta)}{\partial \beta},
\tag{41}
$$





$$k_{\mathrm{U}}(\omega_{\mathrm{r}},\beta,U) = \frac{\partial \tau_{\mathrm{r}}}{\partial U} = 2c_{\mathrm{r}}U\,C_{\tau}(\omega_{\mathrm{r}}R/U,\beta) + c_{\mathrm{r}}U^2\frac{\partial C_{\tau}(\omega_{\mathrm{r}}R/U,\beta)}{\partial \lambda}\frac{\partial \lambda}{\partial U} \tag{42}$$

$$= 2c_{\mathrm{r}}U\,C_{\tau}(\omega_{\mathrm{r}}R/U,\beta) - c_{\mathrm{r}}\omega_{\mathrm{r}}R\frac{\partial C_{\tau}(\omega_{\mathrm{r}}R/U,\beta)}{\partial \lambda}, \tag{43}$$

$$c_{\mathrm{r}} = \frac{1}{2}\rho\pi R^3, \tag{44}$$

where the quantities $k_{\omega_{\mathrm{r}}}$, $k_{\beta}$ and $k_{\mathrm{U}}$ represent the intrinsic speed feedback gain, the linear pitch gain and the linear wind speed gain, respectively. The intrinsic speed feedback gain can also be expressed as a function of the tip-speed ratio by

$$k_{\lambda}(\lambda,\beta,U) = k_{\omega_{\mathrm{r}}}(\omega_{\mathrm{r}},\beta,U)\frac{U}{R}. \tag{45}$$

For aerodynamic rotor stability, the value of $k_{\lambda}$ needs to be negative. In Figure 11 the intrinsic speed feedback gain $k_{\lambda}(\lambda,\bar{\beta},\bar{U})$ is evaluated as a function of the tip-speed ratio at the fine-pitch angle $\beta_0$. For incorporation of the linearized rotor torque in the drivetrain model, Eq. (39) is expressed in the pressure difference over the oil pump

$$\Delta\hat{p}_{\mathrm{h}}(\bar{\omega}_{\mathrm{r}},\bar{\beta},\bar{U}) = k_{\mathrm{Q_{in}}}^*(\bar{\omega}_{\mathrm{r}},\bar{\beta},\bar{U})\hat{Q}_{\mathrm{in}} + k_{\beta}^*(\bar{\omega}_{\mathrm{r}},\bar{\beta},\bar{U})\hat{\beta} + k_{\mathrm{U}}^*(\bar{\omega}_{\mathrm{r}},\bar{\beta},\bar{U})\hat{U}, \tag{46}$$

where the conversions of the required quantities are given by

$$k_{\mathrm{Q_{in}}}^* = k_{\omega_{\mathrm{r}}}\frac{\eta_{\mathrm{m,h}}}{V_{\mathrm{p,h}}^2\eta_{\mathrm{v,h}}}, \qquad k_{\beta}^* = k_{\beta}\frac{\eta_{\mathrm{m,h}}}{V_{\mathrm{p,h}}}, \qquad k_{\mathrm{U}}^* = k_{\mathrm{U}}\frac{\eta_{\mathrm{m,h}}}{V_{\mathrm{p,h}}}. \tag{47}$$

Similarly, the water line pressure as defined in Eq. (12) is linearized with respect to the spear position and flow through the valve

$$\hat{p}_{\mathrm{w,l}}(\hat{Q}_{\mathrm{w}},\hat{s}) = k_{\mathrm{s,s}}(\bar{Q}_{\mathrm{w}},\bar{s})\hat{s} + k_{\mathrm{s,Q_w}}(\bar{Q}_{\mathrm{w}},\bar{s})\hat{Q}_{\mathrm{w}}, \tag{48}$$

where

$$k_{\mathrm{s,s}}(\bar{Q}_{\mathrm{w}},\bar{s}) = \left.\frac{2Q_{\mathrm{w}}^2\rho_{\mathrm{w}}(s-s_{\max})\tan^2(\alpha/2)}{C_{\mathrm{d}}^2 N_{\mathrm{s}}^2\pi^2\left(D_{\mathrm{nz}}^2/4 - (s_{\max}-s)^2\tan^2(\alpha/2)\right)^3}\right|_{\bar{Q}_{\mathrm{w}},\bar{s}}, \tag{49}$$

$$k_{\mathrm{s,Q_w}}(\bar{Q}_{\mathrm{w}},\bar{s}) = \left.\frac{Q_{\mathrm{w}}\rho_{\mathrm{w}}}{C_{\mathrm{d}}^2 N_{\mathrm{s}}^2\pi^2\left(D_{\mathrm{nz}}^2/4 - (s_{\max}-s)^2\tan^2(\alpha/2)\right)^2}\right|_{\bar{Q}_{\mathrm{w}},\bar{s}}. \tag{50}$$

The pressure difference over the oil motor is defined in terms of the water line pressure which is a function of the spear position

$$\Delta\hat{p}_{\mathrm{b}} = \frac{1}{c_{\mathrm{m,bk}}}\Delta\hat{p}_{\mathrm{k}} \approx \frac{1}{c_{\mathrm{m,bk}}}\hat{p}_{\mathrm{w,l}}(s) = \frac{1}{c_{\mathrm{m,bk}}}\left(k_{\mathrm{s,s}}(\bar{Q}_{\mathrm{w}},\bar{s})\hat{s} + k_{\mathrm{s,Q_w}}(\bar{Q}_{\mathrm{w}},\bar{s})\hat{Q}_{\mathrm{w}}\right), \tag{51}$$

where the mechanical and volumetric conversion factors from oil to water pressure and flow are defined as

$$c_{\mathrm{m,bk}} = \frac{V_{\mathrm{p,b}}}{V_{\mathrm{p,k}}}\eta_{\mathrm{m,k}}\eta_{\mathrm{m,b}}, \qquad c_{\mathrm{v,bk}} = \frac{V_{\mathrm{p,k}}}{V_{\mathrm{p,b}}}\eta_{\mathrm{v,k}}\eta_{\mathrm{v,b}} \tag{52}$$





The system defined in Eq. (38) is now presented as a linear state-space system of the form

$$\dot{x} = Ax + Bu + B_{\mathrm{U}}\hat{U} \tag{53}$$

$$y = Cx,$$

and by substitution of the rotor torque and water pressure approximations defined in Eq.s (46) and (51), the state $A$, input $B$,
input disturbance $B_{\mathrm{U}}$ and output $C$ matrices are given by

$$
A = \begin{bmatrix}
0 & 1 & -1 & 0 & 0 \\
-\frac{1}{C_{\mathrm{H}}L_{\mathrm{R}}^*} & -\frac{R_{\mathrm{H}}-k_{Q_{\mathrm{in}}}^*}{L_{\mathrm{R}}^*} & \frac{R_{\mathrm{H}}}{L_{\mathrm{R}}^*} & 0 & \frac{k_\beta^*}{L_{\mathrm{R}}^*} \\
\frac{1}{C_{\mathrm{H}}L_{\mathrm{H}}} & \frac{R_{\mathrm{H}}}{L_{\mathrm{H}}} & -\frac{1}{L_{\mathrm{H}}}\left(R_{\mathrm{H}} + \frac{c_{\mathrm{v,bk}}}{c_{\mathrm{m,bk}}}k_{\mathrm{s,Q_w}}\right) & -\frac{1}{c_{\mathrm{m,bk}}L_{\mathrm{H}}}k_{\mathrm{s,s}} & 0 \\
0 & 0 & 0 & -\frac{1}{t_{\mathrm{s}}} & 0 \\
0 & 0 & 0 & 0 & -\frac{1}{t_\beta}
\end{bmatrix}, \qquad
B = \begin{bmatrix}
0 & 0 \\
0 & 0 \\
0 & 0 \\
\frac{1}{t_{\mathrm{s}}} & 0 \\
0 & \frac{1}{t_\beta}
\end{bmatrix},
$$

$$
B_{\mathrm{U}} = \begin{bmatrix}
0 \\
\frac{k_{\mathrm{U}}^*}{L_{\mathrm{R}}^*} \\
0 \\
0 \\
0
\end{bmatrix}, \qquad
C = \begin{bmatrix}
1 & 0 & 0 & 0 & 0 \\
0 & \frac{1}{V_{\mathrm{p,h}}\eta_{\mathrm{v,h}}} & 0 & 0 & 0 \\
0 & 0 & 0 & 1 & 0 \\
0 & 0 & 0 & 0 & 1
\end{bmatrix}, \tag{54}
$$

with the state, input and output matrices

$$x = \begin{bmatrix} \hat{\mathcal{V}} & \hat{Q}_{\mathrm{in}} & \hat{Q}_{\mathrm{out}} & \hat{s} & \hat{\beta} \end{bmatrix}^T,$$

$$u = \begin{bmatrix} \hat{s}_{\mathrm{ref}} & \hat{\beta}_{\mathrm{ref}} \end{bmatrix}^T, \tag{55}$$

$$y = \begin{bmatrix} \hat{\mathcal{V}} & \hat{\omega}_{\mathrm{r}} & \hat{s} & \hat{\beta} \end{bmatrix}^T.$$

The dynamic model derived in this section is used in Sect. 4.2 to come up with an active spear valve torque control strategy in
the near-rated region.

## 4 Controller design

In this section designs are presented for control in the below- and near-rated operating region. A schematic diagram of the
overall control system is given in Figure 12. It is seen that the turbine is controlled by two distinct Proportional-Integral
(PI) controllers, a spear valve torque and blade pitch controller, acting on individual rotor speed set point errors $e_{\mathrm{s}}$ and $e_\beta$,
respectively. As both controllers have a common control objective of regulating the rotor speed and are implemented in a
decentralized way, it is ensured that they are not active simultaneously. The gain-scheduled pitch controller is designed and
implemented in a similar way as in conventional wind turbines (Jonkman et al., 2009), and is therefore not further elaborated
in this paper. The spear valve torque controller, however, is non-conventional and its control design is outlined in this section.



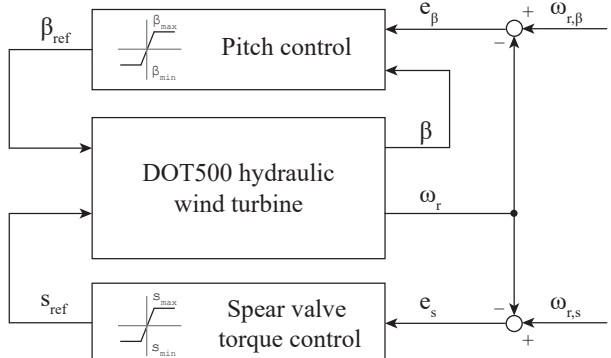

**Figure 12.** Schematic diagram of the DOT500 control strategy. When the control error $e$ is negative, the controllers saturate at their minimum or maximum setting. In the near-rated operating region, the rotor speed is actively regulated to $\omega_{r,s}$ by generating the spear position control signal $s_{ref}$, influencing the fluid pressure and the system torque. When the spear valve is at its rated minimum position, the gain-scheduled pitch controller generates a pitch angle set point $\beta_{ref}$ to regulate the rotor speed at its nominal value $\omega_{r,\beta}$.

For the below-rated control design a passive torque control strategy is employed, of which a description of is given in Sect. 4.1. Subsequently, in Sect. 4.2, the in-field active spear valve control implementation is evaluated using the dynamic drivetrain model.

## 4.1 Passive below-rated torque control

The passive control strategy for below-rated operation is described in this section. Conventionally, in below-rated operating conditions, the power coefficient is maximized by regulating the tip-speed ratio at $\lambda_0$ using generator torque control. Generally, the maximum power coefficient tracking objective is attained by implementing the feed-forward torque control law

$$\tau_{\mathrm{sys}} = \frac{\rho_{\mathrm{air}} \pi R^5 C_{\mathrm{p,max}}}{2\lambda^3} \omega_{\mathrm{r}}^2 = K_{\mathrm{r}} \omega_{\mathrm{r}}^2, \tag{56}$$

where $K_{\mathrm{r}}$ is the optimal mode gain in $\mathrm{Nm\,(rad/s)^{-2}}$.

As the DOT500 drivetrain lacks the option to directly influence the system torque, hydraulic torque control is employed using spear valves. An expression for the system torque for the hydraulic drivetrain is derived by substitution of Eq.s (13) and (15) in Eq. (16)

$$\tau_{\mathrm{sys}} = \frac{V_{\mathrm{p,h}} V_{\mathrm{p,k}}}{V_{\mathrm{p,b}}} \frac{1}{\eta_{\mathrm{m,h}}(\omega_{\mathrm{r}}, \Delta p_{\mathrm{h}}) \eta_{\mathrm{m,b}}(\omega_{\mathrm{b}}, \tau_{\mathrm{b}}) \eta_{\mathrm{m,k}}(\omega_{\mathrm{b}}, \Delta p_{\mathrm{k}})} \Delta p_{\mathrm{k}}, \tag{57}$$

and by substituting Eq.s (12) and (14) an expression as function of the spear position is obtained

$$\tau_{\mathrm{sys}} = \frac{V_{\mathrm{p,h}} V_{\mathrm{p,k}}}{V_{\mathrm{p,b}}} \frac{1}{\eta_{\mathrm{m,h}}(\omega_{\mathrm{r}}, \Delta p_{\mathrm{h}}) \eta_{\mathrm{m,b}}(\omega_{\mathrm{b}}, \tau_{\mathrm{b}}) \eta_{\mathrm{m,k}}(\omega_{\mathrm{b}}, \Delta p_{\mathrm{k}})} \left( \frac{\rho_{\mathrm{w}}}{2} \left( \frac{Q_{\mathrm{w}}}{C_{\mathrm{d}} A_{\mathrm{nz}}(s)} \right)^2 - p_{\mathrm{k,f}} \right). \tag{58}$$



Now substituting Eq.s (6) and (7) in Eq. (8) results in an expression relating the water flow to the rotor speed

$$Q_{\mathrm{w}} = \frac{V_{\mathrm{p,h}} V_{\mathrm{p,k}}}{V_{\mathrm{p,b}}} \frac{\eta_{\mathrm{v,k}} \eta_{\mathrm{v,b}} \eta_{\mathrm{v,h}}}{1} \omega_{\mathrm{r}}. \tag{59}$$

Combining Eq.s (58) and (59), and by disregarding the water pump feed pressure $p_{\mathrm{k,f}}$ gives

$$\tau_{\mathrm{sys}} = \frac{\rho_{\mathrm{w}}}{2 C_{\mathrm{d}}^2 A_{\mathrm{nz}}^2(s)} \left( \frac{V_{\mathrm{p,h}} V_{\mathrm{p,k}}}{V_{\mathrm{p,b}}} \right)^3 \frac{(\eta_{\mathrm{v,h}} \eta_{\mathrm{v,b}} \eta_{\mathrm{v,k}})^2}{\eta_{\mathrm{m,h}}(\omega_{\mathrm{r}}, \Delta p_{\mathrm{h}}) \eta_{\mathrm{m,b}}(\omega_{\mathrm{b}}, \tau_{\mathrm{b}}) \eta_{\mathrm{m,k}}(\omega_{\mathrm{b}}, \Delta p_{\mathrm{k}})} \omega_{\mathrm{r}}^2 = K_{\mathrm{s}} \omega_{\mathrm{r}}^2. \tag{60}$$

Rotor speed variations cause a varying flow through the spear valve, which results in a varying system pressure and thus system torque, regulating the tip-speed ratio of the rotor. The above obtained result shows that when $K_{\mathrm{s}}$ is constant, the tip-speed ratio can be regulated in the below-rated region by a fixed nozzle area $A_{\mathrm{nz}}$. Under ideal circumstances, it is shown in (Diepeveen and Jarquin-Laguna, 2014) that the nozzle area can be chosen constant to let the rotor follow the optimal power coefficient trajectory, and is called passive torque control. This means that no active control is needed up to the near-rated operating region. For this purpose, the optimal mode gain $K_{\mathrm{s}}$ of the system side needs to equal that of the rotor $K_{\mathrm{r}}$ in the below-rated region.

However, for the passive strategy to work out, the combined drivetrain efficiency needs to be consistent in the below-rated operating region. As seen in Eq. (60), the combined efficiency term is a product of the consequent volumetric divided by the mechanical efficiencies of all components, as a function of their current operating point. To assess the consistency of the overall drivetrain efficiency, different operating strategies are examined in Sect. 4.1.1. Subsequently, a component efficiency analysis is given in Sect. 4.1.2.

### 4.1.1 Operational strategies

Because hydraulic components are known to be more efficient in high-load operating conditions (Trostmann, 1995), it might be advantageous for a hydraulic drivetrain to operate the rotor at a lower tip-speed ratio. Operating at a lower tip-speed ratio results in a lower rotational rotor speed and a higher system torque. The resulting higher operational pressures might result in maximization of the total drivetrain efficiency. A consequence of operating the turbine at a lower tip-speed ratio is the decreased rotor power coefficient $C_{\mathrm{p}}$. For these reasons, an analysis of this trade-off is divided into two cases:

- **Case 1:** operating the rotor at its maximum power coefficient $C_{\mathrm{p,max}}$;
- **Case 2:** operation at the maximum torque coefficient $C_{\tau,\mathrm{max}}$.

Referring back to the rotor power/torque curve in Figure 7, and substituting the values for operation at $C_{\mathrm{p,max}}$ and $C_{\tau,\mathrm{max}}$ in Eq. (56), optimal mode gain values of $K_{\mathrm{r,p}} = 1.00 \cdot 10^4 \, \mathrm{Nm \, (rad/s)^{-2}}$ and $K_{\mathrm{r,\tau}} = 2.05 \cdot 10^4 \, \mathrm{Nm \, (rad/s)^{-2}}$ are found for cases 1 and 2, respectively. The result of evaluating the rotor torque path in the below-rated region for the two cases is presented in Figure 13. Due to the lower tip-speed ratio in case 2 , the rotor speed is lower for equal wind speeds; or a higher wind speed is required for operation at the same rotor speed resulting in a higher torque.



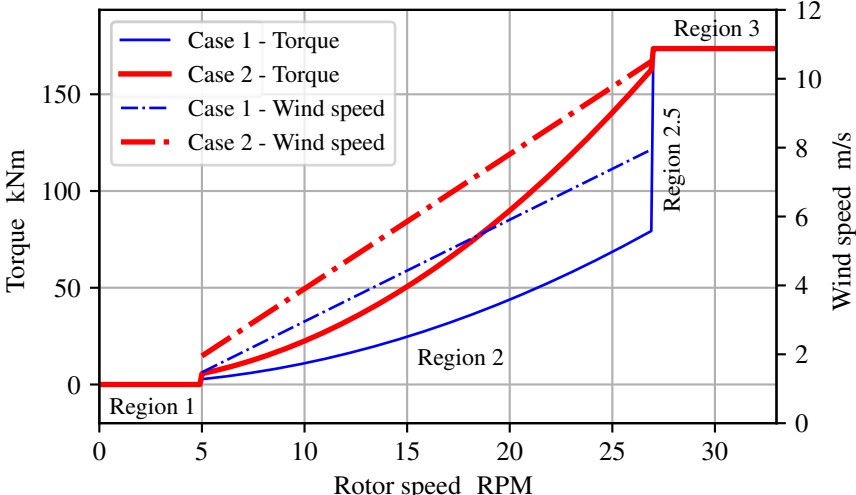

**Figure 13.** Torque control strategies for maintaining a fixed tip-speed ratio $\lambda$, tracking the optimal power coefficient $C_{p,max}$ (case 1) and the maximum torque coefficient $C_{\tau,max}$ (case 2). The dash-dotted lines show the corresponding wind speed to the distinct strategies (right y-axis), and the vertical dashed lines indicate boundaries between operating regions.

### 4.1.2 Drivetrain efficiency and stability analysis

This section presents the available component efficiency data, and evaluates steady-state drivetrain operation characteristics for the two previously introduced operating cases. The component efficiency characteristics primarily influence the steady-state response of the wind turbine, as shown in Eq. (60). To perform a fair comparison between both operating cases, the rotor

5 efficiency is normalized with respect to case 1, resulting in a constant efficiency factor of $0.85$ for operating case 2. Detailed efficiency data is available for the oil pump and motor, however, as no data for the efficiency characteristics of the water pump is available, a constant mechanical and volumetric efficiency of $\eta_{m,k} = 0.83$ and $\eta_{m,k} = 0.93$ are assumed, respectively. The oil pump is supplied with total efficiency data $\eta_{t,h}$ as a function of the (rotor) speed $\omega_r$ and the supplied torque $\tau_{sys}$ (Hägglunds, 2015). An expression relating the mechanical, volumetric and total efficiency is given by

$$10 \quad \eta_{m,h}(\omega_r, \tau_{sys}) = \frac{\eta_{t,h}(\omega_r, \tau_{sys})}{\eta_{v,h}}, \qquad (61)$$

where $\eta_{v,h}$ is taken as $0.98$, and the result is presented in Figure 14 (left). The plotted data points (dots) are interpolated on a mesh grid using a regular grid linear interpolation method from the Python SciPy interpolation toolbox (Scipy.org, 2017). Operating cases 1 and 2 are indicated by the solid lines. The same procedure is performed for the data supplied with the oil motor, of which the result is presented in Figure 14 (right), where $\eta_{v,b}$ is taken as $0.98$. As concluded from the efficiency curves,

15 hydraulic components are generally more efficient in the low-speed high-torque/pressure region. It is immediately clear that





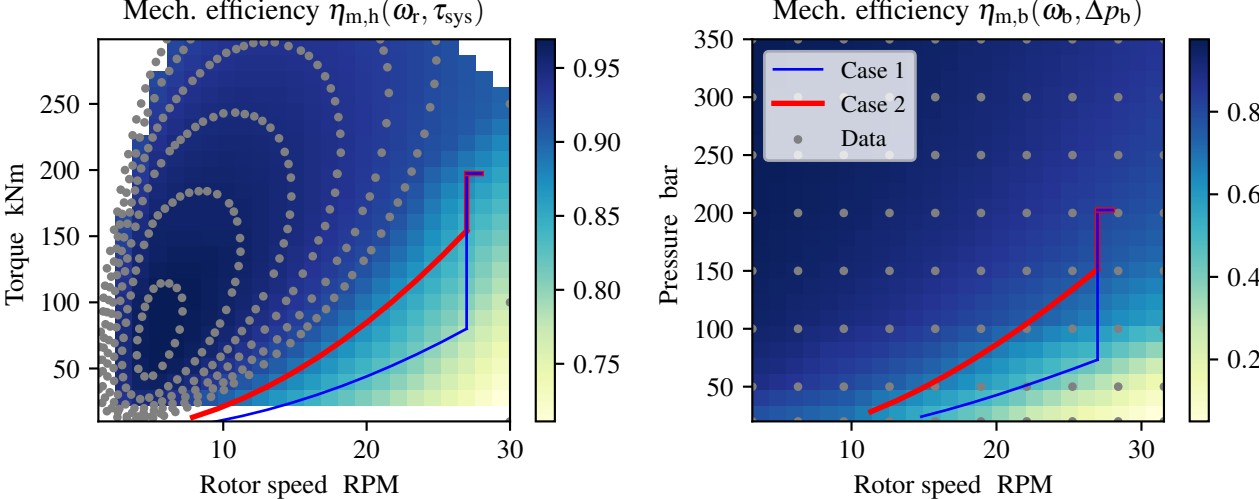

**Figure 14.** Mechanical efficiency mapping of the oil pump and oil motor. Manufacturer supplied data (blue dots) is evaluated using an interpolation function. Operating cases 1 and 2 are indicated by the solid blue and red lines, respectively.

for both the oil pump as well as the motor, operating the rotor at a lower tip-speed ratio (case 2) is beneficial from a component efficiency perspective.

The drivetrain efficiency analysis for both operating cases is given in Figure 15. The lack of efficiency data at lower rotor speeds in the left plot of Figure 15 (case 1) is due to unavailability of data at lower pressures. From the resulting plot it is

concluded that the overall drivetrain efficiency for case 2 is higher and more consistent compared to case 1. The consistency of the total drivetrain efficiency is advantageous for control, as this will enable passive torque control to maintain a constant tip-speed ratio. As a result of this observation, the focus is henceforth shifted to the implementation of a torque control strategy tracking the maximum torque coefficient (case 2).

A stability concern for operation at the maximum torque coefficient needs to be highlighted. For stable operation, the value

of $k_\lambda$ needs to be negative. As shown in Figure 11, it can be seen that the stability boundary is located at a tip-speed ratio of 5.9. Operation at a lower tip-speed ratio results in unstable turbine operation and deceleration of the rotor speed to standstill. However, as concluded in (Schmitz et al., 2013), hydraulic drivetrains can compensate for this theoretical instability, allowing operation at lower tip-speed ratios. Therefore, the case 2 torque control strategy will be designed for the theoretical calculated minimum tip-speed ratio of 5.9, and in-field test results need to confirm the practical feasibility of the implementation.

**4.2   Active near-rated torque control**

A feedback hydraulic torque control is derived for near-rated operation in this section. To this end, active spear position control is employed to regulate the rotor speed. The effect on fluid resonances is analyzed, as these are possibly excited by an




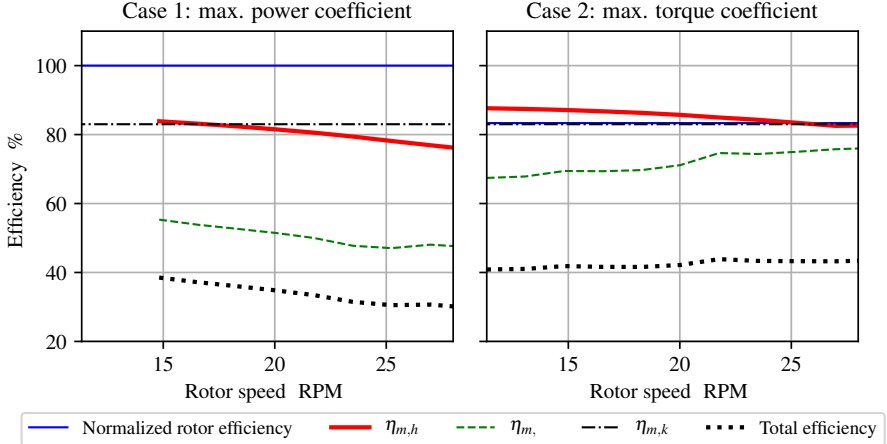

**Figure 15.** Comparison of the total drivetrain efficiency for operating cases 1 and 2. It is observed that the total efficiency is higher in the complete below-rated region for case 2. Also the efficiency over all rotor speeds is more consistent, enabling passive torque control using a constant nozzle area $A_{\mathrm{nz}}$.

increased rotor speed control bandwidth. The in-field tests with corresponding control implementations are performed prior to the theoretical dynamic analysis of the drivetrain. For this reason, the control design and tunings used in-field are evaluated and possible improvements are highlighted. Sections 4.2.1 and 4.2.2 define the modeling parameters of the oil column and spear valve actuator, which is used in Sect. 4.2.3 for spear valve torque controller design.

**4.2.1 Defining the hydraulic model parameters**

The high-pressure oil line in the DOT500 drivetrain is considered to contain SAE30 oil, with a density of $\rho_{\mathrm{o}} = 900\,\mathrm{kg\,m^{-3}}$, and an effective bulk modulus of $K_{\mathrm{f,o}} = 1.5 \cdot 10^9\,\mathrm{Pa}$. The hydraulic line is cylindrical with a length of $L_{\mathrm{l}} = 50\,\mathrm{m}$ and a radius of $r_{\mathrm{l}} = 50 \cdot 10^{-3}\,\mathrm{m}$. The dynamic viscosity of SAE30 oil is taken at a fixed temperature of $20\,^{\circ}\mathrm{C}$, where it reads a value of $\mu = 240 \cdot 10^{-3}\,\mathrm{Pa\,s}$. With this data the hydraulic inductance, resistance and capacitance have calculated values of $L_{\mathrm{H}} = 5.730 \cdot$
$10^6\,\mathrm{kg\,m^{-4}}$, $R_{\mathrm{H}} = 4.889 \cdot 10^6\,\mathrm{kg\,m^{-4}\,s^{-1}}$ and $1/C_{\mathrm{H}} = 3.820 \cdot 10^9\,\mathrm{kg\,m^{-4}\,s^{-2}}$, respectively. Using Eq. (20), the flow is calculated to be laminar as $Re = 1177$ with an oil flow at nominal rotor speed of $14781\,\mathrm{min^{-1}}$, and thus a correction factor $f_{\mathrm{c}} = 4/3$ is applied to the hydraulic inductance.

By substitution of the calculated values in Eq. (30), a visualization of the transfer function frequency response is given in Figure 16 at a range of hydraulic line lengths. In this Bode plot, it is shown that the line length has great influence on
the location of the natural frequency and damping ratio. A longer line shifts the frequency to lower values and increases the damping ratio.




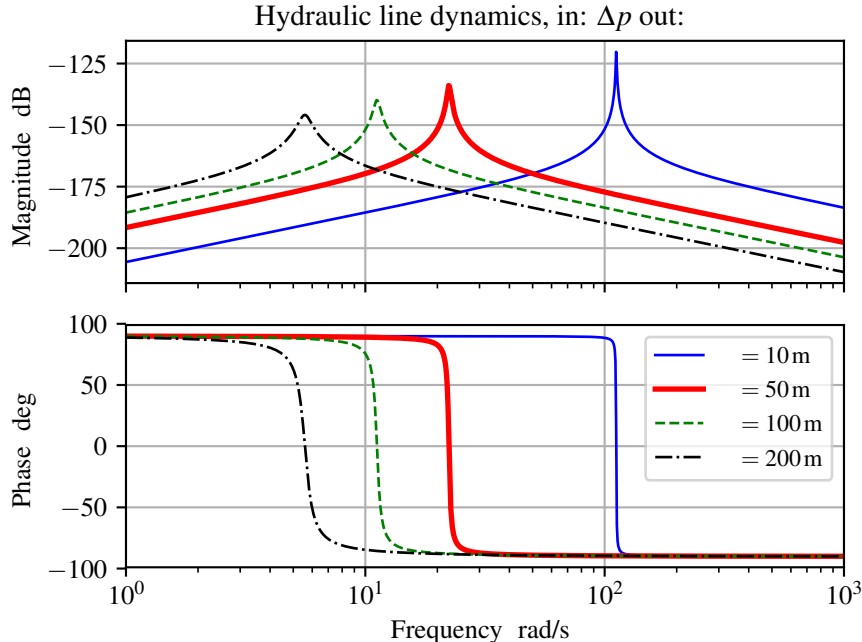

**Figure 16.** Bode plot of a hydraulic control volume modeled as an harmonic oscillator with pressure change $\Delta p$ as input and flow $Q$ as output. The length of the hydraulic line has great influence on the location of the natural frequency and magnitude of the damping ratio.

### 4.2.2 Modeling spear valve characteristics

For determining the spear valve time constant $t_s$ defined in Eq. (36), a Generalized pseudo-random Binary Noise (GBN) identification signal (Godfrey, 1993) is supplied to one of the spear valve actuators. From this test it is seen that the actuator has a fixed and rate-limited positioning speed of $\dot{s}_{\max} = 0.44 \, \text{mm} \, \text{s}^{-1}$, and shows no observable transient response. The measured spear valve position is always higher or lower than the requested set point, which is an inevitable consequence of the deadband control implementation and the absence of proportional spear actuation capabilities.

Because of the non-linear rate-limited response, an actuator model will be parameterized for the worst-case scenario. This is done by evaluating the response at maximum actuation amplitude and determining the corresponding bandwidth, such that closed-loop reference position tracking is ensured. As concluded from in-field experiments, the spear position control range in the near-rated operating region is $s_{\max} = 1.5 \, \text{mm}$. Considering excitation with a sinusoidal signal $s = s_{\max} \sin(\omega_{s,\max} t)$, the corresponding maximum bandwidth $\omega_{s,\max}$, and thus time-constant $t_s$ for reference tracking is given by

$$\dot{s}_{\max} = \frac{1}{2} s_{\max} \omega_{s,\max} \cos(\omega_{s,\max} t), \quad t_s = \frac{1}{\omega_{s,\max}} = 1.69 \, \text{s}. \tag{62}$$

The spear position relates in a non-linear fashion to the applied system torque, as a consequence of the spear valve geometry presented in Figure 9 and 10. Therefore, an evaluation of the spear valve pressure gradient with respect to the spear position



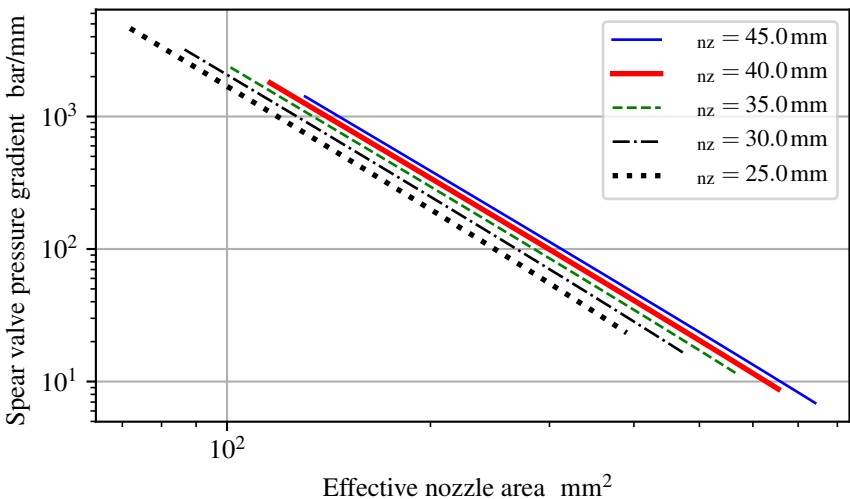

**Figure 17.** Spear valve position pressure gradient, evaluated for different nozzle head diameters at a range of effective nozzle areas. The pressure sensitivity is higher for larger nozzle diameters at equal effective areas. Results are presented in a double-logarithmic plot.

$k_{\mathrm{s,s}}$ is given in Figure 17. This is done for distinct nozzle head diameters at a range of effective nozzle areas. It is shown that the spear pressure gradient with respect to the position is higher for larger nozzle head diameters at equal effective areas.

### 4.2.3 Torque control design and evaluation

The active spear valve torque control strategy employed during the in-field tests is now evaluated. A fixed-gain controller was
implemented for rotor speed control in the near-rated region. As the goal is to make a fair comparison and evaluation of the in-field control design, the same PI controller is used in this analysis.

The dynamic drivetrain model presented in Sect. 3.2 is further analyzed. The linear state-space system in Eq. (54) is evaluated at different operating points in the near-rated region. The operating point is chosen at a rotor speed of $\omega_{\mathrm{r,s}} = 27\,\mathrm{RPM}$, which results in a water pump discharge flow of $\bar{Q}_{\mathrm{w}} = 29651\,\mathrm{min}^{-1}$, taking into account the volumetric losses. For the entire near-
rated region, a range of wind speeds and corresponding model parameters are computed and are listed in Table 2. An analysis is performed on the Single-Input Single-Output (SISO) open-loop transfer system with the speed error $e_{\mathrm{s}}$ as input and rotor speed as output, including the spear valve PI torque controller used during field tests. The gains of the PI controller were $K_{\mathrm{p}} = 3.8 \cdot 10^{-3}\,\mathrm{m\,(rad/s)^{-1}}$, and $K_{\mathrm{i}} = 6.6845 \cdot 10^{-4}\,\mathrm{m\,rad^{-1}}$.

By inspection of the state $A$-matrix, various preliminary remarks can be made regarding system stability and drivetrain
damping. First it is seen that for the $(2,\,2)$-element, the hydraulic resistance $R_{\mathrm{H}}$ influences the intrinsic speed feedback gain $k_{Q_{\mathrm{in}}}^{*}$. It was concluded in Sect. 4.1.2 that the rotor operation is stable for negative values of $k_{\omega_{\mathrm{r}}}$ and thus $k_{Q_{\mathrm{in}}}^{*}$. Thus, the higher the hydraulic resistance, the longer the $(2,\,2)$-element stays negative for decreasing tip-speed ratios, resulting in increased




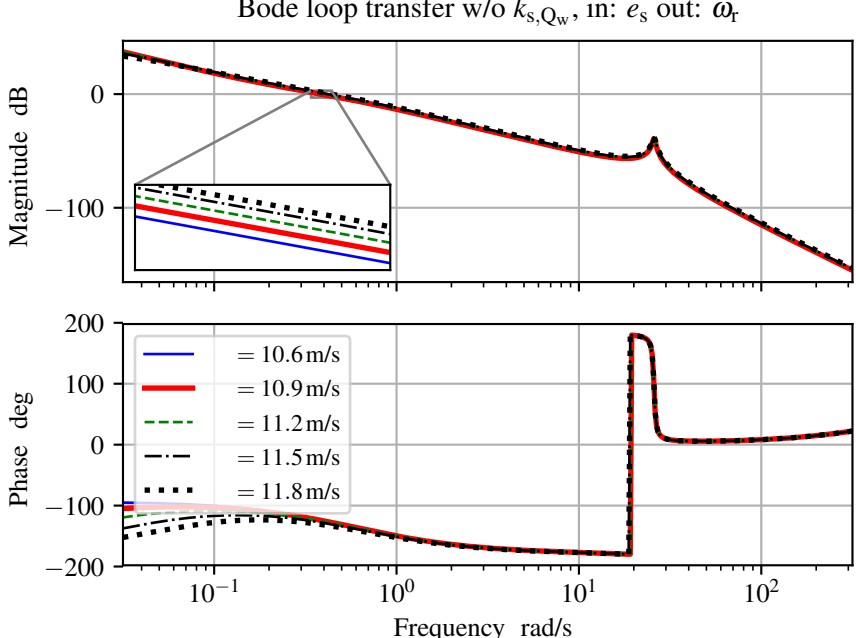

**Figure 18.** Bode plot of open-loop transfer including controller from spear valve position reference to the rotor speed. **without** spear valve pressure feedback gain $k_{s,Q_w}$.

operational stability. This effect has been observed earlier in (Schmitz et al., 2012, 2013), where it was shown that turbines with a hydraulic drivetrain are able to attain lower tip-speed ratios. This is in accordance with Eq.s (23) and (24), where it is shown that the resistance term only influences the damping. Furthermore, the spear valve pressure feedback gain $k_{s,Q_w}$ provides additional system torque when the rotor has a speed increase or overshoot, resulting in additional damping to the

5   (3, 3)-element in the state matrix.

    During analysis, an important result is noticed and is shown by discarding and including the spear valve pressure feedback gain $k_{s,Q_w}$ in the linearized state-space system. The open-loop Bode plot of the system excluding the term is presented in Figure 18, and including the term in Figure 19. It is noted that the damping term completely damps the hydraulic resonance peak in the oil column as a result of the intrinsic flow-pressure feedback effect. At the same time, the attainable bandwidth

10  of the hydraulic torque control implementation is limited. This bandwidth limiting effect becomes more severe by applying longer line lengths and thus larger volumes in the discharge line, as it increases the hydraulic induction. Fortunately, various solutions are possible to mitigate this effect and are discussed next.

    The effect of increasing different model parameters is depicted in Figure 19. In the DOT500 set-up an intermediate oil circuit is used, which will be omitted in the ideal DOT concept. Seawater has a higher effective bulk modulus compared to oil and





**Table 2.** Parameters for linearization of the model in the near-rated operating region.

| Description | Symbol | Value | Unit |
|---|---|---|---|
| Wind speed | $\bar{U}$ | $10.6 - 11.8$ | $\mathrm{m\,s^{-1}}$ |
| Rotor speed set point | $\bar{\omega}_{\mathrm{r}}$ | $27.0$ | RPM |
| Water flow | $\bar{Q}_{\mathrm{w}}$ | $2965$ | $\mathrm{l\,min^{-1}}$ |
| Water pressure | $\bar{p}_{\mathrm{w,l}}$ | $51.4 - 62.4$ | bar |
| Oil flow | $\bar{Q}_{\mathrm{o}}$ | $1354$ | $\mathrm{l\,min^{-1}}$ |
| Oil pressure | $\Delta\bar{p}_{\mathrm{h}}$ | $166 - 201$ | bar |
| Rotor torque | $\bar{\tau}_{\mathrm{r}}$ | $163.8 - 198.7$ | kNm |
| Rotor inertia | $J_{\mathrm{r}}$ | $6.6 \cdot 10^{5}$ | $\mathrm{kg\,m^2}$ |
| Nozzle diameter | $D_{\mathrm{nz}}$ | $38$ | mm |
| Spear coning angle | $\alpha$ | $50$ | deg |
| Number of spear valves | $N_{\mathrm{s}}$ | $2$ | - |
| Discharge coefficient | $C_{\mathrm{d}}$ | $1.0$ | - |
| Effective nozzle area | $A_{\mathrm{nz}}$ | $486.9 - 442$ | $\mathrm{mm^2}$ |
| Spear position | $\bar{s}$ | $4.64 - 4.18$ | mm |
| Density of air | $\rho_{\mathrm{a}}$ | $1.225$ | $\mathrm{kg\,m^{-3}}$ |
| Density of oil | $\rho_{\mathrm{o}}$ | $900$ | $\mathrm{kg\,m^{-3}}$ |
| Density of water | $\rho_{\mathrm{w}}$ | $998$ | $\mathrm{kg\,m^{-3}}$ |
| Component mechanical efficiency | $\eta_{\mathrm{m,x}}$ | $0.85$ | - |
| Component volumetric efficiency | $\eta_{\mathrm{v,x}}$ | $0.95$ | - |
| Hydraulic line radius | $r_{\mathrm{l}}$ | $50.0$ | mm |
| Hydraulic line length | $L_{\mathrm{l}}$ | $50.0$ | m |
| Hydraulic line volume | $V_{\mathrm{H}}$ | $392.6$ | l |
| Hydraulic induction, oil | $L_{\mathrm{H}}$ | $5.730 \cdot 10^{6}$ | $\mathrm{kg\,m^{-4}}$ |
| Hydraulic resistance, oil | $R_{\mathrm{H}}$ | $4.889 \cdot 10^{6}$ | $\mathrm{kg\,m^{-4}s^{-1}}$ |
| Effective bulk modulus, oil | $K_{\mathrm{f}}$ | $1.5$ | GPa |
| Dynamic viscosity, oil | $\mu_{\mathrm{o}}$ | $0.240$ | Pa s |
| Spear valve actuator time constant | $t_{\mathrm{s}}$ | $1.69$ | s |
| Pitch actuator time constant | $t_{\mathrm{s}}$ | $0.5$ | s |

this has a positive effect on the maximum attainable torque control bandwidth for equal line lengths. Moreover, a faster spear valve actuator has a positive influence on the attainable control bandwidth.

From both figures it is also observed that the system dynamics change according to the operating point. At higher wind speeds, the magnitude response results is higher. The effect was earlier seen in Figure 17, where the pressure gradient with





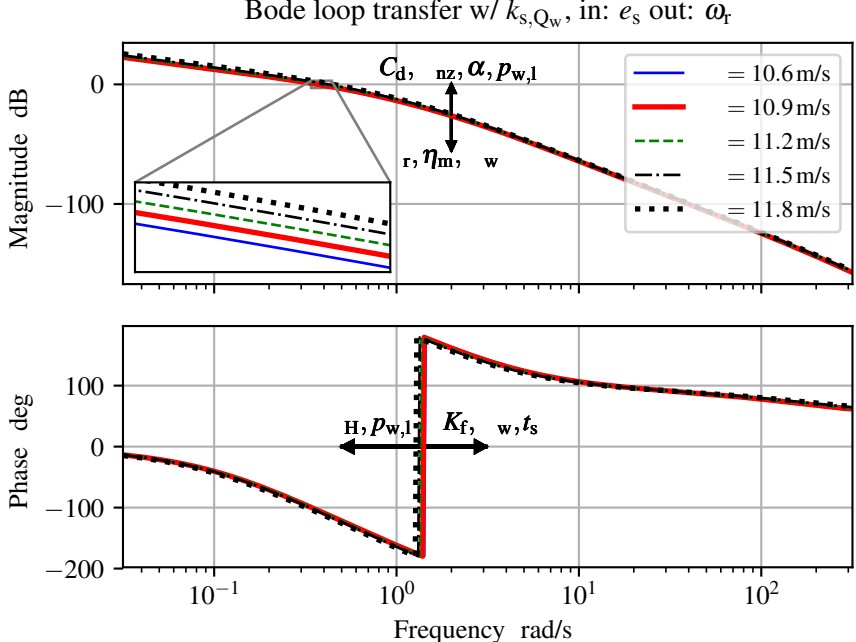

**Figure 19.** Bode plot of the plant for transfer from spear valve position reference to the rotor speed, **with** spear valve pressure feedback gain $k_{s,Q_w}$. The effect of increasing important model parameters on the frequency response is indicated.

respect to the spear valve position is higher for smaller effective nozzle areas. It is concluded that the sizing of the nozzle diameter is a trade-off between the minimal achievable pressure and its controllability with respect to the resolution of the spear positioning and actuation speed.

5      Closed-loop step responses throughout the near-rated region (for different wind speeds) are shown in Figure 20. It is seen that the controller stabilizes the system, and the control bandwidth increases for higher wind speeds. The loop-shaped frequency responses attain a minimum and maximum bandwidth of $0.35$ and $0.43\,\mathrm{rad\,s^{-1}}$ with phase margins of $55$ and $40\,\mathrm{deg}$, respectively. For later control designs a more consistent control bandwidth can be attained by gain-scheduling the controller gains on a measurement of the water pressure or spear position.

## 5   Implementation of control strategy and in-field results

10  This section covers the implementation and evaluation of the derived control strategies on the real-world in-field DOT500 turbine. In accordance with the previous section, a distinction is made between passive and active regulation, for below- and near-rated wind turbine control, respectively. To illustrate the overall control strategy from in-field gathered test data, operational visualizations are given in Sect. 5.1. Evaluation of the effectiveness of the passive below-rated and active near-rated torque control strategy is presented in Sect. 5.2.





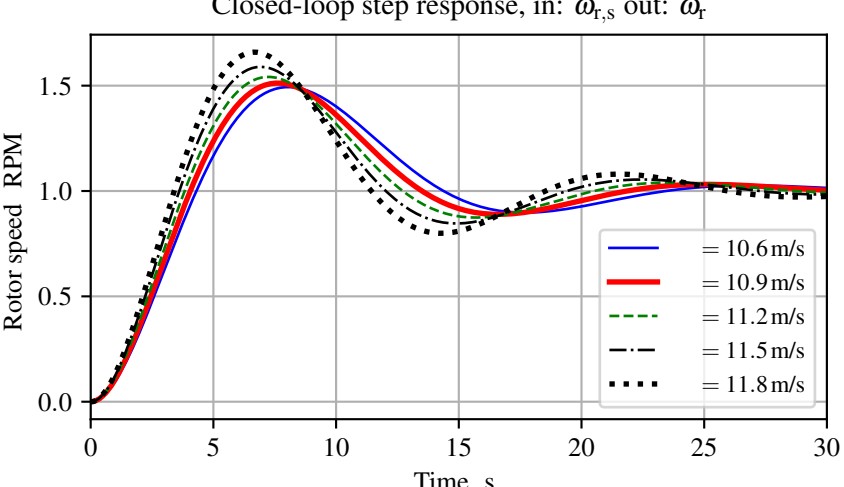

**Figure 20.** Closed-loop step responses of the spear valve torque control implementation. It is shown that the feedback loop is stabilized, but that the dynamics vary with increasing wind speeds. A control implementation that takes care of the varying spear valve position pressure gradient would load to a more consistent response.

## 5.1 Turbine performance characteristics and control strategy

To illustrate the overall control strategy, three-dimensional rotor torque and rotor speed plots are shown as a function of spear position and wind speed in Figure 21. By fixing the valve position at a range of positions, making sure that sufficient data is collected throughout all wind speeds and binning the data accordingly, a steady-state drivetrain performance mapping is

derived. Both figures show the data binned in predefined spear valve positions and wind speeds. This is done on a normalized scale, where $0\%$ is the maximum spear position (larger nozzle area), and $100\%$ the minimum spear position (smaller nozzle area). The absolute difference between the minimum and maximum spear position is $1.5\,\text{mm}$. The spear position is the only control input in the below-rated region, and is independent from other system variables. During data collection, the pitch system regulates the rotor speed up to its nominal set point value when an overspeed occurs.

In both figures the control strategy is indicated by colored lines. For below-rated operation (red), the spear valve position is kept constant: flow fluctuations influence the water discharge pressure and thus the system torque. In near-rated conditions (blue), the spear position is actively controlled by a PI-controller. Under conditions of a constant regulated water flow corresponding to $\omega_{\text{r,s}} = 27\,\text{RPM}$, the controller continuously adjusts the spear position and thus water discharge pressure to regulate the rotor speed. Once the turbine reaches its nominal power output, the rotor limits wind energy power capture using

gain-scheduled PI pitch control (green), maintaining the rotor speed at $\omega_{\text{r},\beta} = 28\,\text{RPM}$.





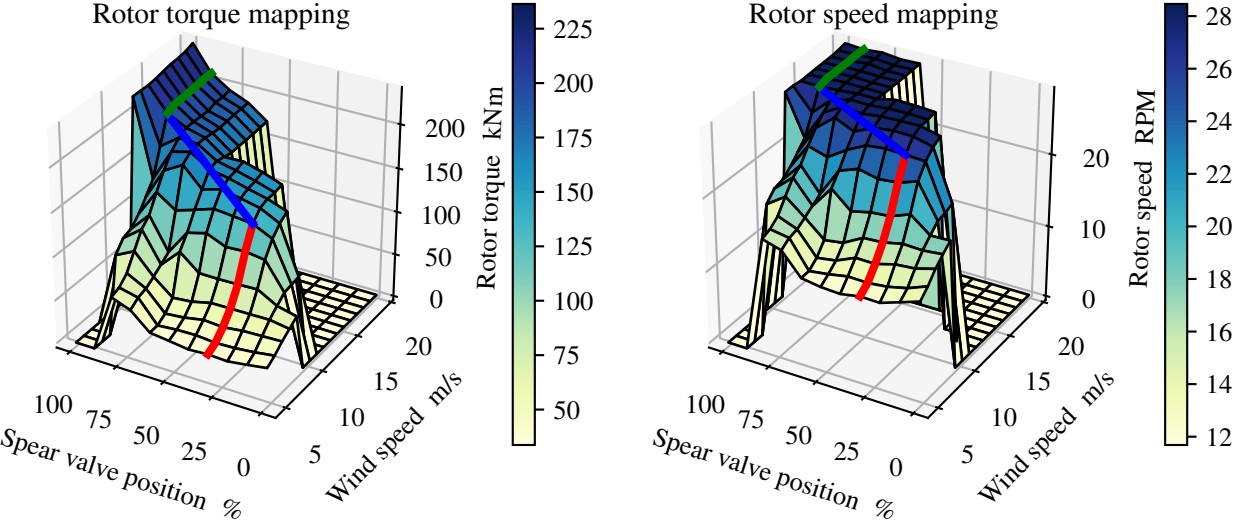

**Figure 21.** Steady-state rotor torque and speed at predefined spear valve positions (nozzle areas) and wind speed conditions. The red line indicates the operation strategy at fixed spear valve position in the below-rated region, whereas the blue trajectory indicates active spear valve position control towards rated conditions. The effect of blade pitching is indicated in green.

## 5.2 Evaluation of the control strategy

Previously, in Sect. 4.1, drivetrain characteristics are deduced from prior component information throughout the wind turbine operating region. Characteristic data from the rotor, the oil pump and the oil motor is evaluated to come up with a hydraulic torque control strategy. Due to the predicted consistent mechanical efficiency of the hydraulic drivetrain, the nozzle area can be

fixed and no active torque control is needed in the below-rated region. This results from the rotor speed being proportional to water flow, and relates to system torque according to Eq. (60). From the analysis it is also concluded that operating at a lower tip-speed ratio, results in a higher and more consistent overall efficiency for this particular drivetrain.

As concluded in Sect. 4.1.2, stable turbine operation is attained when the rotor operates at a tip-speed ratio such that $k_\lambda$ is negative, and Figure 11 shows that the predicted stability boundary is located at a tip-speed ratio of $\lambda = 5.9$. Using the

obtained data from in-field tests, a mapping of the attained tip-speed ratios as function of the spear valve position and rotor speed is given in Figure 22. An anemometer on the nacelle and behind the rotor measures the wind speed. As turbine wind speed measurements are generally considered less reliable (Østergaard et al., 2007) and the effect of induction is not included in this analysis, the obtained results serve as an indication of the turbine behavior. The dashed line indicates the fixed spear position of 70 %, and is chosen as the position for passive torque control in the below-rated region. The attained tip-speed ratio

averages are presented in the left plot, and the right plot shows a two-dimensional visualization of the data indicated by the dashed line, including one standard deviation. Closing the spear valve further for operation at an even lower tip-speed ratio and higher water pressures, resulted in a slowly decreasing rotor speed and thus unstable operation.




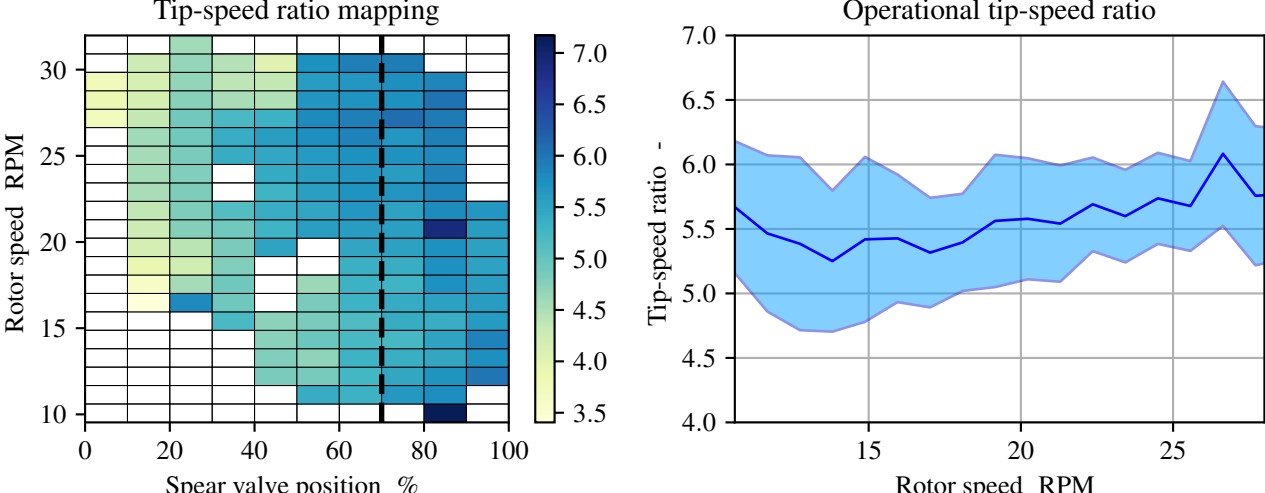

**Figure 22.** Tip-speed ratio mapping as function of spear valve position and rotor speed (left). The dashed line indicates the fixed spear position selected for below-rated passive torque control. The right plot shows the corresponding evaluation of the tip-speed ratios along this path with $1\sigma$ standard deviation bands.

In Figure 22, it is shown that the calculated tip-speed ratio is regulated around a mean of $5.5$ for below-rated conditions. Although the attained value is lower than the theoretical calculated minimum tip-speed ratio of $5.9$, stable turbine operation is attained during in-field tests. A plausible explanation is that the damping characteristics of hydraulic components compensate for instability as predicted in Sect. 4.2.3.

Figure 23 shows reaction torque measurements by the load-pins in the suspension of the oil pump to estimate the attained rotor torque during below-rated operation. From the tip-speed ratio heat map and the rotor torque measurements it is concluded that the case 2 (maximum rotor torque coefficient) strategy works out on the actual turbine, and the passive strategy regulates the torque close to the desired predefined path. However, as can be seen in both figures for lower rotor speeds, the tip-speed ratio attains lower values and the rotor torque increases. An explanation for this effect is the decreased mechanical water pump

efficiency, of which, as earlier stated, the efficiency characteristics are unknown. An analysis of the water pump efficiency is performed using measurement data and show non-constant mechanical efficiency characteristics: the efficiency drops rapidly when the rotor speeds is below $15\,\mathrm{RPM}$.

Finally, the active spear valve torque control strategy is evaluated. The aim is to regulate the rotor speed to a constant reference speed in the near-rated operating region. In-field test results are given in Figure 24, and all values are normalized

for convenient presentation. It is shown that active spear valve control combined with pitch control regulates the wind turbine for (near-)rated conditions in a decentralized way. The spear position tracks the control signal reference, and shows that the strategy has sufficient bandwidth to act as a substitute to conventional torque control.




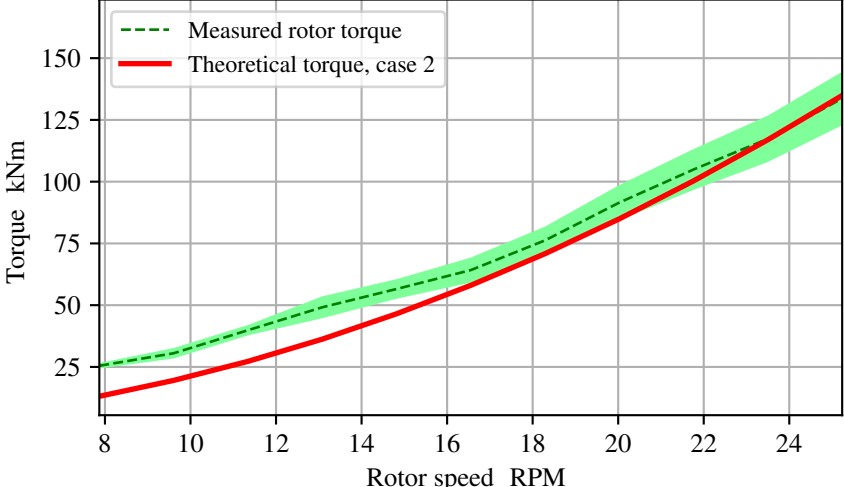

**Figure 23.** Evaluation of the passive torque control strategy, by comparison of the theoretical torque for case 2 to the torque measured by load-pins in the oil pump suspension. For higher speeds, the passive torque control strategy succeeds in near-ideal tracking of the desired case 2 path. At lower rotor speeds, the torque is higher than the aimed theoretical line, resulting from the lower combined drivetrain efficiency in this operating region.

## 6  Conclusions

This paper presents the control design for the intermediate DOT500 hydraulic wind turbine. This turbine with a 500 kW hydraulic drivetrain is deployed in-field and served as proof of concept. The drivetrain included a hydraulic transmission in the form of an oil circuit, as at the time of writing a low-speed high-torque seawater pump was not commercially available, and is being developed by DOT.

First it is concluded that for the employed drivetrain, operating at maximum rotor torque, instead of maximum rotor power, is beneficial for drivetrain efficiency maximization. This results not only in an increased overall efficiency, but comes with an additional advantage of the mechanical efficiency characteristics being consistent for the drivetrain. It is concluded that for a successful application of below-rated passive torque control, a consistent overall drivetrain efficiency is required. Another benefit of the hydraulic drivetrain is the added damping, enabling operation at lower tip-speed ratios. It is shown using in-field measurement data that the passive strategy succeeds in tracking the torque path corresponding to maximum rotor torque for a large envelope in the below-rated region. For a smaller portion, the combined drivetrain mechanical efficiency drops, which results in deviation from the desired trajectory.

Secondly, a drivetrain model including the oil dynamics is derived for spear valve control design in the near-rated region. It is shown that by including a spear valve as a control input, the hydraulic resonance is damped by the flow induced spear valve pressure feedback. This intrinsic pressure feedback effect also limits the attainable torque control bandwidth. However,



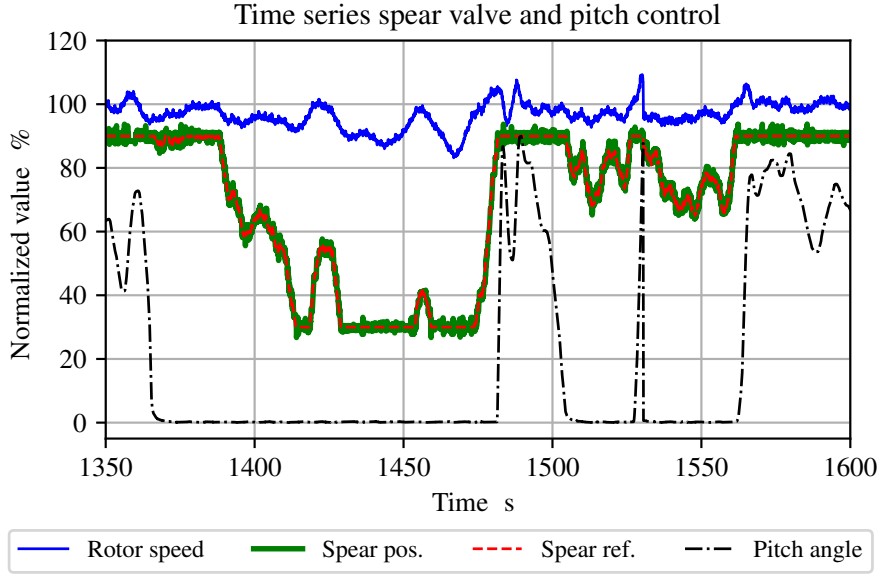

**Figure 24.** A time-series showing the hydraulic control strategy for the DOT500 turbine. The spear valve position actively regulates the rotor speed as a substitute to conventional turbine torque control. In the above-rated region, pitch control is employed to keep the rotor at it nominal speed. All signals in this plot are normalized.

this limiting effect can be coped with by using a stiffer fluid, a decreased hydraulic line volume or a faster spear valve actuator. The sizing of the nozzle head diameter influences the pressure sensitivity with respect to the spear position, and affects the attainable control bandwidth by spear valve actuation speed constraints and positioning accuracy. In-field test results show the practical feasibility of the strategy including spear valve and pitch control inputs to actively regulate the wind turbine in

5 the (near-)rated operating region. Future control designs will be improved by including a control implementation taking into account the varying spear valve pressure gradient. This will result in a higher and more consistent system response.

The ideal DOT concept discards the oil circuit and only uses water hydraulics with an internally developed seawater pump. As a result, the control design process is simplified and the overall drivetrain efficiency will be greatly improved. Future research will focus on the design of a centralized control implementation for DOT wind turbines acting in a hydraulic network.

10 **Appendix A: Definition of hydraulic induction, resistance and capacitance for Sect. 3.2.1**

**Hydraulic induction**: the hydraulic induction $L_\mathrm{H}$ resembles the ease of acceleration of a fluid volume and is related to the fluid inertia $I_\mathrm{f}$ by

$$L_\mathrm{H} = f_\mathrm{c} I_\mathrm{f} = f_\mathrm{c} \frac{\rho L_\mathrm{l}}{A}, \tag{A1}$$





with the assumption that the flow speed profile is radially uniform (Akers et al., 2006). For this reason, a distinction should be made between laminar and turbulent flows in circular lines: the induction of a laminar flow is generally corrected by a factor $f_c = 4/3$, whereas a turbulent flow does not need correction with respect to the fluid inertia $I_f$ (Bansal, 1989).

**Hydraulic resistance**: the hydraulic resistance dissipates energy from a flow in the form of a pressure decrease over a hydraulic
element. In most cases, hydraulic resistances are taken as an advantage by means of control valves. For example, by adjusting a valve set point, one adjusts the resistance to a desired value. Mathematically, the hydraulic resistance relates the flow rate to the corresponding pressure drop

$$\Delta p_{\mathrm{R}} = Q R_{\mathrm{H}}, \tag{A2}$$

analogous to an electrical circuit where the voltage over a resistive element equals the current times the resistance. The hy-
draulic resistance for a hydraulic line with a circular cross section and a laminar flow is

$$R_{\mathrm{H,l}} = \frac{8\mu L_{\mathrm{l}}}{\pi r_{\mathrm{l}}^4}, \tag{A3}$$

which is a constant term independent of the flow rate. For a turbulent fluid flow, the computation of the resistance is more involved and results in a quantity that is dependent on the flow rate and effective pipe roughness. For simulation purposes this would require re-evaluation of the resistance in each time step, or for each operating point during linear analysis. Such a
Non-Linear Time-Variant (NLTV) system is employed in (Buhagiar et al., 2016), updating the resistive terms in each iteration for a hydraulic variable-displacement drivetrain with seawater under turbulent conditions.

**Hydraulic capacitance**: due to fluid compressibility and line elasticity, the amount of fluid can change as a result of pressure changes in a control volume. The effective bulk modulus $K_f$ of a fluid is defined by the pressure increase to the relative decrease of the volume

$$dp = K_{\mathrm{f}} \frac{dV}{V_{\mathrm{H}}}, \qquad K_{\mathrm{f}} = V_{\mathrm{H}} \frac{dp}{dV}. \tag{A4}$$

Subsequently, the pressure change with respect to time is

$$\dot{p} = \frac{dp}{dt} = K_{\mathrm{f}} \frac{1}{V_{\mathrm{H}}} \frac{dV}{dt} = \frac{K_{\mathrm{f}}}{V_{\mathrm{H}}} Q = \frac{1}{C_{\mathrm{H}}} Q, \tag{A5}$$

and thus the hydraulic capacitance $C_{\mathrm{H}}$ is directly proportional to the volume amount and gives the pressure change according to a net flow variation into a control volume.

*Competing interests.* The authors declare that they have no conflict of interest.

*Acknowledgements.* The research presented in this paper was part of the DOT500 ONT project, which was conducted by DOT in collaboration with Delft Univeristy of Technology and executed with funding received from the *Ministerie van Economische zaken via TKI Wind op Zee, Topsector Energie.*




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
