# Peer review of "Control design, implementation and evaluation for an in-field 500 kW wind turbine with a fixed-displacement hydraulic drivetrain"

_Wind Energy Science, 2018_

## Referee Comment (RC1) · Anonymous Referee #1 · 7 Jun 2018

Review of Paper: Control design, implementation and evaluation for an in-field 500kW wind turbine with a fixed-displacement hydraulic drivetrain by S.P. Mulders, N.F. Boudewijn Diepeveen, and J.W. van Wingerden submitted to Wing Energy Science Discussions

General Recommendation This paper presents an interesting analysis of a novel hydraulic wind turbine concept. The concept consists on replacing the conventional mechanical drivetrain components with a seawater pump directly driven by the wind turbine, whose outlet flow is directed to a Pelton generator. However, the analysis pre-

sented in the paper refers to an intermediate solution, in which the seawater pump is driven by a close loop oil-based hydrostatic transmission. The paper topic is certainly relevant for the journal; the approach is rigorous and the authors appear to be very familiar and qualified for work in the field. However, the paper could be significantly improved in certain aspects. Therefore, this Reviewer recommend its publication only after major changes are implemented to the submitted manuscript. Recommendations details [Major] The paper is quite long, it contains too many equations and figures. This Reviewer suggests the authors to reduce the number of equations and figures. Some suggestions are provided in the following comments. Consider also that this Reviewers is asking for some additional details, therefore some additional figures might be necessary in the revised version of the paper. [Minor] Not sure about the significance of Fig. 2, since the concept is quite obvious. In any case, if the authors decide to keep this image, this Reviewer suggests to include labels for the different components represented [Major] The authors provide a quite exhaustive overview of the past effort, which is very appreciated. However, at pag. 2, they affirm "To date, none of the 5 above described full hydraulic concepts made its way to a commercial product. All concepts use oil as the hydraulic medium because of the favorable fluid properties and wide component availability, but therefore also need to operate in closed-loop." This Reviewer has two problems with such sentence: - For the size of the components required for wind turbine applications, there are almost no available commercial products. Those chosen by the authors in their work are probably the among the very few ones available (considering also that they had to turn a motor into a pump!). This is because as the authors stated, there are no successful application for hydraulic wind turbines. Therefore, if there is no market (thus no demand), there is no offer. The message is that nowadays someone wants to design a hydraulic wind turbine, he/she necessitates to design the hydraulic components as well (or partner with a component manufacturer to get a unit specially designed). - The reference to close loop (close circuit ?) hydraulic transmissions (HTs) is questionable. HTs for many mobile applications (wheel loaders, excavators, etc) are close loop, but again the components for these HTs are

too small for wind turbine applications. HTs can also be open-loop. Many HTs for areal platforms, forklifts, hydraulic fan drives are open-loop. What are the requirements that determine the need of having a close loop HT for a wind turbine application? This Reviewer can have some guesses, but this should be better addressed.

[Minor] Was the concept of the paper presented also at the IFK2018 conference? A better reference to that paper, and the novel contents of this paper, should be provided.

[Minor] Figure 3 might not be necessary, can be removed.

[Major] Section 2.2. The hydraulic circuit needs to be better detailed. A more realistic ISO schematic with respect to the one provided in Fig. 6 is needed. The authors give the impression that a pump can be simply be coupled with a motor to form a close loop HT. However, other components are needed to guarantee the operation of the system: - Is a charge pump present? How was that sized? Can be neglected in the analysis? Why? What is the pressure level of the low pressure line? Is a flushing valve / cooling of the hydraulic motor present / necessary for the long operation of the system? A HTs for continuous operation usually necessitates for a significant oversizing of the charge unit for cooling purposes. - Circuit of the water pump. The authors say that there is an external centrifugal pump, which seems to be connected in series with the fixed displacement pump. How is the schematic? Is there a relief valve in between to provide a reference pressure level? Why this part can be neglected in the subsequent analysis? - Pelton Turbine. The concept of using a Pelton Turbine is very interesting. However, it seems that the Head [m] of this turbine is way above to the existing Pelton turbines, so that it might be impossible to borrow an existing design. What is the specific speed of the Pelton Turbine of this paper? Is a commercial Pelton wheel available? Is a two-jet turbine such as the one of Fig. 5 sufficient? This is not the scope of this paper, however, the authors could be more clear on this part, perhaps using more references.

[Major] At page 7, the authors say "After the water flow exits the spear valve, the aim to operate the Pelton turbine generator combination at maximum efficiency is a decoupled

control objective from the rest of the drivetrain, and is outside the scope of this paper." Actually, this sentence is at the basis of many assumptions made in the development of the model and the controller design. This Reviewer, although without specific experience in designing HTs for wind turbine applications, has some conceptual doubts on this choice made by the authors. A HT has to be designed according to the features of both the load and the prime mover; this also drives several choices of constant torque (variable displacement pump) or constant power (variable displacement motor) HTs. In this case, the authors decide to neglect the features of the user (the Pelton wheel). Is that correct? To this Reviewer, it is like affirming that all the points that satisfies Eq. 12 (relation nozzle area and HT pressure) are indifferent for the Pelton turbine. This is quite hard to believe. The Pelton turbine should have preferred operating points that the HTs should be able to handle. This is a very basic question that the authors should address properly. Otherwise their proposed controller might not be beneficial on a real application.

[Major] Section 3.1.2. The authors here affirm "the volumetric efficiency of a pump or motor is generally high and fairly constant over the entire operating range". For a simplified model the assumption of constant efficiency could be a fair starting point. But the statement that hydraulic pumps and motors have a constant volumetric efficiency for any pressure and shaft speed is clearly wrong. Otherwise, all the literature on empirical efficiency models (starting from Merritt in the 60s), standards for measuring volumetric efficiency (ISO, etc), tribological models for studying the lubricating gap flows, would not be justified. Particularly at low speed, the volumetric efficiency can be particularly low for both pumps and motors. Please revise this statement and better justify the assumption of constant efficiency, which can be very limited.

[Minor] Section 3.2.1. here there are several equations that are well known. This section could be reduced

[Major] Section 3.2.2. In the list of assumptions it is stated that the inertia of the hydraulic components is neglected. While it is true that hydraulic components have fast

dynamic, in comparison with other technologies for transmitting power, it has to be proven that within a hydraulic system the hydraulic line is the element with slowest dynamic. This statement, in general is not true, and Merritt never affirmed that. Moreover, the authors consider infinitely rigid lines, therefore the "fastest" lines possible (is this realistic?). Please justify this statement.

[Major] Section 4.4.1. The authors say "hydraulic components are known to be more efficient in high-load operating conditions, it might be advantageous for a hydraulic drivetrain to operate the rotor at a lower tip-speed ratio.". This statement can be arguable. First, shaft speed has a major effect, and not all units have a clear trend with load. Can the authors provide the overall efficiency plot for the commercial units they utilize (even in normalized form?). This is very important, because all the controllers of case 1 and case 2 are based on this assumption!

[Minor] Pag. 21. The reference to Fig 14 might be wrong, since the figure refers to mechanical efficiency.

[Major] 4.2.1. L=50m. . . are the pump and motor connected by a 50 m straight line? If there are line discontinuities, some terms, particularly the inductance terms, can be entirely wrong.

---

## Referee Comment (RC2) · Anonymous Referee #2 · 18 Jun 2018

Thank you for your submission. In general I found the paper to be interesting and well-presented. Background literature was complete and informative, and the introductory material explains well what this paper adds to the growing literature. The figures and illustrations are particularly well done and helpful. The writing is clear, with only very few grammatical errors. The paper is well structured, such that one new to hydraulic-drivetrain wind can follow. Finally, the inclusion of field results and comparing to the theoretical work is informative. Excellent work.

Therefore, only a few comments to be given:

Overall: Is there provided, or could the authors provide, a quick impression of how the efficiency of the proposed system, in total, would compare to a similar conventional system? For example, given the same rotors, what would a standard efficiency to final electrical power be (90%?) and what would it be for this system?

Specific:

3.2.1: "and vice versa for the latter.." I could not fully understand what is meant by this Could a Bode plot of the transfer functions be included to visualize the inverted notch functions?

Eq 38: The B-matrix in this version includes the inputs? There is a dot following the matrix but it is not clear what the dot product will be with? In eq 54 there are only 3 columns total for the 3 B matrices, but are there 4 variables provided in this version?

Section 4.1.1: "advantageous for ... operate a lower tip-speed ratio", this is counter-intuitive, but do I understand correct that although the rotor power will be reduced, the improved hydraulic efficiency will lead to higher final electrical power? Is this demonstrated conclusively?

Section 4.1.2: Does the lower TSR also risk increased occurence of dynamic stall?

Fig 16: Title is incomplete

Fig 18: The legend is hard to understand, why the lower-case bold "without" following the period? Since the phase margin is discussed later, could it be indicated in this figure?

Fig 22: Color legend missing label

Fig 24: Good figure, just wish to confirm, is rotor speed scaled correctly? Does it always stay so close to its maximum? Or is this period special in that there is only a brief excursion into region 2 (which makes sense, you've selected a period covering 2,2.5,3) just want to be sure.

[Figure]

An interesting, and well-presented paper.

---

## Referee Comment (RC3) · Anonymous Referee #3 · 13 Jul 2018

Paper is very well written and subject matter thoroughly presented. Some general comments: towards the beginning of the paper it is stated that "To date, none of the above described full hydraulic concepts made its way to a commercial product", this merits some justification as to why the concept of hydraulic transmission for wind turbines has not been commercially viable so far, and whether the presented work can potentially overcome these barriers to market. One minor comment would be to add some labelling of the components in figure 2.

There are no further comments in addition to what has already been highlighted by the

other two reviewers.

---

## Referee Comment (RC4) · Anonymous Referee #4 · 13 Jul 2018

The research in the paper is original, well conceived, and of interest to the readers of this journal. The other reviewers have done a good job in providing a detailed review.

I only have one comment. The reviews advocate maximizing the torque coefficient rather than maximizing the power coefficient for this particular case. They further demonstrate that this provide more energy for the system considered since hydraulic components in this particular case are more efficient at higher torque and lower speed.

It is important to point out that while the torque coefficient optimizing approach is ad-

vantageous in this particular case, it is not true in general. The overall efficiency of a hydraulic pump or motor is the product of the mechanical efficiency and the volumetric efficiency. Depending on the particulars of the unit and its operating conditions, either of these might be dominant. A truly rigorous approach would be to optimize the system power coefficient with all losses included. This would work for any case.

In their final version the authors must clearly state that the torque coefficient optimization approach they advocate is true in this case, but is not true in general.

---

## Referee Comment (RC5) · Anonymous Referee #5 · 25 Jul 2018

This is an interesting paper that deals with the use of hydraulic transmission systems to enable centralised conversion of wind power into electricity in offshore wind farms. A key advantage of this approach is the significant reduction in the nacelle weight. The paper is relevant and very timely given the present drive by industry to develop larger and heavier rotors. The following is a summary of the main comments that need to be addressed by the authors before the paper may be published:

1. Figure 5 may be deleted without affecting the quality of the paper.
2. Table 1: It is also convenient for the reader to include the rated power of the motors
3. Page 6: The process of matching the pump, motor and Pelton turbine to the available wind turbine should be elaborated in further detail.
4. Page 9: specify the aerofoil data used in plotting Fig. 7.
5. Fig. 14: for ease of comparison, the two plots should have the same colour scale for the mechanical efficiency.
6. Figure 15: Possible design amendments to the system to enhance the overall conversion efficiency should be elaborated in further detail.

Minor comments:

1. Figure 1 should ideally be presented on the same page where it is being referred to in the text.
2. Page 6, line 13 – remove coma after 'in such a way'.
3. Eqt. (11) may be deleted as derived of Eqt (12) is well known.
4. Page 12, line 9 – remove coma after 'into the system'.

---

## Author Comment (AC1) · 27 Jul 2018

| | |
|---|---|
| Date | July 27, 2018 |
| Our reference | n/a |
| Your reference | n/a |
| Contact person | S.P. Mulders |
| Telephone/fax | +31 (0)6 5573 6149 / n/a |
| E-mail | S.P.Mulders@TUDelft.nl |
| Subject | Response to reviewers |

**Delft University of Technology**

Delft Center for Systems and Control

Address
Mekelweg 2 (3ME building)
2628 CD Delft
The Netherlands

www.dcsc.tudelft.nl

Reviewers
*Wind Energy Science*

Dear Reviewers,

First of all, the authors would like to thank the reviewers for their positive and constructive feedback. We believe that the comments help us to significantly improve the quality of the paper. The objective of this document is to respond to the points raised by the reviewers (blue) and to provide an overview of the actions that are taken (red). When in this response the authors refer to adjustments in a particular section, figure or table, we ask the reviewer to refer to the marked-up manuscript version to evaluate the changes.

The document consists of five sections, each addressing the comments of the reviewers separately.

Yours sincerely,

Sebastiaan Paul Mulders
Niels Frederik Boudewijn Diepeveen
Jan-Willem van Wingerden

| | |
|---|---|
| Enclosure(s): | Response to comments of Reviewer 1 |
| | Response to comments of Reviewer 2 |
| | Response to comments of Reviewer 3 |
| | Response to comments of Reviewer 4 |
| | Response to comments of Reviewer 5 |

**Response to comments of Anonymous Referee #1**

**Reviewer 1 comments**: This paper presents an interesting analysis of a novel hydraulic wind turbine concept. The concept consists on replacing the conventional mechanical drivetrain components with a seawater pump directly driven by the wind turbine, whose outlet flow is directed to a Pelton generator. However, the analysis presented in the paper refers to an intermediate solution, in which the seawater pump is driven by a close loop oil-based hydrostatic transmission. The paper topic is certainly relevant for the journal; the approach is rigorous and the authors appear to be very familiar and qualified for work in the field. However, the paper could be significantly improved in certain aspects. Therefore, this Reviewer recommend its publication only after major changes are implemented to the submitted manuscript.

The authors thank the reviewer for his/her thorough review, invaluable comments and remarks. The considerations are especially informative as in particular considerations are raised on the analysis and justification of the hydraulic drivetrain and its components. Processing these comments very much helped the authors in closing the gap between system design and mathematical evaluation of the employed turbine with hydraulic transmission.

1. [Major] The paper is quite long, it contains too many equations and figures. This Reviewer suggests the authors to reduce the number of equations and figures. Some suggestions are provided in the following comments. Consider also that this Reviewers is asking for some additional details, therefore some additional figures might be necessary in the revised version of the paper.

   The authors agree with the fact that this paper is quite long. In the revised version, this issue is addressed by revising the relevance of all content. We are grateful of the reviewer suggestions, and the authors will take these remarks into account.

   The most notable changes are noted:
   - Section 2.2 is extended to provide a more detailed description of the DOT500 prototype set-up;
   - Section 3.2.1 is shortened, now only showing the results. The derivation is moved to the appendix;
   - Section 4.2.2 is shortened.

2. [Minor] Not sure about the significance of fig. 2, since the concept is quite obvious. In any case, if the authors decide to keep this image, this Reviewer suggests to include labels for the different components represented

   Thank you for this comment. The authors agree that this figure is not of direct scientific relevance, however, it presents a nice comparison of the (potential) space-advantage of a wind turbine with hydraulic drivetrain.

   The authors decided to keep the figure, but included a description of the distinct components, as suggested by the reviewer.

3. [Major] The authors provide a quite exhaustive overview of the past effort, which is very appreciated. However, at pag. 2, they affirm "To date, none of the above described full hydraulic concepts made its way to a commercial product. All concepts use oil as the hydraulic medium because of the favorable fluid properties and wide component availability, but therefore also need to operate in closed-loop." This Reviewer has two problems with such sentence:

   - For the size of the components required for wind turbine applications, there are almost no available commercial products. Those chosen by the authors in their work are probably the among the very few ones available (considering also that they had to turn a motor into a pump!). This is because as the authors stated, there are no successful application for hydraulic wind turbines. Therefore, if there is no market (thus no demand), there is no offer. The message is that nowadays someone wants to design a hydraulic wind turbine, he/she necessitates to design the hydraulic components as well (or partner with a component manufacturer to get a unit specially designed).

     The authors agree with the reviewer on this point. DOT is founded with the philosophy that offshore wind can be simplified and exploited more efficiently by centralizing energy production. However, to date, a water pump with capabilities to operate under high load and low speed is not commercially available. DOT is very aware of this, and is therefore (1) working to develop a seawater pump by their selves, and (2) actively cooperating with renowned (water) pump manufacturers on the development of such a pump.

     The in-field tests with an intermediate drivetrain including a closed oil circuit, is the first step towards the final DOT concept. The goal of DOT therefore is to abandon the use of oil all together by making wind turbines to cooperate and use what is abundantly available offshore: seawater. The authors (and DOT) do realize that a lot of hurdles need to be overcome before this goal becomes reality.

- The reference to close loop (close circuit ?) hydraulic transmissions (HTs) is questionable. HTs for many mobile applications (wheel loaders, excavators, etc) are close loop, but again the components for these HTs are too small for wind turbine applications. HTs can also be open-loop. Many HTs for areal platforms, forklifts, hydraulic fan drives are open-loop. What are the requirements that determine the need of having a close loop HT for a wind turbine application? This Reviewer can have some guesses, but this should be better addressed.

  The point raised by the reviewer is valid. A closed circuit drivetrain for a wind turbine utilizing oil as the hydraulic medium is needed because:
  (1) Operating a turbine with oil at remote offshore locations might pollute the environment in the event of a calamity;
  (2) Not having to provide a continuous fresh oil supply to the circuit;

  A consideration for operating in closed circuit is cooling of the hydraulic medium. When losses in hydraulic components are significant and the natural convection of heat to the surroundings is insufficient for cooling, an additional cooling circuit needs to be incorporated.

  The authors recognize that the statement is posed too strong and lacks further explanation. Therefore, the comments of the reviewer and our considerations are processed in the revised version of the introduction. Furthermore, we made sure that we reserved the term "circuit" for hydraulic matters, and "loop" for control purposes.

4. [Minor] Was the concept of the paper presented also at the IFK2018 conference? A better reference to that paper, and the novel contents of this paper, should be provided.

   Yes, this is correct. We did not yet include a reference to the conference paper, as it was not published at time of submission of the WES manuscript.

   The authors included a reference to the conference article in the introduction, and stated clearly what the contribution of this paper is as opposed to the conference article.

5. [Minor] figure 3 might not be necessary, can be removed.

   Thank you for this comment. Because of the length and increased complexity of this paper we included a paper organization flow chart. However, providing both a textual and graphical outline might indeed be redundant.

   The figure is removed from the manuscript.

6. [Major] Section 2.2. The hydraulic circuit needs to be better detailed. A more realistic ISO schematic with respect to the one provided in fig. 6 is needed. The authors give the impression that a pump can be simply be coupled with a motor to form a close loop HT. However, other components are needed to guarantee the operation of the system:

   The authors agree with the reviewer that the hydraulic diagram should be better detailed. The included simplified hydraulic diagram was a trade-off between complexity and relevance for the drivetrain modeling provided in the manuscript. In the real-world set-up, numerous additional components were in place for turbine operation.

   Reconsidering the performed trade-off, the authors revisited the hydraulic diagram by including vital components. The components that are not included/considered for modeling and analysis are presented in gray. In this way, the authors think that the updated diagram provides a middle course from a system design and theoretical modeling point of view.

- Is a charge pump present? How was that sized? Can be neglected in the analysis? Why? What is the pressure level of the low pressure line? Is a flushing valve / cooling of the hydraulic motor present / necessary for the long operation of the system? A HTs for continuous operation usually necessitates for a significant oversizing of the charge unit for cooling purposes.

Correct, charge pumps are presents in both the oil and water circuits. The oil charge pump was sized in such a way that a sufficient flow with a constant (controlled) feed-pressure of 21 bar to the oil pump could be delivered. Additionally when cooling was required, the charge pump supplied more flow to be directed through the parallel cooling circuit connected by a pressure relief valve (see updated hydraulic diagram). For the water circuit, the centrifugal charge pump provided a lower charge pressure of 2.6 bar. This difference in charge pressure is due to the different pump types used: the radial piston oil pump (motor) requires a higher charge pressure, as this is used to actively push the piston bearings to the cam ring; whereas the water plunger pump largely alleviates this requirement.

As for modeling purposes of the closed oil circuit only pressure differences over the hydraulic components (taking into account mechanical/volumetric efficiency losses) are considered, the feed pressure is left out from the analysis. For the open water circuit however, the feed pressure is neglected because of its low value and for convenient derivation of the passive torque control strategy.

Cooling equipment was indeed necessary for the closed oil circuit, and a flush oil cooling circuit was in place to ensure long-term operation and prevent the working fluid from overheating.

The updated manuscript now provides a more detailed description of the aspects discussed above in Section 2.2.

- Circuit of the water pump. The authors say that there is an external centrifugal pump, which seems to be connected in series with the fixed displacement pump. How is the schematic? Is there a relief valve in between to provide a reference pressure level? Why this part can be neglected in the subsequent analysis?

  The reviewer is correct, the centrifugal pump is connected in series to the water pump, and the updated hydraulic diagram presents the working principle. To prevent a disturbed flow entering the water pump, a two-reservoir set-up is used. The speed of the centrifugal pump is controlled to maintain feed pressure of approximately $2.6$ bars. The charge pump is enabled before the water pump starts speeding up to ensure feed-pressure and thus to avoid cavitation. The low-pressure side of the water circuit does not contain a pressure-relief valve, as the water plunger pump allows for a direct feed-though of the flow; the high-pressure side however does include a pressure relief valve.

  Section 2.2 is updated with the details provided in the response to the reviewer.

- Pelton Turbine. The concept of using a Pelton Turbine is very interesting. However, it seems that the Head [m] of this turbine is way above to the existing Pelton turbines, so that it might be impossible to borrow an existing design. What is the specific speed of the Pelton Turbine of this paper? Is a commercial Pelton wheel available? Is a two-jet turbine such as the one of fig. 5 sufficient? This is not the scope of this paper, however, the authors could be more clear on this part, perhaps using more references.

  Thank you for raising this comment, we could have been more clear on this aspect. Pelton turbines are highly specialized pieces of equipment and need to be design for specific condition requirements [1]. The Sy Sima $315$ MW turbine in Norway, for $88.5$ bar of head pressure is to date the largest known [2]. The employed custom manufactured Pelton turbine for the DOT500 is designed to match the nominal pressure and the speed conditions of the connected electrical generator.

  A custom-made Pelton turbine is designed such that the efficiency is optimal under the expected operational conditions. For this, the turbine is designed for optimal operation using 2 spear valves, subject to a nominal flow of 58 l/min. Graphs of the turbine manufacturer (given below) show that the efficiency is primarily a function of the supplied flow, and to a lesser extent of the head. The red line indicates operation with 1 spear valve, the blue line 2 spear valves.

[Figure]

The efficiency aspect is confirmed later by experiments executed by DOT, of which an efficiency evaluation figure is also given below. In the figure an evaluation of the combined spear valve-Pelton efficiency from hydrostatic fluid to mechanical power at the generator axis is given as a function of flow and pressure. Whereas during the experiment the flow was not sufficient to explore the overall characteristics, the results clearly show that the steepest partial efficiency gradient goes with flow; at higher flow rates the gradient with respect to pressure becomes negligible.

[Figure]

In the updated manuscript, more information on the custom-design Pelton turbine is given. Also the nominal operating conditions are discussed, and relevant references are included.

7. [Major] At page 7, the authors say After the water flow exits the spear valve, the aim to operate the Pelton turbine generator combination at maximum efficiency is a decoupled control objective from the rest of the drivetrain, and is outside the scope of this paper. Actually, this sentence is at the basis of many assumptions made in the development of the model and the controller design. This Reviewer, although without specific experience in designing HTs for wind turbine applications, has some conceptual doubts on this choice made by the authors. A HT has to be designed according to the features of both the load and the prime mover; this also drives several choices of constant torque (variable displacement pump) or constant power (variable displacement motor) HTs. In this case, the authors decide to neglect the features of the user (the Pelton wheel). Is that correct? To this Reviewer, it is like affirming that all the points that satisfies Eq. 12 (relation nozzle area and HT pressure) are indifferent for the Pelton turbine. This is quite hard to believe. The Pelton turbine should have preferred operating points that the HTs should be able to handle. This is a very basic question that the authors should address properly. Otherwise their proposed controller might not be beneficial on a real application.

The reviewer correctly points out that the system design, as well as the applied control strategy should go hand-in-hand. Changing the operational strategy on the wind turbine side affects the operating point of the Pelton turbine and thereby for example the maximum amount of energy it can extract from the given flow. This is completely understood by the authors.

The features of the Pelton wheel are not neglected. It is known from literature [3][4] that the ratio of tangential Pelton and water jet speed needs to be maintained at approximately $1/2$. As the Pelton wheel is mechanically coupled to an asynchronous generator, which can change its operational speed, a pressure measurement is used to determine the most favorable (speed) operating point to be as efficient as possible, given the conditions it is subjected to. However, the Pelton wheel will in the given set-up always be subjected to varying conditions, and thus suboptimal operation in the considered drivetrain using fixed-displacement components. This is for now a design choice, and further research needs to be conducted to elaborate on Pelton design and efficiency maximization given the varying operational conditions.

The point of which operational path is most efficient, given varying Pelton conditions, remains. Operation at $C_{\tau,\mathrm{max}}$ will result in higher pressures for equal flows when compared to $C_{\mathrm{p,max}}$ operation. As was concluded in the previous question, the main driver determining the Pelton efficiency is the flow it is subjected to, whereas the head has negligible influence.

The above given considerations are included in the manuscript in Section 2.2.

8. [Major] Section 3.1.2. The authors here affirm "the volumetric efficiency of a pump or motor is generally high and fairly constant over the entire operating range". For a simplified model the assumption of constant efficiency could be a fair starting point. But the statement that hydraulic pumps and motors have a constant volumetric efficiency for any pressure and shaft speed is clearly wrong. Otherwise, all the literature on empirical efficiency models (starting from Merritt in the 60s), standards for measuring volumetric efficiency (ISO, etc), tribological models for studying the lubricating gap flows, would not be justified. Particularly at low speed, the volumetric efficiency can be particularly low for both pumps and motors. Please revise this statement and better justify the assumption of constant efficiency, which can be very limited.

The reviewer makes a very valid point, and we agree with it. The reason we have chosen to assume a constant volumetric efficiency factor is (1) the fact that for most of the given components, no volumetric efficiency data is available, and (2) the aim is to provide a simplified model of the hydraulic drivetrain.

The assumption of a volumetric efficiency is revised in the updated manuscript.

9. [Minor] Section 3.2.1. here there are several equations that are well known. This section could be reduced.

Thank you for this comment. The authors recognize that Section 3.2.1 is lengthy and reduced it. Now, only the major results are presented. However, we would like to be as complete as possible, seen the journal we are publishing in is not primarily focussed on hydraulics. Therefore, we moved the derivation of the results to the appendix.

Section 3.2.1 is shortened and the derivation of the results has been moved to the appendix.

10. [Major] Section 3.2.2. In the list of assumptions it is stated that the inertia of the hydraulic components is neglected. While it is true that hydraulic components have fast dynamic, in comparison with other technologies for transmitting power, it has to be proven that within a hydraulic system the hydraulic line is the element with slowest dynamic. This statement, in general is not true, and Merritt never affirmed that. Moreover, the authors consider infinitely rigid lines, therefore the fastest lines possible (is this realistic?). Please justify this statement.

Thank you for this comment, the authors agree with the reviewer that the (slow) line dynamics cannot be discarded. As exact specifications of the line bulk modulus are unknown (not publicly available), we decided to take a reasonable value of $K_l = 0.8$ GPa from [5], which is twice as low as the bulk modulus value taken for the oil column. The equivalent bulk modulus is calculated by the relation $K_e = (1/K_f + 1/K_l)^{-1} = 0.52$ GPa. The equivalent value is used subsequently in the remainder of the paper.

The assumption of infinitely rigid lines is removed from the analysis in Section 3.2.2, and Appendix A is updated to include relation for calculating the equivalent bulk modulus. The resulting equivalent modulus is subsequently used in the analyses throughout the different sections of the paper. Furthermore, Section 3.1.1 of the manuscript is updated, and now includes a more elaborate justification on why the drivetrain component dynamics are neglected and assumed as analytic expressions.

11. [Major] Section 4.1.1. The authors say "hydraulic components are known to be more efficient in high-load operating conditions, it might be advantageous for a hydraulic drivetrain to operate the rotor at a lower tip-speed ratio". This statement can be arguable. first, shaft speed has a major effect, and not all units have a clear trend with load. Can the authors provide the overall efficiency plot for the commercial units they utilize (even in normalized form?). This is very important, because all the controllers of case 1 and case 2 are based on this assumption!

The authors agree with the comment made by the reviewer, and the statement the reviewer refers to is changed. However, the manuscript already includes (mechanical) efficiency data for the oil pump and motor, given in Figure 13. The figure includes the proposed operational strategies (case 1 and case 2), and a steady-state analysis of the total drivetrain efficiency for both strategies is given in Figure 14.

The titles of the plots in Figure 13 and the legend in Figure 14 are now updated, to make their purpose more clear. Also, the statement the reviewer refers to is adjusted, and an additional consideration on the efficiency aspect is made in Section 4.1.2.

12. [Minor] Pag. 21. The reference to fig 14 might be wrong, since the figure refers to mechanical efficiency.

Thanks for pointing out this mistake. The reference should be pointing to Fig. 13.

The reference is corrected in the revised manuscript.

13. [Major] 4.2.1. L=50m... are the pump and motor connected by a 50 m straight line? If there are line discontinuities, some terms, particularly the inductance terms, can be entirely wrong

Indeed, apart from a swivel which enables continuous yaw motion located below the nacelle, the high pressure lines have no discontinuities, and run from the nacelle all the way to the oil motor located in the monopile. Furthermore, the rotor inertia, which is expressed in terms of hydraulic induction, is predominant in the lumped induction term. The contribution of the hydraulic inductance term is thus negligible, and discrepancies would have a negligible effect on the analysis.

**Response to comments of Anonymous Referee #2**

**Reviewer 2 comments**: Thank you for your submission. In general I found the paper to be interesting and well-presented. Background literature was complete and informative, and the introductory material explains well what this paper adds to the growing literature. The figures and illustrations are particularly well done and helpful. The writing is clear, with only very few grammatical errors. The paper is well structured, such that one new to hydraulic drivetrain wind can follow. Finally, the inclusion of field results and comparing to the theoretical work is informative. Excellent work.

Thank you, the authors are pleased to read this!

Therefore, only a few comments to be given, overall:

1. Is there provided, or could the authors provide, a quick impression of how the efficiency of the proposed system, in total, would compare to a similar conventional system? For example, given the same rotors, what would a standard efficiency to final electrical power be (90%?) and what would it be for this system?

   For the described DOT prototype, the total power transmission efficiency was predictably low, as a result of the double hydraulic circuit. In the below-rated region an efficiency is attained of $30 - 45\,\%$ depending on the operating conditions, whereas in the above-rated region a consistent drivetrain efficiency of $45\,\%$ is attained. As described in the manuscript, off-the-shelf components are used, of which the optimal efficiency operational envelopes do not match. The drivetrain has a fixed-volumetric displacement, which means that the pressure and flow changes according to the turbine operating conditions. Figure 13 shows that drivetrain components have a specific region in which they yield maximum efficiency.

   The above given reasoning holds for the described prototype. It is however yet unclear what the drivetrain efficiency of the final DOT concept will be, as the seawater pump is still under development. An earlier PhD thesis on hydraulic wind turbine networks [7] provides an estimate on the overall conversion efficiency of conventional and hydraulic wind turbines of $82 - 84\,\%$ and $70 - 80\,\%$, respectively.

   The efficiency numbers attained with the intermediate DOT500 prototype are added to Section 2.1.

   Specific:

2. 3.2.1: "and vice versa for the latter.." I could not fully understand what is meant by this. Could a Bode plot of the transfer functions be included to visualize the inverted notch functions?

The authors did separate the theory from the results. For this reason, in Section 3.2.1, only theory is provided, whereas in Section 4.2.1 an illustrative example is given, which considers the system and hydraulic properties of the DOT500 system. In the latter mentioned section, a Bode plot of the of the transfer function $G_{Q/\Delta p}(s)$ is given. For clarity reasons (and considering the length of the paper), a visualization of $G_{\Delta p/Q}(s)$ is omitted in the manuscript, but given in the figure below. It is shown that the inverted notch characteristics is still present. However, exciting the flow (instead of pressure) results in amplification/transmission to pressure in a wider frequency region for shorter line lengths. For longer line lengths, the amplification magnitude increases, but at a more specific interval. This effect is a result of the inverse proportionality between the damping coefficients $\zeta_Q$ and $\zeta_p$.

[Figure]

In Section 3.2.1 the phrase "and vice versa for the latter.." is removed and replaced with a more convenient description. The paragraph now also references to the illustrative example in Figure 15.

3. Eq 38: The B-matrix in this version includes the inputs? There is a dot following the matrix but it is not clear what the dot product will be with? In eq 54 there are only 3 columns total for the 3 B matrices, but are there 4 variables provided in this version?

The representation given by Eq. 38 is the rewritten form of the dynamic system derived in in Eqs. (33)-(37). There is no dot-product after the input vector, this is just punctuation to indicate the end of the sentence. The pressures $\Delta p_\mathrm{h}$ and $\Delta p_\mathrm{b}$ cannot be controlled directly. For this reason, the rotor torque and spear valve pressure characteristics are evaluated and linearized at different operating points. By doing so, a linear state space system defined in Eq. (54) is obtained. By substitution of the linearized characteristics, defined in Eqs. (46) and (51), the terms redistribute in the $A$, $B$ and $B_\mathrm{U}$ matrices: some are defined in the state vectors, others can be regarded as control inputs or wind disturbance inputs in $B$ and $B_\mathrm{U}$.

The authors hope to have clarified the unclarities, and slightly updated the section to improve readability.

4. Section 4.1.1: "advantageous for ... operate a lower tip-speed ratio", this is counterintuitive, but do I understand correct that although the rotor power will be reduced, the improved hydraulic efficiency will lead to higher final electrical power? Is this demonstrated conclusively?

The reviewer is correct. Normal wind turbines operate the rotor at the maximum power coefficient, maximizing the efficiency in the below-rated region. For the DOT500, however, the wind turbine drivetrain is retrofitted, while retaining the original turbine rotor. As hydraulic components are in general more efficient in high-load operating conditions, we additionally perform an analysis for operating the turbine at the maximum possible torque coefficient. The maximum torque coefficient is located at a lower tip-speed ratio (lower rotor speeds, higher torques for equal wind speeds), and corresponds with a lower power coefficient.

So indeed, from a aerodynamic efficiency perspective this is unfavorable, but from a hydraulic drivetrain perspective this might result in an overall efficiency advantage. In the subsequent section, an efficiency analysis is given on a component level for both operating cases in Figure 13, and an overall evaluation of the drivetrain efficiency is presented by Figure 14. The analysis takes into account the reduced rotor power coefficient for operation at maximum torque.

The titles of the plots in Figure 13 and the legend in Figure 14 are now updated, to make their purpose more clear. Also the introductory paragraph of Section 4.1.1 is updated, and a concluding remark referring to the next section where the actual efficiency analysis is performed is added.

5. Section 4.1.2: Does the lower TSR also risk increased occurrence of dynamic stall?

Thank you, this is a very good question. We cite the following phrase from [8]:

"Stall on lifting surfaces is commonly encountered, mostly undesired, and occurs when a critical angle of attack is exceeded. Depending on the unsteady rate of change of the airfoil's angle of attack, static and dynamic stall are distinguished. (...) During dynamic stall, the shear layer rolls up into a large scale dynamic stall vortex which grows locally and temporally until vortex induced separation occurs. During static stall on the other hand, the shear layer rolls up continuously into large-scale structures that grow spatially."

So indeed, there is an increased occurrence of dynamic stall, especially in turbulent wind conditions when the angle of attack continuously varies.

Furthermore, from a discussion with a professor in aerodynamics from our faculty, it became clear that (dynamic) stall could indeed occur in the region of the blade root. Stalling of a larger blade would result in increased loading with a reduced power capture, however, as we are not stalling during normal operation, the effects on loads should be minor. He also clarified that dynamic stalling could even be slightly beneficial, as it introduces a dissipating/damping effect.

We have to admit that we did not perform a detailed analysis on this aspect. The aim of the in-field test was to show the feasibility of the hydraulic drivetrain, and while we ensured safe operation of the turbine, effects such as dynamic stall were disregarded. The authors have noted the comment, and the effects of (dynamic) stall will be considered in later stages of the project.

6. Fig 16: Title is incomplete

Correct, thank you for pointing out. We referred back to the submitted manuscript, but there the title is correct. Somehow, processing of the manuscript during upload must have changed the title by accident.

7. Fig 18: The legend is hard to understand, why the lower-case bold "without" following the period? Since the phase margin is discussed later, could it be indicated in this figure?

Thank you for pointing out this mistake, the point should be a comma. The suggestion of indicating the phase margin is taken, and both bode plots are updated.

Also, again, the upload process changed the figure, by omitting some symbols. The correct figure is shown below for reference.

[Figure]

The mistake in the caption is fixed, and the Bode plots include an indication of the phase margin (PM).

8. Fig 22: Color legend missing label

The point raised by the reviewer is not entirely clear for the authors. Figure 21 is a representation of the tip-speed ratio for a range of turbine operating points. This is also stated title and the caption of the figure. Adding an additional label to the color legend would be redundant in our opinion.

9. Fig 24: Good figure, just wish to confirm, is rotor speed scaled correctly? Does it always stay so close to its maximum? Or is this period special in that there is only a brief excursion into region 2 (which makes sense, youve selected a period covering 2,2.5,3) just want to be sure.

Thank you for this comment. Yes, the rotor speed is scaled correctly, however, we took a part of the time-series where the environmental conditions were such that the turbine operated around region 2.5. What we want to show in this figure is how the spear valve torque controller (only active in region 2.5) works as expected and switches nicely to region 2 (no control), and region 3 (pitch control).

The text is adjusted slightly such that the purpose of the figure is more clear.

An interesting, and well-presented paper.

Thanks again!

**Response to comments of Anonymous Referee #3**

Paper is very well written and subject matter thoroughly presented.

Thank you, the authors are grateful to hear this.

Some general comments: towards the beginning of the paper it is stated that "To date, none of the above described full hydraulic concepts made its way to a commercial product", this merits some justification as to why the concept of hydraulic transmission for wind turbines has not been commercially viable so far, and whether the presented work can potentially overcome these barriers to market.

The authors agree with the reviewer, and a similar points has been posed by Referee #1. We would like to refer the reviewer to the answer given in Question 3-2 in our response to the first referee.

We further elaborated on this point in the introduction of the manuscript according to the comments of both referees.

One minor comment would be to add some labelling of the components in figure 2.

Thank you for this comment. We agree that labelling of the components improves the quality and relevance of the figure.

The figure is updated accordingly, now including labels indicating the components.

**Response to comments of Anonymous Referee #4**

The research in the paper is original, well conceived, and of interest to the readers of this journal. The other reviewers have done a good job in providing a detailed review.

Thank you, we are grateful to read this positive comment.

I only have one comment. The reviews advocate maximizing the torque coefficient rather than maximizing the power coefficient for this particular case. They further demonstrate that this provide more energy for the system considered since hydraulic components in this particular case are more efficient at higher torque and lower speed.

It is important to point out that while the torque coefficient optimizing approach is advantageous in this particular case, it is not true in general. The overall efficiency of a hydraulic pump or motor is the product of the mechanical efficiency and the volumetric efficiency. Depending on the particulars of the unit and its operating conditions, either of these might be dominant. A truly rigorous approach would be to optimize the system power coefficient with all losses included. This would work for any case.

In their final version the authors must clearly state that the torque coefficient optimization approach they advocate is true in this case, but is not true in general.

The authors completely agree with the referee. Only two cases are considered in this paper, namely: operation at rotor maximum torque, and maximum power coefficient. For the specific case presented in the paper, it is found that the maximum torque case results in the highest overall drivetrain efficiency. However, this claim can by no means be generalized for other wind turbines with hydraulic drivetrains. A more rigorous approach would indeed be to optimize the ideal below-rated operational trajectory subject to all component characteristics. However, to perform a more concise analysis, only the two given trajectories are evaluated. For other set-ups it could indeed be the case that a different below-rated operating strategy is beneficial.

The consideration of the reviewer and our response is processed and included in Section 4.1.2.

**Response to comments of Anonymous Referee #5**

This is an interesting paper that deals with the use of hydraulic transmission systems to enable centralized conversion of wind power into electricity in offshore wind farms. A key advantage of this approach is the significant reduction in the nacelle weight. The paper is relevant and very timely given the present drive by industry to develop larger and heavier rotors. The following is a summary of the main comments that need to be addressed by the authors before the paper may be published:

The authors thank the reviewer for his positive comment and considerations raised.

1. Figure 5 may be deleted without affecting the quality of the paper.

   Indeed, the figure could be deleted. However, seen the journal we are publishing in is not primarily focussed on hydraulics, we think that the figure is insightful and serves readers from various disciplines.

   The authors would like to leave the figure as-is in the manuscript.

2. Table 1: It is also convenient for the reader to include the rated power of the motors

   Thank you for this comment, this is indeed a nice addition.

   Table 1 now includes the available power range of all the drivetrain components.

3. Page 6: The process of matching the pump, motor and Pelton turbine to the available wind turbine should be elaborated in further detail.

   Thank you for this comment. A description of the component matching process is not provided in detail in this paper, as the aim of the authors in this paper is to focus on the modeling and control design of the considered hydraulic drivetrain. One of the authors devoted his PhD to the system design of the presented drivetrain [9], and the authors refer the reviewer to this work for further details.

   The reference to the PhD thesis, including an elaborate description of the component matching process, is now also included in Section 2.2.

4. Page 9: specify the aerofoil data used in plotting Fig. 7.

The presented power and torque coefficient curves are obtained from a Bladed wind turbine model which was shared confidentially with DOT. The model includes the requested airfoil data, but for reasons of confidentiality, this information cannot be shared publicly. However, the complete data set including the resulting power, torque and thrust coefficient tables (as a function of tip-speed ratio and blade pitch) is publicly available as an external asset [10].

Section 3.1.1 now includes a reference to the externally available data set which includes power, torque and thrust coefficient tables.

5. Fig. 14: for ease of comparison, the two plots should have the same colour scale for the mechanical efficiency.

We partly agree with the reviewer's remark. The color scales of the left and right plots range from 0.7 - 1.0 and 0.05 - 1.0, respectively. For plots with equal data (efficiency) ranges, we agree with the reviewer that equalizing them would enhance the readability. However, by using the same color bars for the presented plots would make the left plot less convenient to read by a lack of contrast (especially in grayscale). For this reason, the authors decided to leave the plots unchanged.

6. Figure 15: Possible design amendments to the system to enhance the overall conversion efficiency should be elaborated in further detail.

   A similar point is posed by reviewer #4, but focuses on the control aspect, and the authors refer the reviewer to our response in the previous section. In summary: for the considered system, only two scenarios are evaluated (maximum rotor power and maximum rotor torque trajectories) to provide a concise evaluation of both strategies on the overall drivetrain efficiency. Indeed, a more rigorous approach would be to optimize the ideal below-rated operational trajectory subject to all component characteristics.

   From a system design perspective, the presented prototype hydraulic wind turbine has the goal of showing the feasibility of a wind turbine with a hydraulic drivetrain, and is not meant to be kept as-is. In response to the first question of reviewer #2, we added the efficiency numbers of the current set-up to Section 2.1 ($30 - 45$ % below-rated, $45$ % above-rated). Also, in Section 2.1, it is stated that the set-up allows for prototyping, and provides a proof of concept for faster development towards the ideal DOT concept. It is known that the additional components and energy conversions result in a reduced overall efficiency. For an overall increased efficiency, the amount of energy conversion steps need to be reduced. It is stated in the conclusion that by discarding the oil loop in the ideal DOT concept, only including a single water pump in the nacelle, the control design process is simplified and the overall drivetrain efficiency should be greatly improved.

   Regarding the changes already made to the manuscript, the authors think that the point raised by the reviewer is clarified.

Minor comments:

1. Figure 1 should ideally be presented on the same page where it is being referred to in the text.
2. Page 6, line 13 - remove coma after in such a way.
3. Eqt. (11) may be deleted as derived of Eqt (12) is well known.
4. Page 12, line 9 - remove coma after into the system.

Thank you for pointing out these minor remarks.

1. We agree with the reviewer, however, during typesetting the paper will be converted to a two-column format, and thus the complete mark-up will be changed again. We tried to make the figure positions as convenient as possible, but we are reticent on putting too much effort in this for now. However, we certainly keep this comment in mind for all figures during the mark-up of the final version of the manuscript.

2. Thank you, we corrected this.

3. We agree with the reviewer and deleted the equation.

4. Thank you, we corrected this.

**References**

[1] H. Brekke, *Hydraulic turbines: design, erection and operation*. Norwegian University of Science and Technology (NTNU), 2001.

[2] E. Cabrera, V. Espert, and F. Martínez, *Hydraulic Machinery and Cavitation: Proceedings of the XVIII IAHR Symposium on Hydraulic Machinery and Cavitation*. Springer, 2015.

[3] Z. Zhang, "Flow interactions in pelton turbines and the hydraulic efficiency of the turbine system," *Proceedings of the Institution of Mechanical Engineers, Part A: Journal of Power and Energy*, vol. 221, no. 3, pp. 343–355, 2007.

[4] J. Thake, *The Micro-hydro Pelton Turbine Manual*. Rugby, Warwickshire, United Kingdom: Practical Action Publishing, 2000.

[5] L. Hružík, M. Vašina, and A. Bureček, "Evaluation of bulk modulus of oil system with hydraulic line," in *EPJ Web of Conferences*, vol. 45. EDP Sciences, 2013, p. 01041.

[6] Bosch-Rexroth, "Axial piston variable motor a6vm - sales information/data sheet," Bosch-Rexroth, Tech. Rep., June 2012.

[7] A. Jarquin Laguna, "Centralized electricity generation in offshore wind farms using hydraulic networks," Ph.D. dissertation, Delft University of Technology, 2017.

[8] K. Mulleners and M. Raffel, "Static versus dynamic stall development," in *APS Meeting Abstracts*, Nov. 2012, p. L24.008.

[9] N. Diepeveen, "On the application of fluid power transmission in offshore wind turbines," Ph.D. dissertation, Delft University of Technology, 2013.

[10] S. P. Mulders, N. F. B. Diepeveen, and J.-W. van Wingerden, "Data set: Control design, implementation and evaluation for an in-field 500 kW wind turbine with a fixed-displacement hydraulic drivetrain," May 2018. [Online]. Available: https://doi.org/10.5281/zenodo.1250459

---

## Author Comment (AC2) · 27 Jul 2018

[revised manuscript text omitted]

where $J_{\mathrm{r}}$ is the rotor inertia, $\omega_{\mathrm{r}}$ the rotor rotational speed, $\tau_{\mathrm{r}}$ the mechanical torque supplied by the rotor to the low-speed shaft, and $\tau_{\mathrm{sys}}$ the system torque supplied by the hydraulic oil pump to the shaft. The rotor inertia $J_{\mathrm{r}}$ of the rotor is not publicly available. However, an estimation of the rotor inertia is obtained using an empiric relation on blade length given in (Rodriguez et al., 2007), resulting in a value of  $6.6 \cdot 10^5$ kg m$^2$. Moreover, experiments were performed on the actual turbine and confirm this theoretical result (Jager, 2017). The torque supplied by the rotor (Bianchi et al., 2006) is given by

$$\tau_{\mathrm{r}} = \frac{1}{2}\rho_{\mathrm{air}}\pi R^3 U^2 C_{\mathrm{p}}(\lambda,\beta)/\lambda, \tag{2}$$

where the density of air $\rho_{\mathrm{air}}$ is taken as a constant value of  $1.225$ kg m$^{-3}$, $U$ is the velocity of the upstream wind, and $R$ is the blade length of  $22$ m. The power coefficient $C_{\mathrm{p}}$ represents the fraction between the captured rotor power $P_{\mathrm{r}}$ and the

[Figure]

**Figure 6.** Rotor power and torque coefficient curve of the rotor, obtained from a BEM analysis performed on measured blade-geometry data. The maximum power coefficient $C_{\mathrm{p,max}}$ of 0.48 is attained at a tip-speed ratio of 7.8. The maximum torque coefficient of $C_{\tau,\mathrm{max}}$ is given by $7.2 \cdot 10^{-2}$ at a lower tip-speed ratio of 5.9.

available wind power $P_{\mathrm{wind}}$, and is a function of the blade pitch angle $\beta$ and the dimensionless tip-speed ratio $\lambda$ given by

$$\lambda = \omega_{\mathrm{r}} R / U. \tag{3}$$

The power coefficient $C_{\mathrm{p}}$ is related to the torque coefficient

5 $C_\tau(\lambda, \beta) \quad = C_{\mathrm{p}}(\lambda, \beta)/\lambda,$

by $C_\tau(\lambda, \beta) = C_{\mathrm{p}}(\lambda, \beta)/\lambda$ such that Eq. (2) can be rewritten as

$$\tau_{\mathrm{r}} = \frac{1}{2}\rho_{\mathrm{air}}\pi R^3 U^2 C_\tau(\lambda, \beta). \tag{4}$$

The rotor power and torque extraction capabilities from the wind are characterized in respective power and torque coefficient curves. These curves of the actual DOT500 rotor are generated by mapping the actual blade airfoils, and applying Blade

10 Element Momentum (BEM) theory (Burton et al., 2011), and are given in Figure 6 at the blade fine-pitch angle. The fine-pitch angle $\beta_0$ indicates the blade angle resulting in maximum rotor power extraction in the below-rated operating region (Bossanyi, 2000). The theoretical maximum rotor power and torque coefficients equal $C_{\mathrm{p,max}} = 0.48$ and $C_{\tau,\mathrm{max}} = 7.2 \cdot 10^{-2}$, at tip-speed ratios of 7.8 and 5.9, respectively. The complete power, torque and thrust coefficient data set is available as an external supplement under (Mulders et al., 2018b).

The system torque $\tau_{\mathrm{sys}}$ is supplied by the hydraulic drivetrain to the rotor low-speed shaft. This torque is influenced by the components in the drivetrain, which all have their own energy conversion characteristics expressed in efficiency curves. All

[Figure]

**Figure 7.** Flow diagram of the DOT500 hydraulic drivetrain. For steady-state modeling purposes, first the flow path is calculated up to the spear valve. The effective nozzle area and the water flow through the spear valve determine the hydraulic feed line pressure, which influences the system torque $\tau_{\text{sys}}$ to the rotor.

components are off-the-shelf and their combined efficiency characteristics influence the operating behavior of the turbine.

5     Hydraulic components are known for their high torque-to-inertia ratio,  and have high acceleration capabilities  as a result (Merritt, 1967). In typical applications of a hydraulic transmission, the fairly low rotational inertia of pumps and motors is still relevant. However, the considered wind turbine drivetrain is driven by a rotor with a large inertia $J_r$

10     compared to the drivetrain components. Referring to the specification sheet of the oil motor (Bosch-Rexroth, 2012), it is stated that the unit has a moment of inertia of $J_b = 0.55\,\text{kg m}^2$. The resulting reflected inertia to the rotor of $J_{b\rightarrow r} = 0.55/G^2 = 1533.312\,\text{kg m}^2$ is still negligible, where $G^{-1}$ represents the *hydraulic gear ratio* of 52.8. Furthermore, a particular study on this aspect has been carried out in (Kempenaar, 2012), where it is concluded that inclusion of component dynamics does not result in significantly improved model accuracy. For the reasons mentioned, the pumps and motor included

15     in the drivetrain are assumed to have negligible dynamics, and the power conversion (flow-speed, torque-pressure) is given by static relations.

**3.1.2   Analytic drivetrain components description**

A flow diagram of the modeling strategy is presented in Figure 7. To obtain an expression for the system torque $\tau_{\text{sys}}$, the complete hydraulic flow path with its volumetric losses is modeled first. When the flow path reaches the spear valve at the

20     water discharge to the Pelton turbine, the simulation path is reversed to calculate the effect of all component characteristics to the line pressures. The spear valve allows for control of the water discharge pressure, of which the effect propagates back to the system torque $\tau_{\text{sys}}$. The high-pressure oil flow by the oil pump is proportional to the rotor speed

$$Q_{\text{o}} = V_{\text{p,h}}\omega_{\text{r}}\eta_{\text{v,h}}, \tag{5}$$

[revised manuscript text omitted]

$$\Delta p = L_{\text{H}}\dot{Q} + R_{\text{H}}Q + \frac{1}{C_{\text{H}}}\int Q dt, \tag{15}$$

where $L_{\text{H}}$, $R_{\text{H}}$ and $C_{\text{H}}$ are the hydraulic induction, resistance and capacitance (Esposito, 1969), respectively, and are defined in Appendix A. The

$$Re = \frac{D_{\text{l}}v\rho}{\mu},$$

$$0 = \dot{Q} + \frac{R_{\text{H}}}{L_{\text{H}}}Q + \frac{1}{C_{\text{H}}L_{\text{H}}}\int Q dt \tag{16}$$

$$= \dot{Q} + 2\zeta\omega_{\text{n}}Q + \omega_{\text{n}}^2\int Q dt. \tag{17}$$

$$\omega_{\text{n}} = \sqrt{\frac{1}{C_{\text{H}}L_{\text{H}}}},$$

$$\zeta_{\text{p}} = \frac{R_{\text{H}}}{2}\sqrt{\frac{C_{\text{H}}}{L_{\text{H}}}}.$$

 inverse result of Eq. (15) is obtained (Murrenhoff, 2012)  with flow $Q$ as the external excitation and $\Delta p$ as output

$$Q = C_{\text{H}}\Delta\dot{p} + \frac{1}{R_{\text{H}}}\Delta p + \frac{1}{L_{\text{H}}}\int \Delta p dt. \tag{18}$$

$$0 = C_{\text{H}}\Delta\dot{p} + \frac{1}{R_{\text{H}}}\Delta p + \frac{1}{L_{\text{H}}}\int \Delta p dt \tag{19}$$

$$= \Delta\dot{p} + \frac{1}{R_{\text{H}}C_{\text{H}}}\Delta p + \frac{1}{L_{\text{H}}C_{\text{H}}}\int \Delta p dt, \tag{20}$$

and using Eq. , it is seen that the natural frequency remains unchanged with the result obtained in Eq. , but the definition of the damping ratio changes

$$\zeta_\mathrm{Q} = \frac{1}{2R_\mathrm{H}}\sqrt{\frac{L_\mathrm{H}}{C_\mathrm{H}}}.$$

Finally, the differential equation defined by Eq. (15) is expressed as a transfer function in

$$G_{\mathrm{Q}/\Delta_\mathrm{P}}(s) = \frac{1/L_\mathrm{H}}{s + (R_\mathrm{H}/L_\mathrm{H}) + 1/(C_\mathrm{H}L_\mathrm{H}s)} \equiv \frac{s/L_\mathrm{H}}{s^2 + (R_\mathrm{H}/L_\mathrm{H})s + 1/(C_\mathrm{H}L_\mathrm{H})}, \tag{21}$$

and the same is done for Eq. (18)

$$G_{\Delta_\mathrm{P}/\mathrm{Q}}(s) = \frac{1/C_\mathrm{H}}{s + 1/(R_\mathrm{H}C_\mathrm{H}) + 1/(C_\mathrm{H}L_\mathrm{H}s)} \equiv \frac{s/C_\mathrm{H}}{s^2 + 1/(R_\mathrm{H}C_\mathrm{H})s + 1/(C_\mathrm{H}L_\mathrm{H})}. \tag{22}$$

The transfer functions defined in  Eqs. (21) and (22) show the characteristics of an inverted notch with $+1$ and $-1$ slopes on the left and right side of the natural frequency, respectively. This physically means that exciting the system pressure results in a volume velocity change predominantly at the system natural frequency for the former mentioned case. An illustrative Bode plot is given in Sect. 4.2.1. Exciting the flow results in amplification/transmission to pressure in a wider frequency region. This effect is a result of the inverse proportionality between the damping coefficients $\zeta_\mathrm{Q}$ and $\zeta_\mathrm{P}$ (see Appendix B).

**3.2.2 Drivetrain model derivation**

A dynamic model of the DOT500 drivetrain is derived by application of the theory presented in the previous section. The drivetrain is defined from the rotor up to the spear valve, and the following assumptions are made:

- Because of the high torque to inertia ratio of hydraulic components (Merritt, 1967), the dynamics of oil pumps and motors are disregarded and taken as analytic relations;
- Because of the longer line length and higher compressibility of oil compared to the shorter water column, the high-pressure oil line is more critical for control design, and a dynamic model  is implemented for this column only;
-
- The fluids have a constant temperature.

The dynamic system is governed by the following differential equations

$$\mathcal{V} = \mathcal{V}_\mathrm{in} - \mathcal{V}_\mathrm{out}, \qquad \dot{\mathcal{V}} = Q = Q_\mathrm{in} - Q_\mathrm{out}, \tag{23}$$

$$\Delta p_\mathrm{h} = \underbrace{\left(\frac{J_\mathrm{r}\eta_\mathrm{m,h}}{V_\mathrm{p,h}^2\eta_\mathrm{v,h}} + L_\mathrm{H}\right)}_{L_\mathrm{R}^*}\dot{Q}_\mathrm{in} + R_\mathrm{H}(Q_\mathrm{in} - Q_\mathrm{out}) + \frac{K_\mathrm{f}}{V_\mathrm{H}}\frac{K_\mathrm{e}}{V_\mathrm{H}}(\mathcal{V}_\mathrm{in} - \mathcal{V}_\mathrm{out}) = L_\mathrm{R}^*\dot{Q}_\mathrm{in} + R_\mathrm{H}(Q_\mathrm{in} - Q_\mathrm{out}) + \frac{1}{C_\mathrm{H}}\mathcal{V}, \tag{24}$$

$$\Delta p_\mathrm{b} = L_\mathrm{H}\dot{Q}_\mathrm{out} + R_\mathrm{H}(Q_\mathrm{out} - Q_\mathrm{in}) + \frac{K_\mathrm{f}}{V_\mathrm{H}}\frac{K_\mathrm{e}}{V_\mathrm{H}}(\mathcal{V}_\mathrm{out} - \mathcal{V}_\mathrm{in}) = L_\mathrm{H}\dot{Q}_\mathrm{out} + R_\mathrm{H}(Q_\mathrm{out} - Q_\mathrm{in}) - \frac{1}{C_\mathrm{H}}\mathcal{V}, \tag{25}$$

where $K_e$ is the equivalent bulk modulus including the fluid and line compressibility defined in Eq. (A5), and $\mathcal{V}$ is the net volume inflow to the considered oil line, between the oil pump discharge and oil motor feed port. For convenience, mechanical model quantities are expressed hydraulically in terms of fluid flows and pressure differences over the components. Therefore, the rotor inertia $J_r$ is expressed in terms of fluid induction, and is combined with the hydraulic induction term into $L_R^*$.

5   Both the spear position and pitch angle are modeled by a first-order actuator model

$$\dot{s} = \frac{1}{t_s}(s_{\text{ref}} - s), \tag{26}$$

$$\dot{\beta} = \frac{1}{t_\beta}(\beta_{\text{ref}} - \beta), \tag{27}$$

where $t_s$ and $t_\beta$ are the time constant for the spear valve and pitch actuators, respectively, and the phase loss at the actuator bandwidth is assumed to account for actuation delay effects.

10   The above given dynamic equations are written in a state-space representation as

$$\begin{bmatrix} \dot{\mathcal{V}} \\ \dot{Q}_{\text{in}} \\ \dot{Q}_{\text{out}} \\ \dot{s} \\ \dot{\beta} \end{bmatrix} = \begin{bmatrix} 0 & 1 & -1 & 0 & 0 \\ -\frac{1}{C_H L_R^*} & -\frac{R_H}{L_R^*} & \frac{R_H}{L_R^*} & 0 & 0 \\ \frac{1}{C_H L_H} & \frac{R_H}{L_H} & -\frac{R_H}{L_H} & 0 & 0 \\ 0 & 0 & 0 & -\frac{1}{t_s} & 0 \\ 0 & 0 & 0 & 0 & -\frac{1}{t_\beta} \end{bmatrix} \begin{bmatrix} \mathcal{V} \\ Q_{\text{in}} \\ Q_{\text{out}} \\ s \\ \beta \end{bmatrix} + \begin{bmatrix} 0 \\ \frac{1}{L_R^*}\Delta p_h \\ -\frac{1}{L_H}\Delta p_b \\ \frac{1}{t_s}s_{\text{ref}} \\ \frac{1}{t_\beta}\beta_{\text{ref}} \end{bmatrix}. \tag{28}$$

It is seen that the pressure difference over the oil pump and motor appear as inputs, but these quantities cannot be controlled directly. For this reason, linear expressions of the rotor torque and spear valves are defined next. The rotor torque is linearized with respect to the tip-speed ratio, pitch angle and wind speed

15   $$\hat{\tau}_r(\bar{\omega}_r, \bar{\beta}, \bar{U}) = k_{\omega_r}(\bar{\omega}_r, \bar{\beta}, \bar{U})\hat{\omega}_r + k_\beta(\bar{\omega}_r, \bar{\beta}, \bar{U})\hat{\beta} + k_U(\bar{\omega}_r, \bar{\beta}, \bar{U})\hat{U}, \tag{29}$$

where $(\hat{\cdot})$ indicates a value deviation from the operating point, and $(\bar{\cdot})$ is the value at the operating point (Bianchi et al., 2006). Furthermore,

$$k_{\omega_r}(\omega_r, \beta, U) = \frac{\partial \tau_r}{\partial \omega_r} = c_r R U \frac{\partial C_\tau(\omega_r R/U, \beta)}{\partial \lambda}, \tag{30}$$

$$k_\beta(\omega_r, \beta, U) = \frac{\partial \tau_r}{\partial \beta} = c_r U^2 \frac{\partial C_\tau(\omega_r R/U, \beta)}{\partial \beta}, \tag{31}$$

$$k_U(\omega_r, \beta, U) = \frac{\partial \tau_r}{\partial U} = 2c_r U C_\tau(\omega_r R/U, \beta) + c_r U^2 \frac{\partial C_\tau(\omega_r R/U, \beta)}{\partial \lambda} \frac{\partial \lambda}{\partial U} \tag{32}$$

$$= 2c_r U C_\tau(\omega_r R/U, \beta) - c_r \omega_r R \frac{\partial C_\tau(\omega_r R/U, \beta)}{\partial \lambda}, \tag{33}$$

$$c_r = \frac{1}{2}\rho\pi R^3, \tag{34}$$

where the quantities $k_{\omega_r}$, $k_\beta$ and $k_U$ represent the intrinsic speed feedback gain, the linear pitch gain and the linear wind speed gain, respectively. The intrinsic speed feedback gain can also be expressed as a function of the tip-speed ratio by

$$k_\lambda(\lambda, \beta, U) = k_{\omega_r}(\omega_r, \beta, U)\frac{U}{R}. \tag{35}$$

[Figure]

**Figure 10.** The intrinsic speed feedback gain $k_\lambda(\lambda, \bar{\beta}, \bar{U})$ as function of tip-speed ratio $\lambda$, at a fixed pitch angle and wind speed of $-2$ $-2$ deg and $8$ $8$ m s$^{-1}$. Stable turbine operation is attained for non-positive values of $k_\lambda$.

For aerodynamic rotor stability, the value of $k_\lambda$ needs to be negative. In Figure 10 the intrinsic speed feedback gain $k_\lambda(\lambda, \bar{\beta}, \bar{U})$ is evaluated as a function of the tip-speed ratio at the fine-pitch angle $\beta_0$. For incorporation of the linearized rotor torque in the drivetrain model, Eq. (29) is expressed in the pressure difference over the oil pump

$$\Delta\hat{p}_h(\bar{\omega}_r, \bar{\beta}, \bar{U}) = k_{Q_{in}}^*(\bar{\omega}_r, \bar{\beta}, \bar{U})\hat{Q}_{in} + k_\beta^*(\bar{\omega}_r, \bar{\beta}, \bar{U})\hat{\beta} + k_U^*(\bar{\omega}_r, \bar{\beta}, \bar{U})\hat{U},\tag{36}$$

where the conversions of the required quantities are given by

$$k_{Q_{in}}^* = k_{\omega_r}\frac{\eta_{m,h}}{V_{p,h}^2\eta_{v,h}}, \qquad k_\beta^* = k_\beta\frac{\eta_{m,h}}{V_{p,h}}, \qquad k_U^* = k_U\frac{\eta_{m,h}}{V_{p,h}}.\tag{37}$$

Similarly, the water line pressure as defined in Eq. (10) is linearized with respect to the spear position and flow through the valve

$$\hat{p}_{w,l}(\hat{Q}_w, \hat{s}) = k_{s,s}(\bar{Q}_w, \bar{s})\hat{s} + k_{s,Q_w}(\bar{Q}_w, \bar{s})\hat{Q}_w,\tag{38}$$

where

$$k_{s,s}(\bar{Q}_w, \bar{s}) = \left.\frac{2Q_w^2\rho_w(s - s_{max})\tan^2(\alpha/2)}{C_d^2 N_s^2\pi^2\left(D_{nz}^2/4 - (s_{max} - s)^2\tan^2(\alpha/2)\right)^3}\right|_{\bar{Q}_w, \bar{s}},\tag{39}$$

$$k_{s,Q_w}(\bar{Q}_w, \bar{s}) = \left.\frac{Q_w\rho_w}{C_d^2 N_s^2\pi^2\left(D_{nz}^2/4 - (s_{max} - s)^2\tan^2(\alpha/2)\right)^2}\right|_{\bar{Q}_w, \bar{s}}.\tag{40}$$

[revised manuscript text omitted]

---

## Author Response (AR1)

| Date | July 27, 2018 |
|---|---|
| Our reference | n/a |
| Your reference | n/a |
| Contact person | S.P. Mulders |
| Telephone/fax | +31 (0)6 5573 6149 / n/a |
| E-mail | S.P.Mulders@TUDelft.nl |
| Subject | Response to reviewers |

**Delft University of Technology**

Delft Center for Systems and Control

Address
Mekelweg 2 (3ME building)
2628 CD Delft
The Netherlands

www.dcsc.tudelft.nl

Reviewers
*Wind Energy Science*

Dear Reviewers,

First of all, the authors would like to thank the reviewers for their positive and constructive feedback. We believe that the comments help us to significantly improve the quality of the paper. The objective of this document is to respond to the points raised by the reviewers (blue) and to provide an overview of the actions that are taken (red). When in this response the authors refer to adjustments in a particular section, figure or table, we ask the reviewer to refer to the marked-up manuscript version to evaluate the changes.

The document consists of five sections, each addressing the comments of the reviewers separately.

Yours sincerely,

Sebastiaan Paul Mulders
Niels Frederik Boudewijn Diepeveen
Jan-Willem van Wingerden

| Enclosure(s): | Response to comments of Reviewer 1 |
|---|---|
| | Response to comments of Reviewer 2 |
| | Response to comments of Reviewer 3 |
| | Response to comments of Reviewer 4 |
| | Response to comments of Reviewer 5 |

**Response to comments of Anonymous Referee #1**

**Reviewer 1 comments**: This paper presents an interesting analysis of a novel hydraulic wind turbine concept. The concept consists on replacing the conventional mechanical drivetrain components with a seawater pump directly driven by the wind turbine, whose outlet flow is directed to a Pelton generator. However, the analysis presented in the paper refers to an intermediate solution, in which the seawater pump is driven by a close loop oil-based hydrostatic transmission. The paper topic is certainly relevant for the journal; the approach is rigorous and the authors appear to be very familiar and qualified for work in the field. However, the paper could be significantly improved in certain aspects. Therefore, this Reviewer recommend its publication only after major changes are implemented to the submitted manuscript.

The authors thank the reviewer for his/her thorough review, invaluable comments and remarks. The considerations are especially informative as in particular considerations are raised on the analysis and justification of the hydraulic drivetrain and its components. Processing these comments very much helped the authors in closing the gap between system design and mathematical evaluation of the employed turbine with hydraulic transmission.

1. [Major] The paper is quite long, it contains too many equations and figures. This Reviewer suggests the authors to reduce the number of equations and figures. Some suggestions are provided in the following comments. Consider also that this Reviewers is asking for some additional details, therefore some additional figures might be necessary in the revised version of the paper.

   The authors agree with the fact that this paper is quite long. In the revised version, this issue is addressed by revising the relevance of all content. We are grateful of the reviewer suggestions, and the authors will take these remarks into account.

   The most notable changes are noted:
   - Section 2.2 is extended to provide a more detailed description of the DOT500 prototype set-up;
   - Section 3.2.1 is shortened, now only showing the results. The derivation is moved to the appendix;
   - Section 4.2.2 is shortened.

2. [Minor] Not sure about the significance of fig. 2, since the concept is quite obvious. In any case, if the authors decide to keep this image, this Reviewer suggests to include labels for the different components represented

Thank you for this comment. The authors agree that this figure is not of direct scientific relevance, however, it presents a nice comparison of the (potential) space-advantage of a wind turbine with hydraulic drivetrain.

The authors decided to keep the figure, but included a description of the distinct components, as suggested by the reviewer.

3. [Major] The authors provide a quite exhaustive overview of the past effort, which is very appreciated. However, at pag. 2, they affirm "To date, none of the above described full hydraulic concepts made its way to a commercial product. All concepts use oil as the hydraulic medium because of the favorable fluid properties and wide component availability, but therefore also need to operate in closed-loop." This Reviewer has two problems with such sentence:

- For the size of the components required for wind turbine applications, there are almost no available commercial products. Those chosen by the authors in their work are probably the among the very few ones available (considering also that they had to turn a motor into a pump!). This is because as the authors stated, there are no successful application for hydraulic wind turbines. Therefore, if there is no market (thus no demand), there is no offer. The message is that nowadays someone wants to design a hydraulic wind turbine, he/she necessitates to design the hydraulic components as well (or partner with a component manufacturer to get a unit specially designed).

The authors agree with the reviewer on this point. DOT is founded with the philosophy that offshore wind can be simplified and exploited more efficiently by centralizing energy production. However, to date, a water pump with capabilities to operate under high load and low speed is not commercially available. DOT is very aware of this, and is therefore (1) working to develop a seawater pump by their selves, and (2) actively cooperating with renowned (water) pump manufacturers on the development of such a pump.

The in-field tests with an intermediate drivetrain including a closed oil circuit, is the first step towards the final DOT concept. The goal of DOT therefore is to abandon the use of oil all together by making wind turbines to cooperate and use what is abundantly available offshore: seawater. The authors (and DOT) do realize that a lot of hurdles need to be overcome before this goal becomes reality.

- The reference to close loop (close circuit ?) hydraulic transmissions (HTs) is questionable. HTs for many mobile applications (wheel loaders, excavators, etc) are close loop, but again the components for these HTs are too small for wind turbine applications. HTs can also be open-loop. Many HTs for areal platforms, forklifts, hydraulic fan drives are open-loop. What are the requirements that determine the need of having a close loop HT for a wind turbine application? This Reviewer can have some guesses, but this should be better addressed.

  The point raised by the reviewer is valid. A closed circuit drivetrain for a wind turbine utilizing oil as the hydraulic medium is needed because:
  (1) Operating a turbine with oil at remote offshore locations might pollute the environment in the event of a calamity;
  (2) Not having to provide a continuous fresh oil supply to the circuit;

  A consideration for operating in closed circuit is cooling of the hydraulic medium. When losses in hydraulic components are significant and the natural convection of heat to the surroundings is insufficient for cooling, an additional cooling circuit needs to be incorporated.

  The authors recognize that the statement is posed too strong and lacks further explanation. Therefore, the comments of the reviewer and our considerations are processed in the revised version of the introduction. Furthermore, we made sure that we reserved the term "circuit" for hydraulic matters, and "loop" for control purposes.

4. [Minor] Was the concept of the paper presented also at the IFK2018 conference? A better reference to that paper, and the novel contents of this paper, should be provided.

   Yes, this is correct. We did not yet include a reference to the conference paper, as it was not published at time of submission of the WES manuscript.

   The authors included a reference to the conference article in the introduction, and stated clearly what the contribution of this paper is as opposed to the conference article.

5. [Minor] figure 3 might not be necessary, can be removed.

Thank you for this comment. Because of the length and increased complexity of this paper we included a paper organization flow chart. However, providing both a textual and graphical outline might indeed be redundant.

The figure is removed from the manuscript.

6. [Major] Section 2.2. The hydraulic circuit needs to be better detailed. A more realistic ISO schematic with respect to the one provided in fig. 6 is needed. The authors give the impression that a pump can be simply be coupled with a motor to form a close loop HT. However, other components are needed to guarantee the operation of the system:

The authors agree with the reviewer that the hydraulic diagram should be better detailed. The included simplified hydraulic diagram was a trade-off between complexity and relevance for the drivetrain modeling provided in the manuscript. In the real-world set-up, numerous additional components were in place for turbine operation.

Reconsidering the performed trade-off, the authors revisited the hydraulic diagram by including vital components. The components that are not included/considered for modeling and analysis are presented in gray. In this way, the authors think that the updated diagram provides a middle course from a system design and theoretical modeling point of view.

- Is a charge pump present? How was that sized? Can be neglected in the analysis? Why? What is the pressure level of the low pressure line? Is a flushing valve / cooling of the hydraulic motor present / necessary for the long operation of the system? A HTs for continuous operation usually necessitates for a significant oversizing of the charge unit for cooling purposes.

Correct, charge pumps are presents in both the oil and water circuits. The oil charge pump was sized in such a way that a sufficient flow with a constant (controlled) feed-pressure of 21 bar to the oil pump could be delivered. Additionally when cooling was required, the charge pump supplied more flow to be directed through the parallel cooling circuit connected by a pressure relief valve (see updated hydraulic diagram). For the water circuit, the centrifugal charge pump provided a lower charge pressure of 2.6 bar. This difference in charge pressure is due to the different pump types used: the radial piston oil pump (motor) requires a higher charge pressure, as this is used to actively push the piston bearings to the cam ring; whereas the water plunger pump largely alleviates this requirement.

As for modeling purposes of the closed oil circuit only pressure differences over the hydraulic components (taking into account mechanical/volumetric efficiency losses) are considered, the feed pressure is left out from the analysis. For the open water circuit however, the feed pressure is neglected because of its low value and for convenient derivation of the passive torque control strategy.

Cooling equipment was indeed necessary for the closed oil circuit, and a flush oil cooling circuit was in place to ensure long-term operation and prevent the working fluid from overheating.

The updated manuscript now provides a more detailed description of the aspects discussed above in Section 2.2.

- Circuit of the water pump. The authors say that there is an external centrifugal pump, which seems to be connected in series with the fixed displacement pump. How is the schematic? Is there a relief valve in between to provide a reference pressure level? Why this part can be neglected in the subsequent analysis?

  The reviewer is correct, the centrifugal pump is connected in series to the water pump, and the updated hydraulic diagram presents the working principle. To prevent a disturbed flow entering the water pump, a two-reservoir set-up is used. The speed of the centrifugal pump is controlled to maintain feed pressure of approximately $2.6$ bars. The charge pump is enabled before the water pump starts speeding up to ensure feed-pressure and thus to avoid cavitation. The low-pressure side of the water circuit does not contain a pressure-relief valve, as the water plunger pump allows for a direct feed-though of the flow; the high-pressure side however does include a pressure relief valve.

  Section 2.2 is updated with the details provided in the response to the reviewer.

- Pelton Turbine. The concept of using a Pelton Turbine is very interesting. However, it seems that the Head [m] of this turbine is way above to the existing Pelton turbines, so that it might be impossible to borrow an existing design. What is the specific speed of the Pelton Turbine of this paper? Is a commercial Pelton wheel available? Is a two-jet turbine such as the one of fig. 5 sufficient? This is not the scope of this paper, however, the authors could be more clear on this part, perhaps using more references.

  Thank you for raising this comment, we could have been more clear on this aspect. Pelton turbines are highly specialized pieces of equipment and need to be design for specific condition requirements [1]. The Sy Sima $315$ MW turbine in Norway, for $88.5$ bar of head pressure is to date the largest known [2]. The employed custom manufactured Pelton turbine for the DOT500 is designed to match the nominal pressure and the speed conditions of the connected electrical generator.

  A custom-made Pelton turbine is designed such that the efficiency is optimal under the expected operational conditions. For this, the turbine is designed for optimal operation using 2 spear valves, subject to a nominal flow of 58 l/min. Graphs of the turbine manufacturer (given below) show that the efficiency is primarily a function of the supplied flow, and to a lesser extent of the head. The red line indicates operation with 1 spear valve, the blue line 2 spear valves.

[Figure]

The efficiency aspect is confirmed later by experiments executed by DOT, of which an efficiency evaluation figure is also given below. In the figure an evaluation of the combined spear valve-Pelton efficiency from hydrostatic fluid to mechanical power at the generator axis is given as a function of flow and pressure. Whereas during the experiment the flow was not sufficient to explore the overall characteristics, the results clearly show that the steepest partial efficiency gradient goes with flow; at higher flow rates the gradient with respect to pressure becomes negligible.

[Figure]

In the updated manuscript, more information on the custom-design Pelton turbine is given. Also the nominal operating conditions are discussed, and relevant references are included.

7. [Major] At page 7, the authors say After the water flow exits the spear valve, the aim to operate the Pelton turbine generator combination at maximum efficiency is a decoupled control objective from the rest of the drivetrain, and is outside the scope of this paper. Actually, this sentence is at the basis of many assumptions made in the development of the model and the controller design. This Reviewer, although without specific experience in designing HTs for wind turbine applications, has some conceptual doubts on this choice made by the authors. A HT has to be designed according to the features of both the load and the prime mover; this also drives several choices of constant torque (variable displacement pump) or constant power (variable displacement motor) HTs. In this case, the authors decide to neglect the features of the user (the Pelton wheel). Is that correct? To this Reviewer, it is like affirming that all the points that satisfies Eq. 12 (relation nozzle area and HT pressure) are indifferent for the Pelton turbine. This is quite hard to believe. The Pelton turbine should have preferred operating points that the HTs should be able to handle. This is a very basic question that the authors should address properly. Otherwise their proposed controller might not be beneficial on a real application.

The reviewer correctly points out that the system design, as well as the applied control strategy should go hand-in-hand. Changing the operational strategy on the wind turbine side affects the operating point of the Pelton turbine and thereby for example the maximum amount of energy it can extract from the given flow. This is completely understood by the authors.

The features of the Pelton wheel are not neglected. It is known from literature [3][4] that the ratio of tangential Pelton and water jet speed needs to be maintained at approximately $1/2$. As the Pelton wheel is mechanically coupled to an asynchronous generator, which can change its operational speed, a pressure measurement is used to determine the most favorable (speed) operating point to be as efficient as possible, given the conditions it is subjected to. However, the Pelton wheel will in the given set-up always be subjected to varying conditions, and thus suboptimal operation in the considered drivetrain using fixed-displacement components. This is for now a design choice, and further research needs to be conducted to elaborate on Pelton design and efficiency maximization given the varying operational conditions.

The point of which operational path is most efficient, given varying Pelton conditions, remains. Operation at $C_{\tau,\mathrm{max}}$ will result in higher pressures for equal flows when compared to $C_{\mathrm{p,max}}$ operation. As was concluded in the previous question, the main driver determining the Pelton efficiency is the flow it is subjected to, whereas the head has negligible influence.

The above given considerations are included in the manuscript in Section 2.2.

8. [Major] Section 3.1.2. The authors here affirm "the volumetric efficiency of a pump or motor is generally high and fairly constant over the entire operating range". For a simplified model the assumption of constant efficiency could be a fair starting point. But the statement that hydraulic pumps and motors have a constant volumetric efficiency for any pressure and shaft speed is clearly wrong. Otherwise, all the literature on empirical efficiency models (starting from Merritt in the 60s), standards for measuring volumetric efficiency (ISO, etc), tribological models for studying the lubricating gap flows, would not be justified. Particularly at low speed, the volumetric efficiency can be particularly low for both pumps and motors. Please revise this statement and better justify the assumption of constant efficiency, which can be very limited.

The reviewer makes a very valid point, and we agree with it. The reason we have chosen to assume a constant volumetric efficiency factor is (1) the fact that for most of the given components, no volumetric efficiency data is available, and (2) the aim is to provide a simplified model of the hydraulic drivetrain.

The assumption of a volumetric efficiency is revised in the updated manuscript.

9. [Minor] Section 3.2.1. here there are several equations that are well known. This section could be reduced.

Thank you for this comment. The authors recognize that Section 3.2.1 is lengthy and reduced it. Now, only the major results are presented. However, we would like to be as complete as possible, seen the journal we are publishing in is not primarily focussed on hydraulics. Therefore, we moved the derivation of the results to the appendix.

Section 3.2.1 is shortened and the derivation of the results has been moved to the appendix.

10. [Major] Section 3.2.2. In the list of assumptions it is stated that the inertia of the hydraulic components is neglected. While it is true that hydraulic components have fast dynamic, in comparison with other technologies for transmitting power, it has to be proven that within a hydraulic system the hydraulic line is the element with slowest dynamic. This statement, in general is not true, and Merritt never affirmed that. Moreover, the authors consider infinitely rigid lines, therefore the fastest lines possible (is this realistic?). Please justify this statement.

Thank you for this comment, the authors agree with the reviewer that the (slow) line dynamics cannot be discarded. As exact specifications of the line bulk modulus are unknown (not publicly available), we decided to take a reasonable value of $K_l = 0.8$ GPa from [5], which is twice as low as the bulk modulus value taken for the oil column. The equivalent bulk modulus is calculated by the relation $K_e = (1/K_f + 1/K_l)^{-1} = 0.52$ GPa. The equivalent value is used subsequently in the remainder of the paper.

The assumption of infinitely rigid lines is removed from the analysis in Section 3.2.2, and Appendix A is updated to include relation for calculating the equivalent bulk modulus. The resulting equivalent modulus is subsequently used in the analyses throughout the different sections of the paper. Furthermore, Section 3.1.1 of the manuscript is updated, and now includes a more elaborate justification on why the drivetrain component dynamics are neglected and assumed as analytic expressions.

11. [Major] Section 4.1.1. The authors say "hydraulic components are known to be more efficient in high-load operating conditions, it might be advantageous for a hydraulic drivetrain to operate the rotor at a lower tip-speed ratio". This statement can be arguable. first, shaft speed has a major effect, and not all units have a clear trend with load. Can the authors provide the overall efficiency plot for the commercial units they utilize (even in normalized form?). This is very important, because all the controllers of case 1 and case 2 are based on this assumption!

The authors agree with the comment made by the reviewer, and the statement the reviewer refers to is changed. However, the manuscript already includes (mechanical) efficiency data for the oil pump and motor, given in Figure 13. The figure includes the proposed operational strategies (case 1 and case 2), and a steady-state analysis of the total drivetrain efficiency for both strategies is given in Figure 14.

The titles of the plots in Figure 13 and the legend in Figure 14 are now updated, to make their purpose more clear. Also, the statement the reviewer refers to is adjusted, and an additional consideration on the efficiency aspect is made in Section 4.1.2.

12. [Minor] Pag. 21. The reference to fig 14 might be wrong, since the figure refers to mechanical efficiency.

    Thanks for pointing out this mistake. The reference should be pointing to Fig. 13.

    The reference is corrected in the revised manuscript.

13. [Major] 4.2.1. L=50m... are the pump and motor connected by a 50 m straight line? If there are line discontinuities, some terms, particularly the inductance terms, can be entirely wrong

    Indeed, apart from a swivel which enables continuous yaw motion located below the nacelle, the high pressure lines have no discontinuities, and run from the nacelle all the way to the oil motor located in the monopile. Furthermore, the rotor inertia, which is expressed in terms of hydraulic induction, is predominant in the lumped induction term. The contribution of the hydraulic inductance term is thus negligible, and discrepancies would have a negligible effect on the analysis.

**Response to comments of Anonymous Referee #2**

**Reviewer 2 comments**: Thank you for your submission. In general I found the paper to be interesting and well-presented. Background literature was complete and informative, and the introductory material explains well what this paper adds to the growing literature. The figures and illustrations are particularly well done and helpful. The writing is clear, with only very few grammatical errors. The paper is well structured, such that one new to hydraulic drivetrain wind can follow. Finally, the inclusion of field results and comparing to the theoretical work is informative. Excellent work.

Thank you, the authors are pleased to read this!

Therefore, only a few comments to be given, overall:

1. Is there provided, or could the authors provide, a quick impression of how the efficiency of the proposed system, in total, would compare to a similar conventional system? For example, given the same rotors, what would a standard efficiency to final electrical power be (90%?) and what would it be for this system?

   For the described DOT prototype, the total power transmission efficiency was predictably low, as a result of the double hydraulic circuit. In the below-rated region an efficiency is attained of $30 - 45\%$ depending on the operating conditions, whereas in the above-rated region a consistent drivetrain efficiency of $45\%$ is attained. As described in the manuscript, off-the-shelf components are used, of which the optimal efficiency operational envelopes do not match. The drivetrain has a fixed-volumetric displacement, which means that the pressure and flow changes according to the turbine operating conditions. Figure 13 shows that drivetrain components have a specific region in which they yield maximum efficiency.

   The above given reasoning holds for the described prototype. It is however yet unclear what the drivetrain efficiency of the final DOT concept will be, as the seawater pump is still under development. An earlier PhD thesis on hydraulic wind turbine networks [7] provides an estimate on the overall conversion efficiency of conventional and hydraulic wind turbines of $82 - 84\%$ and $70 - 80\%$, respectively.

   The efficiency numbers attained with the intermediate DOT500 prototype are added to Section 2.1.

   Specific:

2. 3.2.1: "and vice versa for the latter.." I could not fully understand what is meant by this. Could a Bode plot of the transfer functions be included to visualize the inverted notch functions?

The authors did separate the theory from the results. For this reason, in Section 3.2.1, only theory is provided, whereas in Section 4.2.1 an illustrative example is given, which considers the system and hydraulic properties of the DOT500 system. In the latter mentioned section, a Bode plot of the of the transfer function $G_{Q/\Delta p}(s)$ is given. For clarity reasons (and considering the length of the paper), a visualization of $G_{\Delta p/Q}(s)$ is omitted in the manuscript, but given in the figure below. It is shown that the inverted notch characteristics is still present. However, exciting the flow (instead of pressure) results in amplification/transmission to pressure in a wider frequency region for shorter line lengths. For longer line lengths, the amplification magnitude increases, but at a more specific interval. This effect is a result of the inverse proportionality between the damping coefficients $\zeta_Q$ and $\zeta_p$.

[Figure]

In Section 3.2.1 the phrase "and vice versa for the latter.." is removed and replaced with a more convenient description. The paragraph now also references to the illustrative example in Figure 15.

3. Eq 38: The B-matrix in this version includes the inputs? There is a dot following the matrix but it is not clear what the dot product will be with? In eq 54 there are only 3 columns total for the 3 B matrices, but are there 4 variables provided in this version?

The representation given by Eq. 38 is the rewritten form of the dynamic system derived in in Eqs. (33)-(37). There is no dot-product after the input vector, this is just punctuation to indicate the end of the sentence. The pressures $\Delta p_\mathrm{h}$ and $\Delta p_\mathrm{b}$ cannot be controlled directly. For this reason, the rotor torque and spear valve pressure characteristics are evaluated and linearized at different operating points. By doing so, a linear state space system defined in Eq. (54) is obtained. By substitution of the linearized characteristics, defined in Eqs. (46) and (51), the terms redistribute in the $A$, $B$ and $B_\mathrm{U}$ matrices: some are defined in the state vectors, others can be regarded as control inputs or wind disturbance inputs in $B$ and $B_\mathrm{U}$.

The authors hope to have clarified the unclarities, and slightly updated the section to improve readability.

4. Section 4.1.1: "advantageous for ... operate a lower tip-speed ratio", this is counterintuitive, but do I understand correct that although the rotor power will be reduced, the improved hydraulic efficiency will lead to higher final electrical power? Is this demonstrated conclusively?

The reviewer is correct. Normal wind turbines operate the rotor at the maximum power coefficient, maximizing the efficiency in the below-rated region. For the DOT500, however, the wind turbine drivetrain is retrofitted, while retaining the original turbine rotor. As hydraulic components are in general more efficient in high-load operating conditions, we additionally perform an analysis for operating the turbine at the maximum possible torque coefficient. The maximum torque coefficient is located at a lower tip-speed ratio (lower rotor speeds, higher torques for equal wind speeds), and corresponds with a lower power coefficient.

So indeed, from a aerodynamic efficiency perspective this is unfavorable, but from a hydraulic drivetrain perspective this might result in an overall efficiency advantage. In the subsequent section, an efficiency analysis is given on a component level for both operating cases in Figure 13, and an overall evaluation of the drivetrain efficiency is presented by Figure 14. The analysis takes into account the reduced rotor power coefficient for operation at maximum torque.

The titles of the plots in Figure 13 and the legend in Figure 14 are now updated, to make their purpose more clear. Also the introductory paragraph of Section 4.1.1 is updated, and a concluding remark referring to the next section where the actual efficiency analysis is performed is added.

5. Section 4.1.2: Does the lower TSR also risk increased occurrence of dynamic stall?

   Thank you, this is a very good question. We cite the following phrase from [8]:

   "Stall on lifting surfaces is commonly encountered, mostly undesired, and occurs when a critical angle of attack is exceeded. Depending on the unsteady rate of change of the airfoil's angle of attack, static and dynamic stall are distinguished. (...) During dynamic stall, the shear layer rolls up into a large scale dynamic stall vortex which grows locally and temporally until vortex induced separation occurs. During static stall on the other hand, the shear layer rolls up continuously into large-scale structures that grow spatially."

   So indeed, there is an increased occurrence of dynamic stall, especially in turbulent wind conditions when the angle of attack continuously varies.

   Furthermore, from a discussion with a professor in aerodynamics from our faculty, it became clear that (dynamic) stall could indeed occur in the region of the blade root. Stalling of a larger blade would result in increased loading with a reduced power capture, however, as we are not stalling during normal operation, the effects on loads should be minor. He also clarified that dynamic stalling could even be slightly beneficial, as it introduces a dissipating/damping effect.

   We have to admit that we did not perform a detailed analysis on this aspect. The aim of the in-field test was to show the feasibility of the hydraulic drivetrain, and while we ensured safe operation of the turbine, effects such as dynamic stall were disregarded. The authors have noted the comment, and the effects of (dynamic) stall will be considered in later stages of the project.

6. Fig 16: Title is incomplete

   Correct, thank you for pointing out. We referred back to the submitted manuscript, but there the title is correct. Somehow, processing of the manuscript during upload must have changed the title by accident.

7. Fig 18: The legend is hard to understand, why the lower-case bold "without" following the period? Since the phase margin is discussed later, could it be indicated in this figure?

   Thank you for pointing out this mistake, the point should be a comma. The suggestion of indicating the phase margin is taken, and both bode plots are updated.

Also, again, the upload process changed the figure, by omitting some symbols. The correct figure is shown below for reference.

[Figure]

The mistake in the caption is fixed, and the Bode plots include an indication of the phase margin (PM).

8. Fig 22: Color legend missing label

The point raised by the reviewer is not entirely clear for the authors. Figure 21 is a representation of the tip-speed ratio for a range of turbine operating points. This is also stated title and the caption of the figure. Adding an additional label to the color legend would be redundant in our opinion.

9. Fig 24: Good figure, just wish to confirm, is rotor speed scaled correctly? Does it always stay so close to its maximum? Or is this period special in that there is only a brief excursion into region 2 (which makes sense, youve selected a period covering 2,2.5,3) just want to be sure.

   Thank you for this comment. Yes, the rotor speed is scaled correctly, however, we took a part of the time-series where the environmental conditions were such that the turbine operated around region 2.5. What we want to show in this figure is how the spear valve torque controller (only active in region 2.5) works as expected and switches nicely to region 2 (no control), and region 3 (pitch control).

   The text is adjusted slightly such that the purpose of the figure is more clear.

   An interesting, and well-presented paper.

   Thanks again!

**Response to comments of Anonymous Referee #3**

Paper is very well written and subject matter thoroughly presented.

Thank you, the authors are grateful to hear this.

Some general comments: towards the beginning of the paper it is stated that "To date, none of the above described full hydraulic concepts made its way to a commercial product", this merits some justification as to why the concept of hydraulic transmission for wind turbines has not been commercially viable so far, and whether the presented work can potentially overcome these barriers to market.

The authors agree with the reviewer, and a similar points has been posed by Referee #1. We would like to refer the reviewer to the answer given in Question 3-2 in our response to the first referee.

We further elaborated on this point in the introduction of the manuscript according to the comments of both referees.

One minor comment would be to add some labelling of the components in figure 2.

Thank you for this comment. We agree that labelling of the components improves the quality and relevance of the figure.

The figure is updated accordingly, now including labels indicating the components.

**Response to comments of Anonymous Referee #4**

The research in the paper is original, well conceived, and of interest to the readers of this journal. The other reviewers have done a good job in providing a detailed review.

Thank you, we are grateful to read this positive comment.

I only have one comment. The reviews advocate maximizing the torque coefficient rather than maximizing the power coefficient for this particular case. They further demonstrate that this provide more energy for the system considered since hydraulic components in this particular case are more efficient at higher torque and lower speed.

It is important to point out that while the torque coefficient optimizing approach is advantageous in this particular case, it is not true in general. The overall efficiency of a hydraulic pump or motor is the product of the mechanical efficiency and the volumetric efficiency. Depending on the particulars of the unit and its operating conditions, either of these might be dominant. A truly rigorous approach would be to optimize the system power coefficient with all losses included. This would work for any case.

In their final version the authors must clearly state that the torque coefficient optimization approach they advocate is true in this case, but is not true in general.

The authors completely agree with the referee. Only two cases are considered in this paper, namely: operation at rotor maximum torque, and maximum power coefficient. For the specific case presented in the paper, it is found that the maximum torque case results in the highest overall drivetrain efficiency. However, this claim can by no means be generalized for other wind turbines with hydraulic drivetrains. A more rigorous approach would indeed be to optimize the ideal below-rated operational trajectory subject to all component characteristics. However, to perform a more concise analysis, only the two given trajectories are evaluated. For other set-ups it could indeed be the case that a different below-rated operating strategy is beneficial.

The consideration of the reviewer and our response is processed and included in Section 4.1.2.

**Response to comments of Anonymous Referee #5**

This is an interesting paper that deals with the use of hydraulic transmission systems to enable centralized conversion of wind power into electricity in offshore wind farms. A key advantage of this approach is the significant reduction in the nacelle weight. The paper is relevant and very timely given the present drive by industry to develop larger and heavier rotors. The following is a summary of the main comments that need to be addressed by the authors before the paper may be published:

The authors thank the reviewer for his positive comment and considerations raised.

1. Figure 5 may be deleted without affecting the quality of the paper.

   Indeed, the figure could be deleted. However, seen the journal we are publishing in is not primarily focussed on hydraulics, we think that the figure is insightful and serves readers from various disciplines.

   The authors would like to leave the figure as-is in the manuscript.

2. Table 1: It is also convenient for the reader to include the rated power of the motors

   Thank you for this comment, this is indeed a nice addition.

   Table 1 now includes the available power range of all the drivetrain components.

3. Page 6: The process of matching the pump, motor and Pelton turbine to the available wind turbine should be elaborated in further detail.

   Thank you for this comment. A description of the component matching process is not provided in detail in this paper, as the aim of the authors in this paper is to focus on the modeling and control design of the considered hydraulic drivetrain. One of the authors devoted his PhD to the system design of the presented drivetrain [9], and the authors refer the reviewer to this work for further details.

   The reference to the PhD thesis, including an elaborate description of the component matching process, is now also included in Section 2.2.

4. Page 9: specify the aerofoil data used in plotting Fig. 7.

The presented power and torque coefficient curves are obtained from a Bladed wind turbine model which was shared confidentially with DOT. The model includes the requested airfoil data, but for reasons of confidentiality, this information cannot be shared publicly. However, the complete data set including the resulting power, torque and thrust coefficient tables (as a function of tip-speed ratio and blade pitch) is publicly available as an external asset [10].

Section 3.1.1 now includes a reference to the externally available data set which includes power, torque and thrust coefficient tables.

5. Fig. 14: for ease of comparison, the two plots should have the same colour scale for the mechanical efficiency.

We partly agree with the reviewer's remark. The color scales of the left and right plots range from 0.7 - 1.0 and 0.05 - 1.0, respectively. For plots with equal data (efficiency) ranges, we agree with the reviewer that equalizing them would enhance the readability. However, by using the same color bars for the presented plots would make the left plot less convenient to read by a lack of contrast (especially in grayscale). For this reason, the authors decided to leave the plots unchanged.

6. Figure 15: Possible design amendments to the system to enhance the overall conversion efficiency should be elaborated in further detail.

A similar point is posed by reviewer #4, but focuses on the control aspect, and the authors refer the reviewer to our response in the previous section. In summary: for the considered system, only two scenarios are evaluated (maximum rotor power and maximum rotor torque trajectories) to provide a concise evaluation of both strategies on the overall drivetrain efficiency. Indeed, a more rigorous approach would be to optimize the ideal below-rated operational trajectory subject to all component characteristics.

From a system design perspective, the presented prototype hydraulic wind turbine has the goal of showing the feasibility of a wind turbine with a hydraulic drivetrain, and is not meant to be kept as-is. In response to the first question of reviewer #2, we added the efficiency numbers of the current set-up to Section 2.1 ($30-45$ % below-rated, $45$ % above-rated). Also, in Section 2.1, it is stated that the set-up allows for prototyping, and provides a proof of concept for faster development towards the ideal DOT concept. It is known that the additional components and energy conversions result in a reduced overall efficiency. For an overall increased efficiency, the amount of energy conversion steps need to be reduced. It is stated in the conclusion that by discarding the oil loop in the ideal DOT concept, only including a single water pump in the nacelle, the control design process is simplified and the overall drivetrain efficiency should be greatly improved.

Regarding the changes already made to the manuscript, the authors think that the point raised by the reviewer is clarified.

Minor comments:

1. Figure 1 should ideally be presented on the same page where it is being referred to in the text.
2. Page 6, line 13 - remove coma after in such a way.
3. Eqt. (11) may be deleted as derived of Eqt (12) is well known.
4. Page 12, line 9 - remove coma after into the system.

Thank you for pointing out these minor remarks.

1. We agree with the reviewer, however, during typesetting the paper will be converted to a two-column format, and thus the complete mark-up will be changed again. We tried to make the figure positions as convenient as possible, but we are reticent on putting too much effort in this for now. However, we certainly keep this comment in mind for all figures during the mark-up of the final version of the manuscript.

2. Thank you, we corrected this.

3. We agree with the reviewer and deleted the equation.

4. Thank you, we corrected this.

$$\tau_{\mathrm{r}} = \frac{1}{2}\rho_{\mathrm{air}}\pi R^3 U^2 C_{\mathrm{p}}(\lambda,\beta)/\lambda, \tag{2}$$

where the density of air $\rho_{\mathrm{air}}$ is taken as a constant value of  $1.225$ kg m⁻³, $U$ is the velocity of the upstream wind, and $R$ is the blade length of  $22$ m. The power coefficient $C_{\mathrm{p}}$ represents the fraction between the captured rotor power $P_{\mathrm{r}}$ and the

[revised manuscript text omitted]

$$\underline{\zeta_{\mathrm{Q}}} \quad = \frac{1}{2R_{\mathrm{H}}}\sqrt{\frac{L_{\mathrm{H}}}{C_{\mathrm{H}}}}.$$

Finally, the differential equation defined by Eq. (15) is expressed as a transfer function

$$G_{\mathrm{Q}/\Delta\mathrm{p}}(s) = \frac{1/L_{\mathrm{H}}}{s + (R_{\mathrm{H}}/L_{\mathrm{H}}) + 1/(C_{\mathrm{H}}L_{\mathrm{H}}s)} \equiv \frac{s/L_{\mathrm{H}}}{s^2 + (R_{\mathrm{H}}/L_{\mathrm{H}})s + 1/(C_{\mathrm{H}}L_{\mathrm{H}})}, \tag{21}$$

and the same is done for Eq. (18)

$$G_{\Delta\mathrm{p}/\mathrm{Q}}(s) = \frac{1/C_{\mathrm{H}}}{s + 1/(R_{\mathrm{H}}C_{\mathrm{H}}) + 1/(C_{\mathrm{H}}L_{\mathrm{H}}s)} \equiv \frac{s/C_{\mathrm{H}}}{s^2 + 1/(R_{\mathrm{H}}C_{\mathrm{H}})s + 1/(C_{\mathrm{H}}L_{\mathrm{H}})}. \tag{22}$$

The transfer functions defined in Eqs. (21) and (22) show the characteristics of an inverted notch with $+1$ and $-1$ slopes on the left and right side of the natural frequency, respectively. This physically means that exciting the system pressure results
10 in a volume velocity change predominantly at the system natural frequency for the former mentioned case. An illustrative Bode plot is given in Sect. 4.2.1. Exciting the flow results in amplification/transmission to pressure in a wider frequency region. This effect is a result of the inverse proportionality between the damping coefficients $\zeta_{\mathrm{Q}}$ and $\zeta_{\mathrm{P}}$ (see Appendix B).

**3.2.2 Drivetrain model derivation**

15 A dynamic model of the DOT500 drivetrain is derived by application of the theory presented in the previous section. The drivetrain is defined from the rotor up to the spear valve, and the following assumptions are made:

- Because of the high torque to inertia ratio of hydraulic components (Merritt, 1967), the dynamics of oil pumps and motors are disregarded and taken as analytic relations;
- Because of the longer line length and higher compressibility of oil compared to the shorter water column, the high-
20   pressure oil line is more critical for control design, and a dynamic model  is implemented for this column only;
-
- The fluids have a constant temperature.

The dynamic system is governed by the following differential equations

$$\mathcal{V} = \mathcal{V}_{\mathrm{in}} - \mathcal{V}_{\mathrm{out}}, \qquad \dot{\mathcal{V}} = Q = Q_{\mathrm{in}} - Q_{\mathrm{out}}, \tag{23}$$

$$\Delta p_{\mathrm{h}} = \underbrace{\left(\frac{J_{\mathrm{r}}\eta_{\mathrm{m,h}}}{V_{\mathrm{p,h}}^2 \eta_{\mathrm{v,h}}} + L_{\mathrm{H}}\right)}_{L_{\mathrm{R}}^*}\dot{Q}_{\mathrm{in}} + R_{\mathrm{H}}(Q_{\mathrm{in}} - Q_{\mathrm{out}}) + \frac{K_{\mathrm{f}}}{V_{\mathrm{H}}}\frac{K_{\mathrm{e}}}{V_{\mathrm{H}}}(\mathcal{V}_{\mathrm{in}} - \mathcal{V}_{\mathrm{out}}) = L_{\mathrm{R}}^*\dot{Q}_{\mathrm{in}} + R_{\mathrm{H}}(Q_{\mathrm{in}} - Q_{\mathrm{out}}) + \frac{1}{C_{\mathrm{H}}}\mathcal{V}, \tag{24}$$

$$\Delta p_{\mathrm{b}} = L_{\mathrm{H}}\dot{Q}_{\mathrm{out}} + R_{\mathrm{H}}(Q_{\mathrm{out}} - Q_{\mathrm{in}}) + \frac{K_{\mathrm{f}}}{V_{\mathrm{H}}}\frac{K_{\mathrm{e}}}{V_{\mathrm{H}}}(\mathcal{V}_{\mathrm{out}} - \mathcal{V}_{\mathrm{in}}) = L_{\mathrm{H}}\dot{Q}_{\mathrm{out}} + R_{\mathrm{H}}(Q_{\mathrm{out}} - Q_{\mathrm{in}}) - \frac{1}{C_{\mathrm{H}}}\mathcal{V}, \tag{25}$$

where $K_e$ is the equivalent bulk modulus including the fluid and line compressibility defined in Eq. (A5), and $\mathcal{V}$ is the net volume inflow to the considered oil line, between the oil pump discharge and oil motor feed port. For convenience, mechanical model quantities are expressed hydraulically in terms of fluid flows and pressure differences over the components. Therefore, the rotor inertia $J_r$ is expressed in terms of fluid induction, and is combined with the hydraulic induction term into $L_R^*$.

5    Both the spear position and pitch angle are modeled by a first-order actuator model

$$\dot{s} = \frac{1}{t_s}(s_{\text{ref}} - s), \tag{26}$$

$$\dot{\beta} = \frac{1}{t_\beta}(\beta_{\text{ref}} - \beta), \tag{27}$$

where $t_s$ and $t_\beta$ are the time constant for the spear valve and pitch actuators, respectively, and the phase loss at the actuator bandwidth is assumed to account for actuation delay effects.

10    The above given dynamic equations are written in a state-space representation as

$$
\begin{bmatrix} \dot{\mathcal{V}} \\ \dot{Q}_{\text{in}} \\ \dot{Q}_{\text{out}} \\ \dot{s} \\ \dot{\beta} \end{bmatrix} = \begin{bmatrix} 0 & 1 & -1 & 0 & 0 \\ -\frac{1}{C_H L_R^*} & -\frac{R_H}{L_R^*} & \frac{R_H}{L_R^*} & 0 & 0 \\ \frac{1}{C_H L_H} & \frac{R_H}{L_H} & -\frac{R_H}{L_H} & 0 & 0 \\ 0 & 0 & 0 & -\frac{1}{t_s} & 0 \\ 0 & 0 & 0 & 0 & -\frac{1}{t_\beta} \end{bmatrix} \begin{bmatrix} \mathcal{V} \\ Q_{\text{in}} \\ Q_{\text{out}} \\ s \\ \beta \end{bmatrix} + \begin{bmatrix} 0 \\ \frac{1}{L_R^*}\Delta p_h \\ -\frac{1}{L_H}\Delta p_b \\ \frac{1}{t_s}s_{\text{ref}} \\ \frac{1}{t_\beta}\beta_{\text{ref}} \end{bmatrix}. \tag{28}
$$

It is seen that the pressure difference over the oil pump and motor appear as inputs, but these quantities cannot be controlled directly. For this reason, linear expressions of the rotor torque and spear valves are defined next. The rotor torque is linearized with respect to the tip-speed ratio, pitch angle and wind speed

15    $\hat{\tau}_r(\bar{\omega}_r, \bar{\beta}, \bar{U}) = k_{\omega_r}(\bar{\omega}_r, \bar{\beta}, \bar{U})\hat{\omega}_r + k_\beta(\bar{\omega}_r, \bar{\beta}, \bar{U})\hat{\beta} + k_U(\bar{\omega}_r, \bar{\beta}, \bar{U})\hat{U}, \tag{29}$

where $(\hat{\cdot})$ indicates a value deviation from the operating point, and $(\bar{\cdot})$ is the value at the operating point (Bianchi et al., 2006). Furthermore,

$$k_{\omega_r}(\omega_r, \beta, U) = \frac{\partial \tau_r}{\partial \omega_r} = c_r R U \frac{\partial C_\tau(\omega_r R/U, \beta)}{\partial \lambda}, \tag{30}$$

$$k_\beta(\omega_r, \beta, U) = \frac{\partial \tau_r}{\partial \beta} = c_r U^2 \frac{\partial C_\tau(\omega_r R/U, \beta)}{\partial \beta}, \tag{31}$$

$$k_U(\omega_r, \beta, U) = \frac{\partial \tau_r}{\partial U} = 2c_r U C_\tau(\omega_r R/U, \beta) + c_r U^2 \frac{\partial C_\tau(\omega_r R/U, \beta)}{\partial \lambda}\frac{\partial \lambda}{\partial U} \tag{32}$$

$$= 2c_r U C_\tau(\omega_r R/U, \beta) - c_r \omega_r R \frac{\partial C_\tau(\omega_r R/U, \beta)}{\partial \lambda}, \tag{33}$$

$$c_r = \frac{1}{2}\rho\pi R^3, \tag{34}$$

where the quantities $k_{\omega_r}$, $k_\beta$ and $k_U$ represent the intrinsic speed feedback gain, the linear pitch gain and the linear wind speed gain, respectively. The intrinsic speed feedback gain can also be expressed as a function of the tip-speed ratio by

$$k_\lambda(\lambda, \beta, U) = k_{\omega_r}(\omega_r, \beta, U)\frac{U}{R}. \tag{35}$$

[Figure]

**Figure 10.** The intrinsic speed feedback gain $k_\lambda(\lambda, \bar{\beta}, \bar{U})$ as function of tip-speed ratio $\lambda$, at a fixed pitch angle and wind speed of $-2$ $-2$ deg and $8$ $8$ m s$^{-1}$. Stable turbine operation is attained for non-positive values of $k_\lambda$.

For aerodynamic rotor stability, the value of $k_\lambda$ needs to be negative. In Figure 10 the intrinsic speed feedback gain $k_\lambda(\lambda, \bar{\beta}, \bar{U})$ is evaluated as a function of the tip-speed ratio at the fine-pitch angle $\beta_0$. For incorporation of the linearized rotor torque in the drivetrain model, Eq. (29) is expressed in the pressure difference over the oil pump

$$\Delta \hat{p}_h(\bar{\omega}_r, \bar{\beta}, \bar{U}) = k^*_{Q_{in}}(\bar{\omega}_r, \bar{\beta}, \bar{U})\hat{Q}_{in} + k^*_\beta(\bar{\omega}_r, \bar{\beta}, \bar{U})\hat{\beta} + k^*_U(\bar{\omega}_r, \bar{\beta}, \bar{U})\hat{U}, \tag{36}$$

where the conversions of the required quantities are given by

$$k^*_{Q_{in}} = k_{\omega_r} \frac{\eta_{m,h}}{V^2_{p,h} \eta_{v,h}}, \qquad k^*_\beta = k_\beta \frac{\eta_{m,h}}{V_{p,h}}, \qquad k^*_U = k_U \frac{\eta_{m,h}}{V_{p,h}}. \tag{37}$$

Similarly, the water line pressure as defined in Eq. (10) is linearized with respect to the spear position and flow through the valve

$$\hat{p}_{w,l}(\hat{Q}_w, \hat{s}) = k_{s,s}(\bar{Q}_w, \bar{s})\hat{s} + k_{s,Q_w}(\bar{Q}_w, \bar{s})\hat{Q}_w, \tag{38}$$

where

$$k_{s,s}(\bar{Q}_w, \bar{s}) = \frac{2Q^2_w \rho_w (s - s_{max}) \tan^2(\alpha/2)}{C^2_d N^2_s \pi^2 \left(D^2_{nz}/4 - (s_{max} - s)^2 \tan^2(\alpha/2)\right)^3} \Bigg|_{\bar{Q}_w, \bar{s}}, \tag{39}$$

$$k_{s,Q_w}(\bar{Q}_w, \bar{s}) = \frac{Q_w \rho_w}{C^2_d N^2_s \pi^2 \left(D^2_{nz}/4 - (s_{max} - s)^2 \tan^2(\alpha/2)\right)^2} \Bigg|_{\bar{Q}_w, \bar{s}}. \tag{40}$$

The pressure difference over the oil motor is defined in terms of the water line pressure which is a function of the spear position

$$\Delta \hat{p}_{b} = \frac{1}{c_{m,bk}} \Delta \hat{p}_{k} \approx \frac{1}{c_{m,bk}} \hat{p}_{w,l}(s) = \frac{1}{c_{m,bk}} \left( k_{s,s}(\bar{Q}_{w}, \bar{s})\hat{s} + k_{s,Q_{w}}(\bar{Q}_{w}, \bar{s})\hat{Q}_{w} \right), \tag{41}$$

where the mechanical and volumetric conversion factors from oil to water pressure and flow are defined as

$$c_{m,bk} = \frac{V_{p,b}}{V_{p,k}} \eta_{m,k} \eta_{m,b}, \qquad c_{v,bk} = \frac{V_{p,k}}{V_{p,b}} \eta_{v,k} \eta_{v,b} \tag{42}$$

The system defined in Eq. (28) is now presented as a linear state-space system of the form

$$\dot{x} = Ax + Bu + B_{U}\hat{U} \tag{43}$$

$$y = Cx,.$$

 Substitution of the rotor torque and water pressure approximations defined by Eqs. (36) and (41) in Eq. (28), the state $A$, input $B$,  wind disturbance $B_{U}$ and output $C$ matrices are given by

$$A = \begin{bmatrix} 0 & 1 & -1 & 0 & 0 \\ -\frac{1}{C_{H}L_{R}^{*}} & -\frac{R_{H}-k_{Q_{in}}^{*}}{L_{R}^{*}} & \frac{R_{H}}{L_{R}^{*}} & 0 & \frac{k_{\beta}^{*}}{L_{R}^{*}} \\ \frac{1}{C_{H}L_{H}} & \frac{R_{H}}{L_{H}} & -\frac{1}{L_{H}}\left(R_{H}+\frac{c_{v,bk}}{c_{m,bk}}k_{s,Q_{w}}\right) & -\frac{1}{c_{m,bk}L_{H}}k_{s,s} & 0 \\ 0 & 0 & 0 & -\frac{1}{t_{s}} & 0 \\ 0 & 0 & 0 & 0 & -\frac{1}{t_{\beta}} \end{bmatrix}, \qquad B = \begin{bmatrix} 0 & 0 \\ 0 & 0 \\ 0 & 0 \\ \frac{1}{t_{s}} & 0 \\ 0 & \frac{1}{t_{\beta}} \end{bmatrix},$$

$$B_{U} = \begin{bmatrix} 0 \\ \frac{k_{U}^{*}}{L_{R}^{*}} \\ 0 \\ 0 \\ 0 \end{bmatrix}, \qquad C = \begin{bmatrix} 1 & 0 & 0 & 0 & 0 \\ 0 & \frac{1}{V_{p,h}\eta_{v,h}} & 0 & 0 & 0 \\ 0 & 0 & 0 & 1 & 0 \\ 0 & 0 & 0 & 0 & 1 \end{bmatrix}, \tag{44}$$

with the state, input and output matrices

$$x = \begin{bmatrix} \hat{\mathcal{V}} & \hat{Q}_{in} & \hat{Q}_{out} & \hat{s} & \hat{\beta} \end{bmatrix}^{T},$$

$$u = \begin{bmatrix} \hat{s}_{ref} & \hat{\beta}_{ref} \end{bmatrix}^{T}, \tag{45}$$

$$y = \begin{bmatrix} \hat{\mathcal{V}} & \hat{\omega}_{r} & \hat{s} & \hat{\beta} \end{bmatrix}^{T}.$$

The dynamic model derived in this section is used in Sect. 4.2 to come up with an active spear valve torque control strategy in the near-rated region.

[Figure]

**Figure 11.** Schematic diagram of the DOT500 control strategy. When the control error $e$ is negative, the controllers saturate at their minimum or maximum setting. In the near-rated operating region, the rotor speed is actively regulated to $\omega_{r,s}$ by generating the spear position control signal $s_{ref}$, influencing the fluid pressure and the system torque. When the spear valve is at its rated minimum position, the gain-scheduled pitch controller generates a pitch angle set point $\beta_{ref}$ to regulate the rotor speed at its nominal value $\omega_{r,\beta}$.

**4 Controller design**

In this section designs are presented for control in the below- and near-rated operating region. A schematic diagram of the overall control system is given in Figure 11. It is seen that the turbine is controlled by two distinct Proportional-Integral
5 (PI) controllers, a spear valve torque and blade pitch controller, acting on individual rotor speed set point errors $e_s$ and $e_\beta$, respectively. As both controllers have a common control objective of regulating the rotor speed and are implemented in a decentralized way, it is ensured that they are not active simultaneously. The gain-scheduled pitch controller is designed and implemented in a similar way as in conventional wind turbines (Jonkman et al., 2009), and is therefore not further elaborated in this paper. The spear valve torque controller, however, is non-conventional and its control design is outlined in this section.
10 For the below-rated control design a passive torque control strategy is employed, of which a description of is given in Sect. 4.1. Subsequently, in Sect. 4.2, the in-field active spear valve control implementation is evaluated using the dynamic drivetrain model.

**4.1 Passive below-rated torque control**

The passive control strategy for below-rated operation is described in this section. Conventionally, in below-rated operating
15 conditions, the power coefficient is maximized by regulating the tip-speed ratio at $\lambda_0$ using generator torque control. Generally, the maximum power coefficient tracking objective is attained by implementing the feed-forward torque control law

$$\tau_{sys} = \frac{\rho_{air}\pi R^5 C_{p,max}}{2\lambda^3}\omega_r^2 = K_r\omega_r^2,\tag{46}$$

[revised manuscript text omitted]
_{\mathrm{m,h}}(\omega_{\mathrm{r}}, \tau_{\mathrm{sys}}) = \frac{\eta_{\mathrm{t,h}}(\omega_{\mathrm{r}}, \tau_{\mathrm{sys}})}{\eta_{\mathrm{v,h}}}, \tag{51}$$

where $\eta_{\mathrm{v,h}}$ is taken as $0.98$, and the result is presented in Figure 13 (left). The plotted data points (dots) are interpolated on a mesh grid using a regular grid linear interpolation method from the Python SciPy interpolation toolbox (Scipy.org, 2017). Operating cases 1 and 2 are indicated by the solid lines. The same procedure is performed for the data supplied with the oil motor, of which the result is presented in Figure 13 (right), where $\eta_{\mathrm{v,b}}$ is taken as $0.98$. As concluded from the efficiency curves, hydraulic components are generally more efficient in the low-speed high-torque/pressure region. It is immediately clear that for both the oil pump as well as the motor, operating the rotor at a lower tip-speed ratio (case 2) is beneficial from a component efficiency perspective.

The drivetrain efficiency analysis for both operating cases is given in Figure 14. The lack of efficiency data at lower rotor speeds in the left plot of Figure 14 (case 1) is due to unavailability of data at lower pressures. From the resulting plot it is concluded that the overall drivetrain efficiency for case 2 is higher and more consistent compared to case 1. The consistency of the total drivetrain efficiency is advantageous for control, as this  enables passive torque control to maintain a constant tip-speed ratio. As a result of this observation, the focus is henceforth shifted to the implementation of a torque control strategy tracking the maximum torque coefficient (case 2).

It should be stressed that this operational strategy is beneficial for the considered drivetrain, but can by no means be generalized for other wind turbines with hydraulic drivetrains. As the overall efficiency of hydraulic components is a product of mechanical and volumetric efficiency, a more rigorous approach would be to optimize the ideal below-rated operational

[Figure]

**Figure 14.** Comparison of the total drivetrain efficiency for operating cases 1 and 2. It is observed that the total efficiency is higher in the complete below-rated region for case 2. Also the efficiency over all rotor speeds is more consistent, enabling passive torque control using a constant nozzle area $A_{\mathrm{nz}}$.

trajectory subject to all component characteristics. However, to perform a more concise analysis, only the two given trajectories are evaluated.

A stability concern for operation at the maximum torque coefficient needs to be highlighted. For stable operation, the value of $k_\lambda$ needs to be negative. As shown in Figure 10, it  is seen that the stability boundary is located at a tip-speed ratio of $5.9$. Operation at a lower tip-speed ratio results in unstable turbine operation and deceleration of the rotor speed to standstill. However, as concluded in (Schmitz et al., 2013), hydraulic drivetrains can compensate for this theoretical instability, allowing operation at lower tip-speed ratios. Therefore, the case 2 torque control strategy  is designed for the theoretical calculated minimum tip-speed ratio of $5.9$, and in-field test results need to confirm the practical feasibility of the implementation.

**4.2 Active near-rated torque control**

A feedback hydraulic torque control is derived for near-rated operation in this section. To this end, active spear position control is employed to regulate the rotor speed. The effect on fluid resonances is analyzed, as these are possibly excited by an increased rotor speed control bandwidth. The in-field tests with corresponding control implementations are performed prior to the theoretical dynamic analysis of the drivetrain. For this reason, the control design and tunings used in-field are evaluated and possible improvements are highlighted. Sections 4.2.1 and 4.2.2 define the modeling parameters of the oil column and spear valve actuator, which is used in Sect. 4.2.3 for spear valve torque controller design.

**4.2.1 Defining the hydraulic model parameters**

The high-pressure oil line in the DOT500 drivetrain is considered to contain SAE30 oil, with a density of $\rho_\text{o} = 900$ $\rho_\text{o} = 900$ kg m$^{-3}$, and an effective bulk modulus of $K_\text{f,o} = 1.5 \cdot 10^9$ Pa $K_\text{f,o} = 1.5$ GPa. The hydraulic line is cylindrical with a length of $L_\text{l} = 50$ m and $L_\text{l} = 50$ m, a radius of $r_\text{l} = 50 \cdot 10^{-3}$ m. $r_\text{l} = 43.3$ mm and a bulk modulus of $K_\text{l} = 0.80$ GPa (Hružík et al., 2013). According to Eq. A5, the equivalent bulk modulus becomes $K_\text{e} = 0.52$ GPa. The dynamic viscosity of SAE30 oil is taken at a fixed temperature of $20\,^\circ$C, where it reads a value of $\mu = 240 \cdot 10^{-3}$ Pa $\mu = 240$ mPa s. With this data the hydraulic inductance, resistance and capacitance have calculated values of $L_\text{H} = 5.730 \cdot 10^6$ $L_\text{H} = 7.64 \cdot 10^6$ kg m$^{-4}$, $R_\text{H} = 4.889 \cdot 10^6$ $R_\text{H} = 8.69 \cdot 10^6$ kg m$^{-4}$ s$^{-1}$ and $1/C_\text{H} = 3.820 \cdot 10^9$ $1/C_\text{
[revised manuscript text omitted]